# KAT2-mediated acetylation switches the mode of PALB2 chromatin association to safeguard genome integrity

Marjorie Fournier[1]*[†], Amélie Rodrigue[2], Larissa Milano[2], Jean-Yves Bleuyard[1], Anthony M Couturier[1], Jacob Wall[1], Jessica Ellins[3], Svenja Hester[1,4], Stephen J Smerdon[5‡], László Tora[6,7,8,9], Jean-Yves Masson[2]*, Fumiko Esashi[1]*

[1]Sir William Dunn School of Pathology, University of Oxford, Oxford, United Kingdom; [2]CHU de Québec Research Center, Oncology Division; Department of Molecular Biology, Medical Biochemistry and Pathology, Laval University Cancer Research Center, Québec, Canada; [3]Department of Biochemistry, University of Oxford, Oxford, United Kingdom; [4]Advanced Proteomics Facility, University of Oxford, Oxford, United Kingdom; [5]The Francis Crick Institute, London, United Kingdom; [6]Institut de Génétique et de Biologie Moléculaire et Cellulaire, Illkirch, France; [7]Centre National de la Recherche Scientifique, Illkirch, France; [8]Institut National de la Santé et de la Recherche Médicale, Illkirch, France; [9]Université de Strasbourg, Illkirch, France

*For correspondence:
marjorie.fournier@bioch.ox.ac.uk (MF);
Jean-Yves.Masson@crchudequebec.ulaval.ca (JYM);
fumiko.esashi@path.ox.ac.uk (FE)

Present address: [†]Department of Biochemistry, University of Oxford, Oxford, United Kingdom; [‡]Institute of Cancer and Genomic Sciences, University of Birmingham, Edgbaston, United Kingdom

Competing interest: The authors declare that no competing interests exist.

**Summary** The tumour suppressor PALB2 stimulates RAD51-mediated homologous recombination (HR) repair of DNA damage, whilst its steady-state association with active genes protects these loci from replication stress. Here, we report that the lysine acetyltransferases 2A and 2B (KAT2A/2B, also called GCN5/PCAF), two well-known transcriptional regulators, acetylate a cluster of seven lysine residues (7K-patch) within the PALB2 chromatin association motif (ChAM) and, in this way, regulate context-dependent PALB2 binding to chromatin. In unperturbed cells, the 7K-patch is targeted for KAT2A/2B-mediated acetylation, which in turn enhances the direct association of PALB2 with nucleosomes. Importantly, DNA damage triggers a rapid deacetylation of ChAM and increases the overall mobility of PALB2. Distinct missense mutations of the 7K-patch render the mode of PALB2 chromatin binding, making it either unstably chromatin-bound (7Q) or randomly bound with a reduced capacity for mobilisation (7R). Significantly, both of these mutations confer a deficiency in RAD51 foci formation and increase DNA damage in S phase, leading to the reduction of overall cell survival. Thus, our study reveals that acetylation of the ChAM 7K-patch acts as a molecular switch to enable dynamic PALB2 shuttling for HR repair while protecting active genes during DNA replication.

## Editor's evaluation

The manuscript provides fundamental insights into the role of acetylation of PALB2, a protein involved in Fanconi anemia and homologous recombination though its association with BRCA1 and BRCA2. The evidence that PALB2 acetylation regulates its nuclear mobility is multi-faceted and convincing and the major strength is the definition of the role of de-acetylation of the ChAM domain of PALB2 to mobilize the protein under genotoxic stress. Individuals with an interest in genome stability will be the audience for this important study.

## Introduction

PALB2, the partner and localizer of the breast cancer susceptibility 2 protein (BRCA2) (*Xia et al., 2006*), plays essential roles in the maintenance of cellular homeostasis and disease prevention in humans. Biallelic mutations in *PALB2* cause Fanconi anaemia (FA), a rare genetic disorder characterised by bone marrow failure, developmental abnormalities, and an increased incidence of childhood cancers (*Reid et al., 2007*; *Xia et al., 2007*). Hereditary monoallelic *PALB2* mutations also increase the risk of breast and pancreatic cancer (*Jones et al., 2009*; *Rahman et al., 2007*), similarly to inherited *BRCA1* and *BRCA2* mutations (*ODonovan and Livingston, 2010*). The physiological importance of PALB2 is further highlighted by the recent large-scale functional analysis of PALB2 mutations in cancer patients (*Boonen et al., 2019*; *Rodrigue et al., 2019*; *Wiltshire et al., 2020*). Canonically, PALB2 works together with BRCA1 and BRCA2 to promote error-free repair of highly genotoxic double-strand DNA breaks (DSBs) by homologous recombination (HR) (*Ducy et al., 2019*). In this process, BRCA1 acts as a DNA damage sensor, which in turn recruits PALB2 and BRCA2 to sites of DNA damage. Subsequently, the essential RAD51 recombinase is recruited to form nucleoprotein filaments, which catalyse the strand invasion and homology search phases of HR repair (*Sy et al., 2009b*; *Xia et al., 2006*).

Besides the role of PALB2 in promoting HR, our recent study revealed a repair-independent role of PALB2 in protecting transcriptionally active chromatin during DNA replication (*Bleuyard et al., 2017b*). This role of PALB2 is mediated through its high-affinity binding partner the MORF-related gene on chromosome 15 protein (MRG15), which recognises an epigenetic marker of active genes, histone H3 trimethylated at lysine 36 (H3K36me³), via its N-terminal chromodomain (*Bleuyard et al., 2017b*; *Hayakawa et al., 2010*; *Sy et al., 2009a*). Moreover, PALB2 intrinsic chromatin association is reinforced by its chromatin-association motif (ChAM), an evolutionarily conserved domain uniquely found in PALB2 orthologues, which directly binds to nucleosomes (*Bleuyard et al., 2012*). Notably, our genome-wide chromatin immunoprecipitation coupled to high-throughput sequencing (ChIP-seq) analyses revealed that PALB2 associates with a small fraction of actively transcribed genes. Notably, locus-specific analyses through ChIP followed by quantitative PCR (ChIP-qPCR) showed a decrease in PALB2 association with these genes upon exposure to an inhibitor of DNA topoisomerase I (TOP1), camptothecin (CPT), suggesting that the mode of PALB2 chromatin association is actively regulated (*Bleuyard et al., 2017b*). Despite these observations, the regulatory mechanism by which PALB2 switches between different modes of chromatin association (i.e. damage-induced association to promote HR repair versus steady-state association to protect active genes during DNA replication) remains unclear.

Numerous studies in recent decades have provided evidence that reversible post-translational modifications (PTMs), such as phosphorylation, ubiquitylation, SUMOylation, poly(ADP-ribosyl)ation, methylation, and acetylation, are orchestrated to promote genome stability, including the DNA damage response (DDR) (*Dantuma and van Attikum, 2016*). For example, the damage-responsive ATM and ATR kinases mediate phosphorylation of PALB2 at residues S59, S177, and S376, which in turn facilitates PALB2 interaction with BRCA1, RAD51 foci formation, and hence HR repair of DSBs (*Ahlskog et al., 2016*; *Buisson et al., 2017*; *Guo et al., 2015*). Conversely, in G1, the E3 ligase KEAP1-CUL3-RBX1 ubiquitylates PALB2 at K25, a key residue involved in BRCA1 interaction and, in this way, suppresses PALB2-BRCA1 interaction and HR activation (*Orthwein et al., 2015*). Furthermore, our recent work identified PALB2 as a key substrate of the lysine acetyltransferases 2A (KAT2A/GCN5) and 2B (KAT2B/PCAF) (*Fournier et al., 2016*), two well-known transcriptional regulators (reviewed in *Nagy et al., 2010*), in undamaged cells. However, the physiological role of these acetylation events is as yet unknown. Notably, KAT2A/2B use the metabolite acetyl coenzyme A (acetyl-CoA) as a cofactor (*Tanner et al., 2000*), and hence are proposed to fine-tune cellular processes in accordance with the metabolic status of the cell (*Wellen et al., 2009*). Therefore, an understanding of the functional significance of PALB2 acetylation would have important implications in the context of tumorigenesis, as cancer cells frequently exhibit reprogrammed metabolism and elevated genome instability (*Fouad and Aanei, 2017*).

In this study, we investigated the role of KAT2A/2B-mediated lysine acetylation in regulating PALB2. We found that KAT2A/2B acetylate a cluster of seven lysine residues (the 7K-patch) within the PALB2 ChAM. ChAM acetylation enhanced its direct association with nucleosomes. Notably, DNA damage triggered rapid ChAM deacetylation and increased the mobility of PALB2. Importantly, lysine

to glutamine (Q) or arginine (R) substitutions in the 7K-patch rendered PALB2 either constitutively unbound or non-specifically chromatin-bound, resulting in impaired RAD51 foci formation in S phase and reduced cell survival. On the basis of these observations, we propose that PALB2 chromatin association is dynamically regulated by KAT2A/2B in a context-dependent manner, which plays a significant role in the maintenance of genome stability.

# Materials and methods

## Cell culture and cell lines

All cells were grown at 37 °C in an incubator with a humidified atmosphere with 5% $CO_2$. HEK293T cells were grown in Dulbecco's Modified Eagle's Medium (DMEM, Sigma) supplemented with 10% fetal bovine serum (FBS), 100 U/mL penicillin, and 0.1 mg/mL streptomycin. A U2OS Flp-In T-REx P2shRNA cell line (*Bleuyard et al., 2017b*), carrying a doxycycline-inducible shRNA targeting the endogenous *PALB2* 3'-UTR, referred to as U2OS-shPALB2, was used to generate stable isogenic cell lines with constitutive or inducible expression of N3xFLAG- or FLAG-EGFP-PALB2 variants, respectively. U2OS-shPALB2 cells were co-transfected with pOG44 and pcDNA5/FRT GW/N3×FLAG-PALB2 or pcDNA5/FRT/TO/FLAG-EGFP-PALB2 (7Q or 7R), and resultant stable cell lines were selected in DMEM supplemented with 10% FBS, 100 U/mL penicillin, 0.1 mg/mL streptomycin, 10 µg/mL blasticidin, 200 µg/mL hygromycin B (100 µg/mL to maintain the cell lines), and 1 µg/mL puromycin. Stable U2OS-shPALB2 lines carrying the empty pcDNA5/FRT GW/N3×FLAG vector or the pcDNA5/FRT GW/N3×FLAG-PALB2 WT vector have been described previously (*Bleuyard et al., 2017b*). All cell lines tested negative for *Mycoplasma* contamination using the MycoAlert detection kit (Lonza).

## Antibodies

The primary antibodies used for western blot (WB: with their respective working dilutions) were: anti-FLAG (Sigma F1804, mouse, WB: 1/1000), anti-pan-acetyl lysine (AcK) (Cell Signaling Technology 9441 S, rabbit, WB: 1/1000), anti-PALB2 (Bethyl A301-246A, rabbit, WB: 1/500; in-house antibody raised in rabbit (*Rodrigue et al., 2019*), WB: 1/5000), anti-BRCA2 (Millipore OP95, mouse, WB: 1/1000), anti-RAD51 (*Yata et al., 2014*) (7946, rabbit, WB: 1/5000), anti-lamin A (Sigma L1293, rabbit, WB: 1/2000), anti-γ-H2A.X (Millipore 05–636, mouse, WB: 1/1000), anti-GFP (Sigma G1544, mouse, WB: 1/1000), anti-histone H3 (Bethyl A300-823A, rabbit, WB: 1/1000), anti-MRG15 (Cell Signaling Technology D2Y4J, rabbit, WB: 1/1000), anti-BRCA1 (Sigma OP107, mouse, WB: 1/1000), anti-GST (Santa Cruz Biotechnology sc-138, mouse, WB: 1/1000), biotin-HRP conjugated (Sigma A0185, mouse, WB: 1/1000), anti-KAT2A/GCN5 (Cell Signaling Technology 3305, rabbit, WB: 1/1000), anti-α-tubulin (Cell Signaling Technology 3873, mouse, WB: 1/2000) and anti-vinculin (Sigma V9131, mouse, WB: 1/200,000). Secondary antibodies coupled with horseradish peroxidase (HRP): goat anti-mouse (Dako P0447, 1/1000; Jackson ImmunoResearch 515-035-062, 1/20,000), goat anti-rabbit (Dako P0448, 1/1000; Jackson ImmunoResearch 111-035-144, 1/20,000). Antibodies used for immunofluorescence (IF) were: anti-γ-H2A.X (Millipore 05–636, mouse, IF: 1/2000) and anti-RAD51 (BioAcademia 70–001, rabbit, IF: 1/1000). Alexa Fluor conjugated secondary antibodies: goat anti-mouse (Invitrogen A-11001, IF: 1/1000; or Invitrogen A-11017, IF: 1/400) and goat anti-rabbit (Invitrogen A-11011, IF: 1/1000). For ChIP, control IgG (Jackson Immunoresearch 015-000-003, mouse) and anti-FLAG (Sigma F1804, mouse) were used.

## Plasmids

For bacterial expression, full-length PALB2 and fragments 1–4 were PCR amplified using primer pairs, numbered 1 and 8 (full length), 1 and 2 (Fr. 1), 3 and 4 (Fr. 2), 5 and 6 (Fr. 3), or 7 and 8 (Fr. 4) listed in *Table 1*, from pCMV-SPORT6-PALB2 (IMAGE clone 6045564, Source BioSciences) and cloned into the BamHI/NotI sites of the pGEX-6P-1 vector (GE Healthcare). For mammalian expression of ChAM fragments of varying lengths, PALB2 cDNA was first PCR amplified using primer pairs, numbered 9 and 11 (#1), 9 and 12 (#2), 10 and 12 (#3), 10 and 11 (#4), or 10 and 13 (#5) listed in *Table 1*, cloned into the BamHI/XhoI sites of the pENTR3C Gateway entry vector (Thermo Fisher Scientific), and subsequently transferred to pcDNA-DEST53 (Invitrogen) using Gateway cloning. PALB2 Q and R missense mutations were introduced by inverse PCR, where 5'-phosphorylated oligonucleotides containing the desired mutations were used to create blunt-ended products, which were then recircularised by

**Table 1.** List of oligonucleotides used in this study.

| Name | Sequence | No. |
|---|---|---|
| PALB2-F1_fo1 | 5'-atggatccatggacgagcctccc-3' | 1 |
| PALB2-F1_re1 | 5'-atgcggccgcattagaacttgtgggcag-3' | 2 |
| PALB2-F2_fo1 | 5'-atggatccgcacaaggcaaaaaaatg-3' | 3 |
| PALB2-F2_re1 | 5'-atgcggccgctgtgatactgagaaaagac-3' | 4 |
| PALB2-F3_fo1 | 5'-atggatccttatccttggatgatgatg-3' | 5 |
| PALB2-F3_re1 | 5'-atgcggccgcagctttccaaagagaaac-3' | 6 |
| PALB2-F4_fo1 | 5'-atggatcctgttccgtagatgtgag-3' | 7 |
| PALB2-F4_re1 | 5'-atgcggccgcttatgaatagtggtatacaaat-3' | 8 |
| PALB2_395_Fo | 5'- actggatcctcttgcacagtgcctg-3' | 9 |
| PALB2_353_Fo | 5'- actggatccaaatctttaaaatctcccagtg-3' | 10 |
| PALB2_450_Re | 5'- tatctcgagttaattttttacttgcatccttatttttta-3' | 11 |
| PALB2_433_Re | 5'- tatctcgagttacaaatgactctgaatgacagc-3' | 12 |
| PALB2_499_Re | 5'- tatctcgagttacaagtcattatcttcagtggg-3' | 13 |
| Patch 1-K-Rev | 5'-tcagagtcatttggatgtcaagaaaaaaggttt-3' | 14 |
| Patch 2-WT-Fwd | 5'-aaaaataaaaataaggatgcaagtaaaaat-3 | 15 |
| Patch 1-R-Rev | 5'-tcagagtcatttggatgtcaggagaagagggttt-3' | 16 |
| Patch 2-R-Fwd_FL | 5'-agaaatagaaatagggatgcaagtagaaatttaaacctttccaat-3' | 17 |
| Patch 1-Q-Rev | 5'-tcagagtcatttggatgtccagcaacaaggttt-3' | 18 |
| Patch 2-Q-Fwd_FL | 5'-caaaatcaaaatcaggatgcaagtcaaaatttaaacctttccaat-3' | 19 |
| Patch 2-Q-Fwd_ChAM | 5'-caaaatcaaaatcaggatgcaagtcaaaattgagcggccgcact-3' | 20 |
| Beta-Actin_in3-fo | 5'-taacactggctcgtgtgacaa-3' | 21 |
| Beta-Actin_in3-re | 5'-aagtgcaaagaacacggctaa-3' | 22 |
| Chr5_TCOF1_peak2_fo | 5'-ctacccgatccctcaggtca-3' | 23 |
| Chr5_TCOF1_peak2_re | 5'-tcagggctctatgaggggac-3' | 24 |
| Chr11_WEE1_mid_fo | 5'-ggccgaggcttgaggtatatt-3' | 25 |
| Chr11_WEE1_mid_re | 5'-ataaccccaaagaacacaggtca-3' | 26 |

intramolecular ligation. For bacteria expression of ChAM missense variants, pGEX4T3-ChAM (*Bleuyard et al., 2017b*) was modified using primer pairs numbered 18 and 20 (7Q), 18 and 15 (3Q4K), 14 and 20 (3K4Q), or 15 and 16 (3R4K) listed in *Table 1*. For PALB2 7Q and 7R full-length missense variants, pENTR3C-PALB2 was modified using primer pairs numbered 18 and 19 (7Q), or 16 and 17 (7R) listed in *Table 1*. To generate N3xFLAG-fusion or FLAG-EGFP-fusion mammalian expression vectors, PALB2 variants were subsequently transferred to pcDNA5/FRT-GW/N3×FLAG using Gateway cloning (*Bleuyard et al., 2017b*), or cloned into the NotI/XhoI sites of pcDNA5/FRT/TO/FLAG-EGFP (*Bleuyard et al., 2017b*).

## DNA damage and drug treatment

For ionising radiation-induced DNA damage, cells were exposed to 4 Gy γ-rays using a [137]Cs source delivering a dose rate of 1.68 Gy/min (Gravatom) or a CellRad X-ray irradiator (Precision X-Ray Inc). KDAC inhibition was performed by treating cells with a cocktail of 5 mM sodium butyrate (NaB, Sigma 303410), 5 µM trichostatin (TSA, Sigma T8552) and 0.5 mM nicotinamide (NaM) for 2 hr at 37 °C. DMSO was used as negative vehicle control.

## siRNA treatment

For KAT2A/GCN5 and KAT2B/PCAF knockdowns, U2OS cells at 30% confluence were transfected using DharmaFECT 1 (Dharmacon) according to the manufacturer's instructions, with 50 pmol each of ON-Targetplus SMARTpools siRNAs targeting KAT2A (Dharmacon L-009722-02-0005) and KAT2B (Dharmacon L-005055-00-0005) in serum-free DMEM. As a negative control, 100 pmol of ON-TAR-GETplus non-targeting pool siRNAs (Dharmacon D001810-10-05) were used. The serum-free medium was replaced with DMEM supplemented with 10% FBS at 24 hr after transfection, and after further 48 hr incubation, the cells were collected by trypsinisation (total of 72 hr siRNA exposure).

## Fluorescence recovery after photobleaching (FRAP)

Cells were plated into CELLview cell culture dishes (Greiner Bio-One) and analysed in phenol red-free Leibovitz's L15 medium (Gibco). FRAP experiments were performed on a spinning-disk confocal microscope (Ultra-View Vox, Perkin Elmer) mounted on an IX81 Olympus microscope with an Olympus 60x1.4 oil PlanApo objective, in a controlled chamber at 37 °C and 5% $CO_2$ (TOKAI HIT stage top incubator). The fluorescence signal was detected using an EMCCD camera (ImagEM, Hamamatsu C9100-13). Cells were bleached in the GFP channel at maximum laser power with a single pulse for 20 ms, within a square region of interest of 5 µm². After bleaching, GFP fluorescence recovery was monitored within the bleached area every second for 40 s. FRAP parameters were controlled using Volocity software 6.0 (Quorum Technologies). FRAP data were fitted and normalised for overall bleaching of the entire cell (whole-cell) using the FRAP plugins in ImageJ/Fiji (https://imagej.net/mbf/intensity_vs_time_ana.htm#FRAP) (*Schindelin et al., 2012*). From the FRAP curve fitting, half-time recovery time values after photobleaching ($t_{1/2}$) were extracted and plotted in GraphPad Prism 7.02 (GraphPad Software), in which statistical analyses were performed.

## Protein purification

FLAG-KAT2A, FLAG-KAT2A catalytic mutant, and FLAG-KAT2B proteins were purified as described previously (*Fournier et al., 2016*). GST-PALB2 full-length and fragments were purified from 1 L of ArcticExpress cells (Agilent Technologies), grown at 37 °C in LB broth medium containing 50 µg/mL ampicillin and 25 µg/mL gentamycin. Protein expression was induced by 0.1 mM IPTG exposure for 24 hr at 13 °C. Cells were collected by centrifugation for 15 min at 1,400 × *g* at 4 °C and washed with ice-cold phosphate-buffered saline (PBS). Cells lysis was performed by 30 min incubation on ice in 15 mL of ice-cold extraction buffer (50 mM Tris-HCl pH 8.0, 150 mM KCl, 1 mM EDTA, 2 mM DTT, 10% glycerol, and protease inhibitor cocktail (PIC, Sigma P2714)) supplemented with 2 mg/mL Lysozyme (Sigma) and 0.2% Triton X-100, followed by sonication. Cell lysates were collected after 45 min centrifugation at 35,000 × *g*, 4 °C. GST-fusion proteins were pulled down with glutathione Sepharose 4B beads (GE Healthcare), pre-washed with ice-cold PBS. After overnight incubation at 4 °C on a rotating wheel, beads were washed three times with ice-cold extraction buffer, three times with 5 mL of ice-cold ATP-Mg buffer (50 mM Tris-HCl pH 7.5, 500 mM KCl, 2 mM DTT, 20 mM $MgCl_2$, 5 mM ATP, and 10% glycerol) to release chaperone binding from the recombinant protein PALB2 and three times with ice-cold equilibration buffer (50 mM Tris-HCl pH 8.8, 150 mM KCl, 2 mM DTT, and 10% glycerol). Proteins were eluted from beads in ice-cold elution buffer (50 mM Tris-HCl pH 8.8, 150 mM KCl, 2 mM DTT, 0.1% Triton X-100, 25 mM L-glutathione, and 10% glycerol).

For GFP-ChAM purification for mass spectrometry analysis, HEK293T cells (3×10⁷ cells) transiently expressing GFP-ChAM were collected by centrifugation for 5 min at 500 × *g*, 4 °C and washed once with ice-cold PBS. Cells were further resuspended in 5 mL ice-cold sucrose buffer (10 mM Tris-HCl pH 8.0, 20 mM KCl, 250 mM sucrose, 2.5 mM $MgCl_2$, 10 mM benzamidine hydrochloride (Benz-HCl) and PIC). After addition of Triton X-100 (Sigma) to a final concentration of 0.3% w/v, the cell suspension was vortexed four times for 10 s at 1 min intervals. The intact nuclei were collected by centrifugation for 5 min at 500 × *g*, 4 °C, and the supernatant was discarded. The nuclear pellet was washed once with ice-cold sucrose buffer and resuspended in ice-cold NETN250 buffer (50 mM Tris-HCl pH 8.0, 250 mM NaCl, 2 mM EDTA, 0.5% NP-40, 10 mM Benz-HCl and PIC). After 30 min incubation on ice, the chromatin fraction was collected by centrifugation for 5 min at 500 × *g*, 4 °C, washed once with 5 mL ice-cold NETN250 buffer and lysed for 15 min at room temperature (RT) in ice-cold NETN250 buffer supplemented 5 mM $MgCl_2$ and 125 U/mL benzonase (Novagen 71206–3). After addition of EDTA and EGTA to respective final concentrations of 5 mM and 2 mM to inactivate the benzonase and

centrifugation for 30 min at 16,100 × $g$, 4 °C, the supernatant was collected as the chromatin-enriched fraction. GFP-ChAM was pulled down using 15 µL GFP-Trap Agarose (Chromotek), pre-washed three times with ice-cold NETN250 buffer and blocked for 3 hr at 4 °C on a rotating wheel with 500 µL ice-cold NETN250 buffer supplemented with 2 mg/mL bovine serum albumin (BSA, Sigma). After 3 hr protein binding at 4 °C on a rotating wheel, the GFP-Trap beads were collected by centrifugation for 5 min at 1,000 × $g$, 4 °C and washed four times with ice-cold NETN250 buffer.

For the analysis of ChAM acetylation upon DNA damage, a GFP-ChAM fusion was transiently expressed from pDEST53-GFP-ChAM for 24 hr in HEK293T cells. Whole-cell extracts (WCEs) were prepared from ~1.5×10$^7$ cells resuspended in NETN150 buffer (50 mM Tris-HCl pH 8.0, 150 mM NaCl, 2 mM EDTA and 0.5% NP-40 alternative [NP-40 hereafter] [Millipore 492018]) supplemented with 1 mM DTT, PIC, lysine deacetylase inhibitor (5 mM NaB), 1 mM MgCl$_2$ and 125 U/mL benzonase. After 30 min incubation on ice, cell debris was removed by 30 min centrifugation at 4 °C, and the supernatant was collected as WCE. WCE was then incubated with 15 µl of GFP-Trap Agarose for GFP-pull down. After 1 hr protein binding at 4 °C on a rotating wheel, GFP-Trap beads were collected by 5 min centrifugation at 500 × $g$ at 4 °C and washed three times with NET150 buffer (50 mM Tris-HCl pH 8.0, 150 mM NaCl and 2 mM EDTA) supplemented with 0.1% NP-40, 1 mM DTT, PIC, 5 mM NaB and 1 mM MgCl$_2$. Proteins were eluted from beads by heating at 85 °C for 10 min in Laemmli buffer supplemented with 10 mM DTT. The proteins were separated by SDS-PAGE and analysed by western blot.

For the nucleosome pull-down assays, GFP-ChAM variants were affinity-purified from HEK293T cells (3×10$^7$ cells) following transient expression. After collecting cells by centrifugation for 5 min at 500 × $g$, 4 °C, the cell pellet was washed twice with ice-cold PBS and resuspended in ice-cold NETN150 buffer supplemented with 10 mM Benz-HCl and PIC. After 30 min incubation on ice, the chromatin was pelleted by centrifugation for 5 min at 500 × $g$, 4 °C, and the supernatant was collected as NETN150 soluble fraction and centrifuged for 30 min at 16,100 × $g$, 4 °C to remove cell debris and insoluble material. For each sample, 10 µL of GFP-Trap Agarose were washed three times with 500 µL ice-cold NETN150 buffer. NETN150 soluble proteins (2.5 mg) in a total volume of 1 mL ice-cold NETN150 buffer were incubated with the GFP-Trap beads to perform a GFP pull-down. After 2 hr incubation at 4 °C on a rotating wheel, the GFP-Trap beads were collected by centrifugation for 5 min at 500 × $g$, 4 °C and washed four times with ice-cold NETN150 buffer. Human nucleosomes were partially purified from HEK293T cells (4×10$^7$ cells), collected by centrifugation for 5 min at 500 × $g$, 4 °C, washed twice with ice-cold PBS and lysed in ice-cold NETN150 buffer supplemented with 10 mM Benz-HCl and PIC. After 30 min of incubation on ice, the chromatin was pelleted by centrifugation for 5 min at 500 × $g$, 4 °C, washed once with ice-cold NETN150 buffer and digested for 12 min at 37 °C with 50 gel units of micrococcal nuclease (NEB) per milligram of DNA in NETN150 buffer containing 5 mM CaCl$_2$, using 200 µL buffer per mg of DNA. The reaction was stopped with 5 mM EGTA and the nucleosome suspension cleared by centrifugation for 30 min at 16,100 × $g$, 4 °C.

## Acetyltransferase assays

Radioactive $^{14}$C-acetyltransferase assays on recombinant proteins were performed by incubating purified GST-PALB2 (full-length and fragments) or purified RAD51 with purified FLAG-KAT2B in the presence of $^{14}$C-labeled acetyl-CoA in the reaction buffer (50 mM Tris-HCl pH 8.0, 10% glycerol, 100 mM EDTA, 50 mM KCl, 0.1 M NaB, PIC, and 5 mM DTT) for 1 hr at 30 °C. The reactions were stopped by addition of Laemmli buffer containing 10% beta-mercaptoethanol, boiled for 5 min, resolved by SDS-PAGE, and stained using Coomassie blue to reveal overall protein distribution. The acrylamide gel was then dried and exposed to phosphorimager to reveal $^{14}$C-labeled proteins. Non-radioactive acetyltransferase assays were performed as described above using cold acetyl-CoA instead. After 1 hr incubation at 30 °C, the reactions were stopped by addition of Laemmli buffer containing 10 mM DTT, boiled for 5 min, resolved by SDS-PAGE, and after Ponceau S staining of the membrane to reveal overall protein distribution, analysed by western blot using anti-acetyl lysine antibody. Acetyltransferase assays performed for mass spectrometry analyses were performed as previously described (*Fournier et al., 2016*).

## Nucleosome pull-down assay

Nucleosome pull-down assays were performed by mixing 250 µg of partially purified nucleosomes and GFP-ChAM variants immobilised on GFP-Trap beads in NETN150 buffer supplemented with

2 mg/mL BSA, followed by 30 min incubation at RT, then 1.5 h incubation at 4 °C, on a rotating wheel. GFP-Trap beads were further washed four times with NETN150 buffer, and samples were analysed by SDS-PAGE and western blot.

## Chemical cell fractionation and whole-cell extract

HEK293T cells transiently expressing GFP-ChAM variants were collected using TrypLE Express reagent (Gibco), washed twice with ice-cold PBS and resuspended in sucrose buffer (10 mM Tris-HCl pH 7.5, 20 mM KCl, 250 mM sucrose, 2.5 mM $MgCl_2$, 10 mM Benz-HCl and PIC), using 1 mL buffer per 100 mg of weighed cell pellet. After addition of Triton X-100 (Sigma) to a final concentration of 0.3% w/v, the cell suspensions were vortexed three times for 10 s at 1 min intervals. The intact nuclei were collected by centrifugation for 5 min at 500 x g, 4 °C, and the supernatant collected as the cytoplasmic fraction. The nuclei pellet was washed once with ice-cold sucrose buffer and resuspended in ice-cold NETN150 buffer (50 mM Tris-HCl pH 8.0, 150 mM NaCl, 2 mM EDTA, 0.5% NP-40, 10 mM Benz-HCl and PIC), using 400 µL buffer per 100 mg of initially weighed cell pellet. After 30 min incubation on ice, the chromatin fraction was collected by centrifugation for 5 min at 500 x g, 4 °C and the supernatant collected as nuclear soluble fraction. The chromatin pellet was washed once with ice-cold NETN150 buffer and finally solubilised for 1 hr on ice in NETN150 buffer containing 2 mM $MgCl_2$ and 125 U/mL Benzonase nuclease (Merck Millipore), using 250 µL buffer per 100 mg of initial weighed cell pellet. Cytoplasmic, nuclear soluble and chromatin-enriched fractions were centrifuged for 30 min at 16,100 x g, 4 °C to remove cell debris and insoluble material. For whole-cell extract, cells were directly lysed in NETN150 buffer containing 2 mM $MgCl_2$ and 125 U/mL Benzonase for 1 hr on ice and centrifuged for 30 min at 16,100 x g, 4 °C to remove cell debris and insoluble material.

## Cell survival

U2OS-shPALB2 cells complemented with FLAG-PALB2 WT or its variants were seeded in 96-wells plates and grown in the presence or absence of 2 µg/mL doxycycline for 4 days at 37 °C. Cell survival was then measured using WST-1 reagent (Roche Applied Science) following manufacturer's protocol.

## Protein structure prediction with AlphaFold2

The predictions of full-length PALB2 (amino acid 1–1186) and the ChAM variants (amino acid 395–450) were conducted via the ColabFold: AlphaFold2 using MMseqs2 (https://colab.research.google.com/github/sokrypton/ColabFold/blob/main/AlphaFold2.ipynb) (*Mirdita et al., 2022*; *Steinegger and Söding, 2017*). The resultant structures were visualised using UCSF Chimera (https://www.cgl.ucsf.edu/chimera/) (*Pettersen et al., 2004*).

## Immunofluorescence microscopy

For γ-H2A.X foci analysis, cells were grown on coverslips and washed with PBS before pre-extraction with 0.1% Triton in PBS for 30 s at RT. Cells were then fixed twice with 4% PFA in PBS, first for 10 min on ice and then for 10 min at RT. After permeabilisation in 0.5% Triton X-100 in PBS for 10 min at RT, cells were blocked with 5% BSA in PBS supplemented with 0.1% Tween 20 solution (PBS-T-0.1) and incubated with anti-γ-H2A.X antibody for 3 hr at RT. After washing with PBS-T-0.1 for 5 min at RT, cells were incubated with secondary antibodies coupled with a fluorophore, washed with PBS-T-0.1 for 5 min at RT, and mounted on slides using a DAPI-containing solution. Cells were analysed on a spinning-disk confocal microscope (Ultra-View Vox, Perkin Elmer) mounted on an IX81 Olympus microscope, with a 40x1.3 oil UPlan FL objective. The fluorescence signal was detected using an EMCCD camera (ImagEM, Hamamatsu C9100-13). Images were processed in Image J (https://imagej.nih.gov/ij/) (*Schneider et al., 2012*).

## Click-iT fluorescent EdU labelling and immunofluorescence microscopy

For γ-H2A.X and RAD51 foci analysis, U2OS-shPALB2 stably expressing FLAG (EV) or FLAG-PALB2 variants were grown on coverslips in the presence of 2 µg/mL doxycycline for 4 or 5 days. When applicable, cells were exposed to irradiation at 4 Gy using a CellRad X-ray irradiator (Precision X-Ray Inc) and returned at 37 °C for 2 hr 30 min. Then, cells were incubated with 10 µM EdU in media for 30 min at 37 °C before washing with PBS and fixation in 4% PFA in PBS for 10 min. After permeabilisation in 0.5% Triton X-100 in PBS for 5 min, cells were blocked with 1% BSA, 10% goat serum in

PBS for 30 min. EdU staining was performed with the Click-iT Alexa Fluor 647 Imaging Kit for 30 min (Invitrogen C10340), using 1/50 the recommended volume of Alexa Fluor azide, and samples were protected from light from this point on. Primary antibody incubation was performed for 1 hr with anti-γ-H2A.X (Millipore 05–636) and anti-RAD51 (BioAcademia 70–001) diluted in PBS-1% BSA at 1:2000 and 1:1000, respectively. Secondary antibodies Alexa Fluor 488 goat anti-mouse (Invitrogen A-11001) and Alexa Fluor 568 goat anti-rabbit (Invitrogen A-11011) were diluted 1:1000 in PBS-1% BSA and applied for 1 hr. Nuclei were stained for 10 min with 4, 6-diamidino-2-phenylindole (DAPI) prior to mounting on slides with ProLong Gold antifade solution (Invitrogen). All immunofluorescence steps were performed at RT with 3 intervening PBS washes. Slide images were acquired on a CellDiscoverer 7 widefield imaging system (Carl Zeiss Microscopy) using a 50 x/1.2 water immersion objective with a ×0.5 magnification changer. Acquired images were processed and analysed using Zeiss ZEN (blue edition) software.

### Chromatin Immunoprecipitation

ChIP was performed according to *Bleuyard et al., 2017a* with modifications. In brief, U2OS-shPALB2 stably expressing FLAG (EV) or FLAG-PALB2 variants were treated with 2 µg/mL doxycycline for 5 days and, where indicated, with 4 Gy IR before being harvested with trypsin and washed twice with PBS. For each ChIP, $2 \times 10^7$ cells were pelleted and resuspended in 2 ml of PBS to be then fixed for 8 min at RT with 1% formaldehyde and quenched for 5 min with 125 mM glycine. After two washes with ice-cold PBS, cells were incubated for 10 min on ice in 1 mL of lysis buffer (10 mM PIPES pH 7.5, 85 mM KCl, 0.5% NP-40, 10 mM Benz-HCl, PIC). Isolated nuclei were pelleted, resuspended in 1 mL of ChIP buffer (20 mM Tris·HCl pH 7.5, 150 mM NaCl, 1 mM EDTA, 0.5 mM EGTA, 1% Triton X-100, 0.1% sodium deoxycholate, 0.1% SDS, 10 mM Benz-HCl, PIC) and incubated for 10 min on ice. Sonication was carried out at 4 °C in 15 ml polystyrene Falcon tubes using a water-bath Bioruptor (Diagenode) on high setting for 40 cycles of 30 s on and 30 s off. Chromatin samples were centrifuged for 10 min at 16,100 × g, 4 °C and pre-cleared by adding 20 µL of protein G Dynabeads (Thermo Fisher 10003D) and incubating for 1 hr at 4 °C on a rotating wheel. From this, 800 µL chromatin was mixed with 10 µg of control mouse IgG (Jackson Immunoresearch 015-000-003) or mouse anti-FLAG antibody (Sigma F1804) in a final volume of 1 mL ChIP buffer and incubated overnight at 4 °C on a rotating wheel. A total of 100 µL of protein G Dynabeads blocked overnight with 5 mg/mL BSA in ChIP buffer was added to each sample. After 2 hr of incubation at 4 °C on a rotating wheel, beads were washed twice with low-salt wash buffer (20 mM Tris·HCl pH 8, 150 mM NaCl, 2 mM EDTA, 1% Triton X-100, 0.1% SDS, 10 mM Benz-HCl, PIC), twice with high-salt wash buffer (20 mM Tris·HCl pH 8, 500 mM NaCl, 2 mM EDTA, 1% Triton X-100, 0.1% SDS, 10 mM Benz-HCl, PIC), once with LiCl wash buffer (20 mM Tris·HCl pH 8, 250 mM LiCl, 1 mM EDTA, 1% NP-40, 1% sodium deoxycholate, 10 mM Benz-HCl, PIC), and twice with TE buffer (10 mM Tris·HCl pH 8.0, 1 mM EDTA). Beads were resuspended in 200 µL elution buffer (TE buffer, 1% SDS) and incubated for 1 hr at 65 °C, with vortexing every 15 min. After overnight cross-link reversal and proteinase K and RNase A treatment, samples were extracted with phenol/chloroform and ethanol-precipitated. ChIP-qPCR analysis was performed on a QuantStudio 7 Pro real-time PCR system (Thermo Fisher) using Applied Biosystems PowerUp SYBR Green Master Mix (Thermo Fisher). Primer sequences are listed in *Table 1*.

### Time-lapse microscopy analysis of PALB2 recruitment to laser-induced DNA damage

For time-lapse microscopy, U2OS-shPALB2 cells expressing FE-PALB2 WT, 7R and 7Q, and depleted of endogenous PALB2 via doxycycline exposure, were micro-irradiated using point bleach mode for 200 ms with a 405 nm UV-laser (100% output) at the following settings: format 512X512 pixels, scan speed 100 Hz, mode bidirectional, zoom 2 X, 16-bit image depth. To monitor the recruitment of FE-PALB2 to laser-induced DNA damage sites, cells were imaged every 30 s for 15 min on a Leica TCS SP5 II confocal microscope driven by Leica LAS AF software. The fluorescence intensity of FE-PALB2 at DNA damage sites relative to an unirradiated area was quantified and plotted over time.

### Sequence alignment analysis

Sequences of PALB2 orthologues from 12 species were retrieved from Ensembl (http://www.ensembl.org) and NCBI (https://www.ncbi.nlm.nih.gov) and aligned using MUSCLE (http://www.ebi.ac.uk/Tools/msa/muscle/).

## Results

### PALB2 is acetylated within a 7K-patch in its ChAM domain

Our previous shotgun mass spectrometry (MS) study identified a number of lysine residues within the central region of PALB2 as KAT2A/2B-dependent endogenous acetylation acceptors in HeLa cells (*Fournier et al., 2016*; *Figure 1A*, highlighted in blue). To confirm the direct acetylation of PALB2 by KAT2A/2B, we performed in vitro acetylation assays using either recombinant full-length PALB2 or a series of PALB2 fragments (*Figure 1A*) with purified KAT2A or KAT2B. Lysine acetylation of full-length PALB2 and fragment 2 (residues 295–610), which encompasses the DNA/chromatin binding region of PALB2, was clearly visible by western blot against acetylated lysine (pan-AcK) or $^{14}$C-autoradiography following in vitro acetylation with KAT2A or KAT2B, but not with a catalytically inactive mutant of KAT2A (KAT2A-ED) (*Figure 1B*, *Figure 1—figure supplement 1A and B*). Our MS analyses of these products identified seven acetylated lysine residues: K436, K437, K438, K441, K443, K445 and K449, in a cluster in the PALB2 chromatin association motif (ChAM), denoted the 7K-patch (*Figure 1C*, highlighted in orange). Notably, while sequence alignment of PALB2 orthologues showed some divergence in the C-terminal region of ChAM, an enrichment of lysine residues was consistently detected (*Bleuyard et al., 2012*; *Figure 1D*). Clusters of lysine residues were also frequently found in other KAT2A/2B substrates (*Fournier et al., 2016*), corroborating our finding that the ChAM C-terminal region is favourably targeted for KAT2A/2B-mediated acetylation.

### The 7K-patch promotes ChAM nucleosome association

To evaluate the impact of the 7K-patch on ChAM chromatin association, we generated a series of GFP-tagged ChAM truncations (*Figure 2A*). Critically, when expressed in HEK293T cells, the ChAM fragment (PALB2 residues 395–450) was found enriched in the chromatin fraction in highly acetylated forms (*Figure 2A and B*, *Figure 2—figure supplement 1A and B*). Further analysis of ChAM truncations revealed that the region containing the 7K-patch was essential for its chromatin association (*Figure 2A and B*). Considering the possibility that the deletion of the 7K-patch might lead to aberrant subcellular localisation and contribute to the reduced chromatin enrichment, we also affinity purified the corresponding fragments from the cytoplasmic fraction of transfected HEK293T cells and assessed their interaction with separately prepared nucleosomes (*Figure 2—figure supplement 1A*). The pull-down assay showed that deletion of the 7K-patch impaired ChAM interaction with nucleosomes (*Figure 2A and C*), corroborating our observations above.

To further assess the direct role of 7K-patch acetylation in nucleosome binding, we produced recombinant GST fusions of ChAM and its variants harbouring glutamine (Q) or arginine (R) substitutions at all seven lysine residues (7Q or 7R, respectively), three conserved lysine residues, K436, K437, and K438 (3Q4K), or the four remaining lysine residues, K441, K443, K445, and K449 (3K4Q) within the 7K-patch (*Figure 2A*). K to Q substitutions nullify the positive charge of the lysine side chains and are therefore commonly considered to mimic acetyl-lysine, albeit with a reduction in the side chain size (*Figure 2—figure supplement 2A*). Conversely, K to R substitution maintains a positive charge but is unable to accept acetylation and, hence, is widely considered to mimic constitutively non-acetyl-lysine, albeit with an increase in the side chain size (*Figure 2—figure supplement 2A*). Our in silico AlphaFold2 modelling suggest that ChAM has a propensity to form an alpha-helical structures with a less defined C-terminal section, potentially allowing it to explore a spectrum of structural configurations for optimum nucleosome binding (*Figure 2—figure supplement 2B*). On the contrary, 7Q or 7R substitutions make this section more defined, potentially constraining its capacity to bind nucleosomes in optimal configurations (*Figure 2—figure supplement 2C and D*).

Significantly, in vitro nucleosome pull-down assays showed that, while the 7R variant largely maintains its interaction with nucleosomes (*Figure 2D and E*), all variants with Q substitutions abolished this nucleosome binding property (*Figure 2F and G*). This observation appeared to be at odds with our observation that ChAM acetylation was most notable on the chromatin-enriched fraction

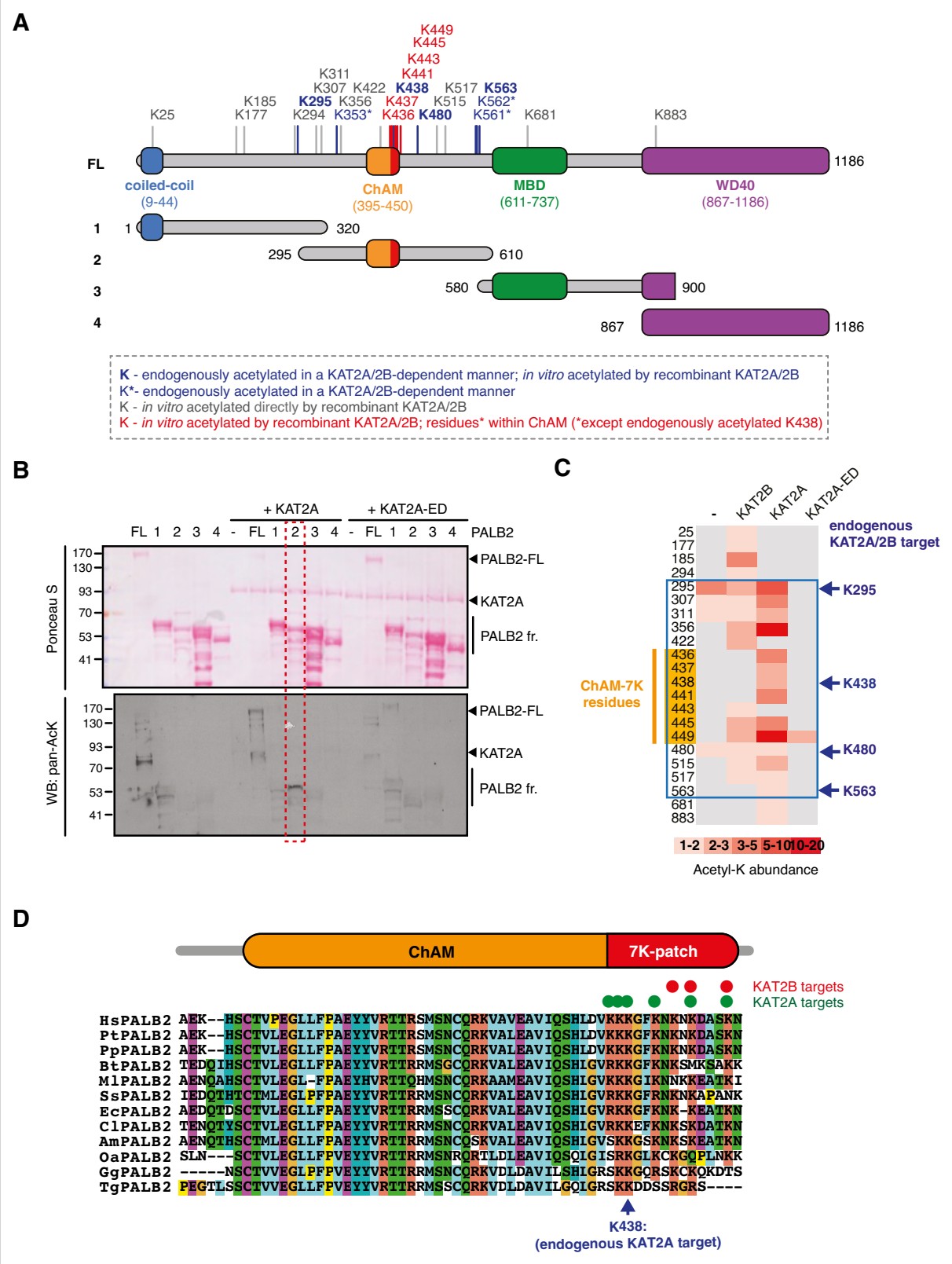

**Figure 1.** KAT2A/2B acetylate PALB2 within a 7K-patch in its ChAM domain. (**A**) PALB2 lysine residues identified as acceptors of KAT2A/2B-dependent acetylation in vivo and in vitro by tandem MS analyses are shown in blue and black, respectively. An asterisk indicates residues that are only detected in endogenous PALB2, but not in PALB2 acetylated in vitro. The acetylated lysine residues within the ChAM are highlighted in red. Full-length PALB2 (FL,~131,3 kDa) and fragments 1–4 used for in vitro acetylation assays are also shown. 1: PALB2 1–320 (36.2 kDa), 2: 295–610 (35.6 kDa), 3: 580–900

*Figure 1 continued on next page*

*Figure 1 continued*

(35.3 kDa) and 4: 867–1186 (35.3 kDa). (**B**) In vitro acetylation of PALB2 by KAT2A and catalytically inactive KAT2A-ED. Purified GST-fusions of PALB2 full-length (158.1 kDa) and fragments 1 (63 kDa), 2 (62.4 kDa), 3 (62.1 kDa), and 4 (62.1 kDa) (depicted in **A**) were incubated with either purified Flag-KAT2A or Flag-KAT2A-ED in the presence of acetyl-CoA, followed by SDS-PAGE. Total and acetylated proteins were visualized by Ponceau S and anti-acetyl lysine (pan-AcK) western blot, respectively. PALB2 fragment 2 acetylation by KAT2A is highlighted with a red dashed box. (**C**) A heat map of acetylated lysine residues, as identified by quantitative MS analysis of in vitro acetylated PALB2 FL or fragment 2 by KAT2B, KAT2A or a catalytically inactive KAT2A (KAT2A-ED). The abundance of each acetyl-lysine was evaluated as previously described (*Fournier et al., 2016*). The lysine residues which were detected as endogenous acetylation acceptor are shown to the left. (**D**) ChAM protein sequences from twelve PALB2 orthologues were aligned using MUSCLE (EMBL-EBI sequence analysis tool) and visualised using the ClustalW software with default colour-coding. Lysine residues acetylated by KAT2A and KAT2B in vitro are respectively highlighted by red and green dots. Hs (*Homo sapiens*, human), Pt (*Pan troglodytes*, chimpanzee), Pp (*Pan paniscus*, bonobo), Bt (*Bos taurus*, cow), Ml (*Myotis lucifugus*, little brown bat), Ss (*Sus scrofa*, wild boar), Ec (*Equus caballus*, horse), Cl (*Canis lupus familiaris*, dog), Am (*Ailuropoda melanoleuca*, giant panda), Oa (*Ovis aries*, sheep), Gg (*Gallus gallus*, red junglefowl), Tg (*Taeniopygia guttata*, zebra finch).

The online version of this article includes the following source data and figure supplement(s) for figure 1:

**Source data 1.** Original images for *Figure 1B*.

**Figure supplement 1.** PALB2 is directly acetylated by KAT2B.

**Figure supplement 1—source data 1.** Original images for *Figure 1—figure supplement 1A*.

**Figure supplement 1—source data 2.** Original images for *Figure 1—figure supplement 1B*.

(*Figure 2—figure supplement 1A and B*). Considering that the size of the glutamine side chain is significantly smaller than the corresponding acetyl-lysine side chain (*Figure 2—figure supplement 2A and D*), we reasoned it highly unlikely that the K to Q substitutions accurately mimic the properties of acetylated ChAM. Collectively, our experimental results showed that K to Q substitutions within the 7K-patch fully perturbed ChAM ability to bind nucleosomes; the 7Q variant is hereafter considered to behave as a ChAM null mutant, rather than an acetyl-mimetic.

## Acetylation within the 7K-patch enhances ChAM nucleosome association

KAT2A/2B associate with chromatin and facilitate transcription by acetylating histones (*Nagy et al., 2010*). Concordantly, the level of acetylation in the GFP-ChAM fragment affinity purified from the chromatin-enriched fraction was notably higher than that from the cytoplasmic or nuclear soluble fraction (*Figure 2—figure supplement 1A and B*). Our MS analysis of the chromatin-associated GFP-ChAM fragment detected acetylation on all seven lysine residues of the 7K-patch (*Figure 3A*, marked with arrows). To accurately assess the impact of the ChAM acetylation, synthetic peptides corresponding to the minimal ChAM (PALB2 residues 395–450) and containing acetyl-lysine at the evolutionarily conserved K436, K437, and/or K438 were generated, and their biochemical properties were characterised. Markedly, nucleosome pull-down assays revealed clear nucleosome binding by acetylated ChAM peptides, but not by their non-acetylated counterpart, where salmon sperm DNA was provided in excess to outcompete ChAM electrostatic interaction with DNA (*Figure 3B*). This effect was readily detectable with the ChAM peptide containing a mono-acetylation at K438, which was detected in our original MS study of the endogenous HeLa KAT2A/2B-acetylome (*Fournier et al., 2016*; *Figures 1A, C, D , and 3A*, highlighted in blue). Together, these results demonstrate that acetylation within its 7K-patch enhances ChAM nucleosome binding.

## PALB2 mobility increases upon deacetylation

Next, we set out to investigate whether lysine acetylation might facilitate PALB2 chromatin association in a cellular context. To evaluate the impact of native PALB2 acetylation, we down-regulated KAT2A/2B activity using siRNA and assessed PALB2 mobility as a surrogate measure of its chromatin association. To this end, a tandem FLAG- and EGFP-tagged full-length wild-type (WT) PALB2 (FE-PALB2) was conditionally expressed in a U2OS cell line in which endogenous *PALB2* was down-regulated by a doxycycline-inducible short hairpin RNA (U2OS-shPALB2) (*Figure 3—figure supplement 1A*), and the mobility of FE-PALB2 was assessed using fluorescence recovery after photobleaching (FRAP). This analysis revealed that KAT2A/2B siRNA treatment led to an increase in FE-PALB2 diffusion rate (reduced FRAP $t_{1/2}$; *Figure 3C and D*, *Figure 3—figure supplement 1B and C*, *Figure 3—videos 1; 2*), concomitant with reduced levels of global acetylation (*Fournier et al., 2016*). Conversely, the

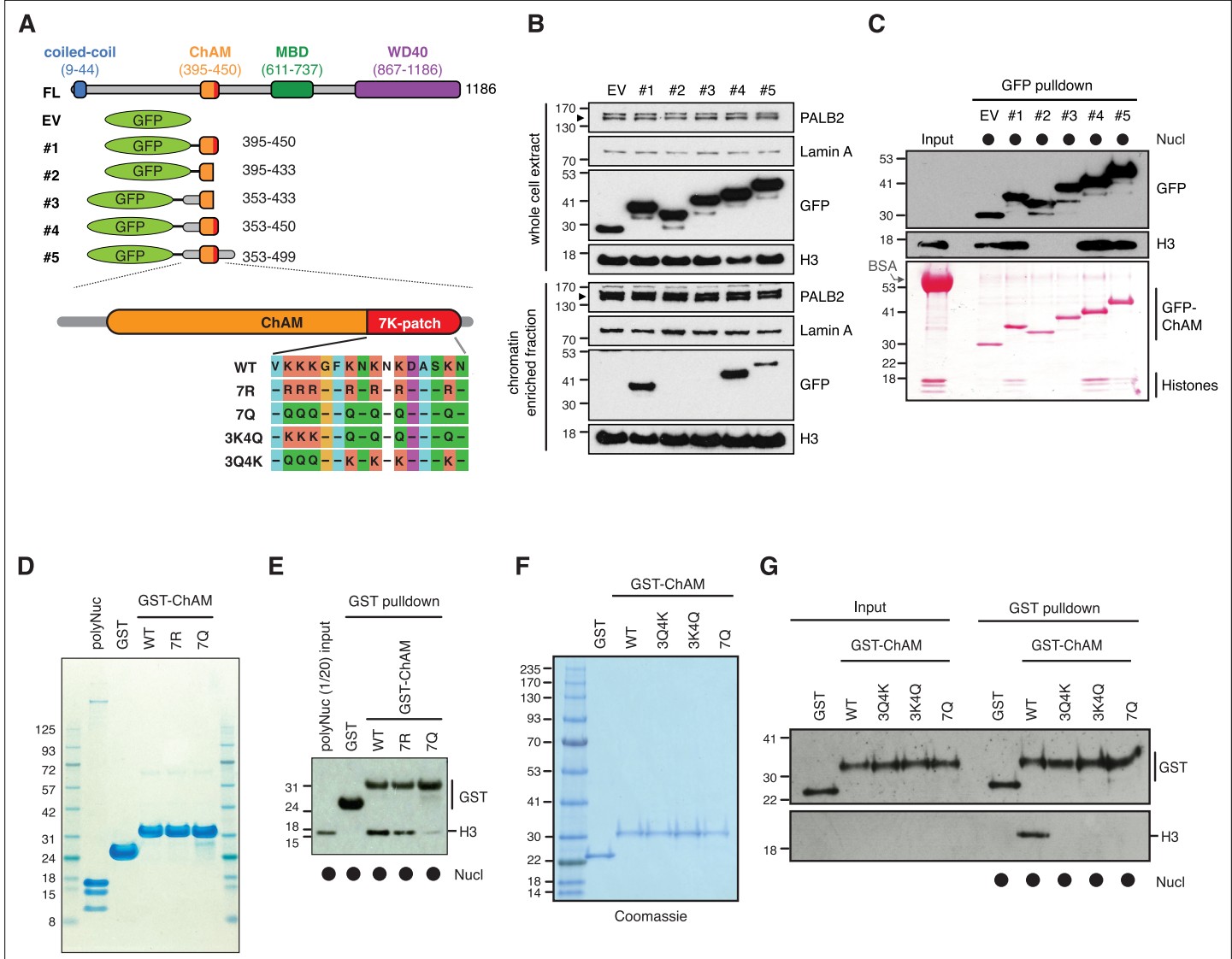

**Figure 2.** The 7K-patch promotes ChAM nucleosome association. (**A**) Diagram showing the full-length (FL) PALB2, GFP-ChAM fragments #1 to #5 and GST-ChAM variants used in this study. ChAM C-terminal lysine-rich patch (7K-patch, residues 436–449) is highlighted in red. (**B**) Western blot analysis assessing the chromatin enrichment of GFP-ChAM fragments transiently expressed in HEK293T cells. Empty GFP vector (EV) was used as a negative control. Whole-cell extract was also prepared to compare GFP-ChAM fragments expression levels. Lamin A and histone H3 were used as controls for the cellular fractionation. (**C**) In vitro nucleosome binding assay using GFP and GFP-ChAM fragments. Partially purified human nucleosomes captured by GFP and GFP-ChAM fragments immobilised on beads were detected by anti-histone H3 western blot and Ponceau S staining. (**D**) Coomassie blue staining of GST and GST-ChAM 7R and 7Q variants purified from *E. coli* and separated by SDS-PAGE. (**E**) In vitro binding assays using immobilised GST-ChAM 7R and 7Q variants and purified HeLa poly-nucleosomes. GST was used as a negative control. The nucleosome binding efficiency was assessed by anti-histone H3 western blot. (**F**) Coomassie blue staining of GST and GST-ChAM 3Q4K, 3K4Q and 7Q variants purified from E. coli and separated by SDS-PAGE. (**G**) In vitro binding assays using immobilised GST-ChAM 3Q4K, 3K4Q and 7Q variants and purified HeLa mononucleosomes. GST was used as a negative control. The nucleosome binding efficiency was assessed by anti-histone H3 western blot.

The online version of this article includes the following source data and figure supplement(s) for figure 2:

**Source data 1.** Original images for *Figure 2B*.

**Source data 2.** Original images for *Figure 2C*.

**Source data 3.** Original images for *Figure 2D* (left) and *Figure 2E* (right).

**Source data 4.** Original images for *Figure 2F* (top) and *Figure 2G* (bottom).

**Figure supplement 1.** Chromatin-associated ChAM is highly acetylated in vivo.

**Figure supplement 1—source data 1.** Original images for *Figure 2—figure supplement 1B*.

**Figure supplement 2.** Structural modelling of PALB2.

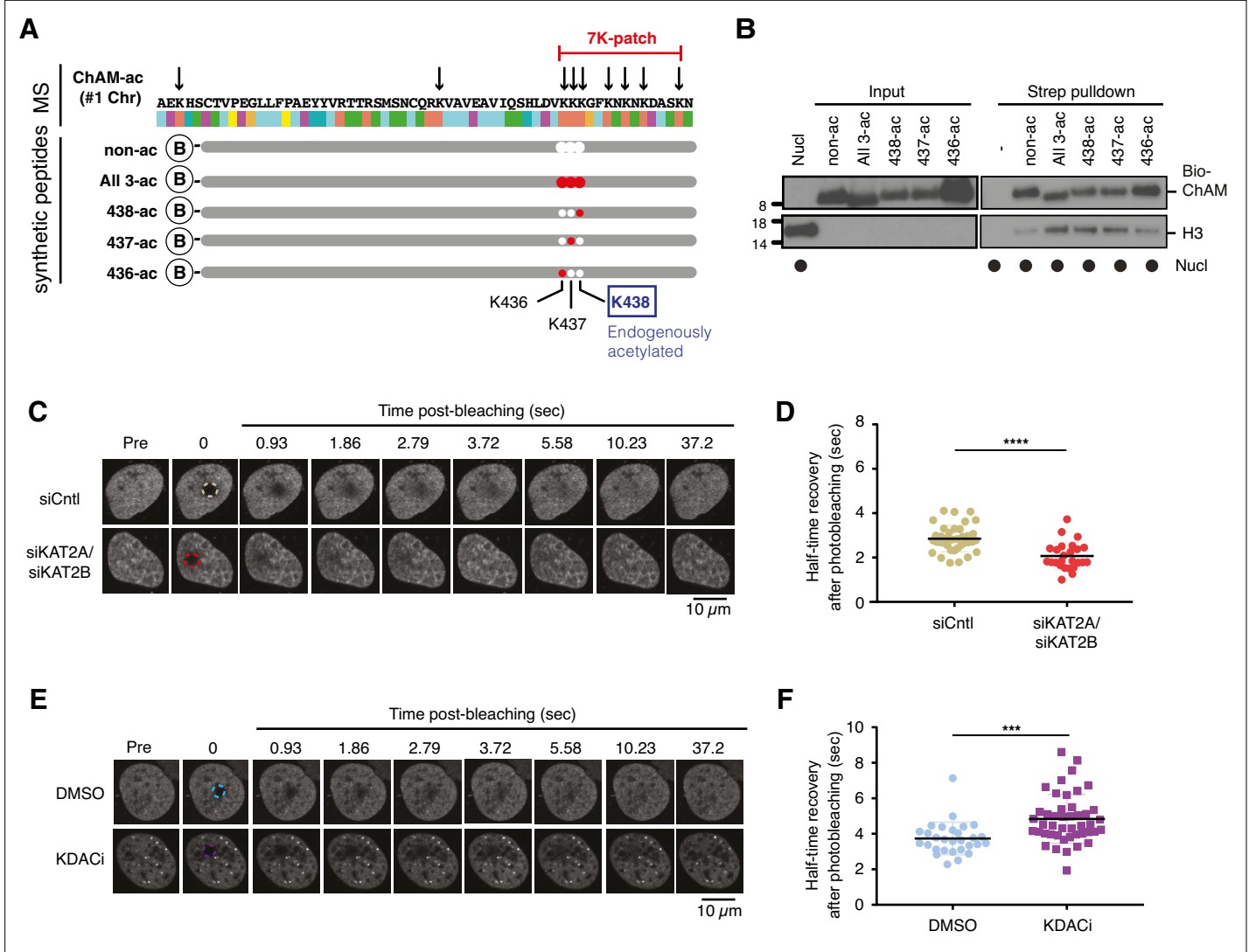

**Figure 3.** Acetylation within the 7K-patch enhances ChAM nucleosome association. (**A**) In vivo acetylated lysine residues in chromatin-associated ChAM, detected by tandem MS analysis, are highlighted by arrows in the upper part (MS). Synthetic biotin-ChAM peptides non-acetylated (non-ac) or acetylated at single lysine residues 436 (436-ac), 437 (437-ac), and 438 (438-ac), or all three lysine residues (All 3-ac) are shown in the lower part. The position of the 7K-patch is highlighted in red. The position of the K438 residue, the endogenous target of KAT2A/2B within the ChAM (shown in *Figure 1A*), is highlighted in a box. (**B**) In vitro nucleosome binding assays for the synthetic biotin-ChAM peptides. After incubation with purified HeLa polynucleosomes in the presence of 5 µg salmon sperm DNA, biotin-ChAM peptides were pulled-down using streptavidin beads (Strep pulldown), and associated nucleosomes were detected by anti-histone H3 western blot. (**C-D**) FRAP assay of FE-PALB2 expressed in U2OS-shPALB2 cells, treated either with siRNA targeting KAT2A and KAT2B (siKAT2A/siKAT2B) or with negative control siRNA (siCntl). Representative images of live cells before bleaching (pre-bleaching) and during a 37.5 s recovery period (post-bleaching) are shown in (*C*), where dashed circles indicate bleached areas. FRAP data are quantified and plotted in (*D*). Dots represent values of half-recovery time ($t_{1/2}$) for individual cells and bars mean values ± SD (siCntl, n=59; siKAT2A/2B, n=51). Statistical analyses were performed using GraphPad Prism 7.02 and p-values are for the unpaired Student's *t*-test (*** p<0.0006). (**E-F**) As in *C-D*, except cells were treated with a cocktail of lysine deacetylase inhibitors (KDACi, 5 mM sodium butyrate, 5 µM trichostatin A, and 5 mM nicotinamide) or DMSO as a control. Representative images of live cells before bleaching (pre-bleaching) and during a 37.5 s recovery period (post-bleaching) are shown in (**E**), where dashed circles indicate bleached areas. FRAP data are quantified and plotted in (**F**). Dots represent values of half-recovery time ($t_{1/2}$) for individual cells and bars mean values ± SD (DMSO, n=61; KDAC, n=94). Statistical analyses were performed using GraphPad Prism 7.02 and *p*-values are for the unpaired Student's *t*-test (**** p<0.0001). Representative real-time cell images are shown in *Figure 3—video 1* (siCntl), *Figure 3—video 1* (siKAT2A/siKAT2B), *Figure 3—video 1* (DMSO) and *Figure 3—video 4* (KDACi).

The online version of this article includes the following video, source data, and figure supplement(s) for figure 3:

**Source data 1.** Original images for *Figure 3B*.

**Source data 2.** Numerical data for *Figure 3D* and *Figure 3F*.

*Figure 3 continued on next page*

*Figure 3 continued*

**Figure supplement 1.** PALB2 mobility increases upon deacetylation.

**Figure supplement 1—source data 1.** Original images for *Figure 3—figure supplement 1B* (left) and *Figure 3—figure supplement 1D* (right).

**Figure supplement 1—source data 2.** Numerical data for *Figure 3—figure supplement 1C and E*.

**Figure 3—video 1.** Representative FRAP image of FE-PALB2 expressed in U2OS cells treated with control siRNA (siCntl), related to Figure 3C and D.
https://elifesciences.org/articles/57736/figures#fig3video1

**Figure 3—video 2.** Representative FRAP image of FE-PALB2 expressed in U2OS cell treated with siRNA targeting KAT2A and KAT2B (siKAT2A/siKAT2B), related to Figure 3C and D.
https://elifesciences.org/articles/57736/figures#fig3video2

**Figure 3—video 3.** Representative FRAP image of FE-PALB2 expressed in U2OS cell treated with DMSO, related to Figure 3E and F.
https://elifesciences.org/articles/57736/figures#fig3video3

**Figure 3—video 4.** Representative FRAP image of FE-PALB2 expressed in U2OS cell treated with a cocktail of lysine deacetylase inhibitors (KDACi), related to Figure 3E and F.
https://elifesciences.org/articles/57736/figures#fig3video4

inhibition of lysine deacetylases (KDAC), which increased the global lysine acetylation levels, decreased the diffusion rate of FE-PALB2 (increased FRAP $t_{1/2}$; *Figure 3E and F*, *Figure 3—figure supplement 1D and E*, *Figure 3—videos 3 and 4*). It is noteworthy that hyper-acetylation of histones promotes chromatin relaxation, which can increase the local mobility of chromatin. Hence, the reduced PALB2 mobility in cells treated with KDAC inhibitors suggests that PALB2 chromatin association is stimulated by lysine acetylation.

## DNA damage triggers ChAM deacetylation and increases PALB2 mobility

Our observations above indicate that PALB2 acetylation promotes its association with nucleosomes/chromatin, thereby restricting its mobility (*Figure 4A*). However, static PALB2 chromatin association could be toxic to cells when DSBs at random locations need to be recognised by the repair complex. Hence, we hypothesised that ChAM acetylation might be dynamically controlled upon stochastic DNA damage, such that PALB2 can be mobilised. Indeed, in HEK293 cells exposed to ionizing radiation (IR), rapid deacetylation of the ChAM fragment was detected at 15 min and persisted for at least 2 hr (*Figure 4B and C*). Furthermore, FRAP analyses of full-length FE-PALB2 in U2OS cells revealed a clear increase in the diffusion rate following IR treatment, decreasing FRAP $t_{1/2}$ by 22% compared to control conditions (*Figure 4D and E*, *Figure 4—figure supplement 1A-C*, *Figure 4—videos 1 and 2*). In both HEK293 and U2OS cells, IR-induced DNA damage response was confirmed by γ-H2A.X staining (*Figure 4B*, *Figure 4—figure supplement 1B*). Since PALB2 mobility is suppressed when lysine acetylation is up-regulated (*Figure 3E and F*), it is unlikely that the damage-induced increase of PALB2 mobility is associated with chromatin relaxation caused by histone acetylation, an event which is observed upon DNA damage (*Ziv et al., 2006*). Together, these results support the notion that DNA damage triggers ChAM deacetylation and PALB2 mobilisation.

## PALB2 ChAM 7K-patch acetylation defines its context-dependent association with actively transcribed genes

Exposure to IR stochastically induces various DNA lesions, including DSBs, single-strand breaks (SSBs), base damage and DNA-protein cross-links, at random genomic locations. In S phase, these lesions perturb DNA replication and, in some circumstances, could be converted to DSBs. Given that IR triggers ChAM deacetylation and PALB2 mobilisation, we considered that PALB2 ChAM deacetylation might promote translocation of PALB2 from its intrinsically associated actively transcribed loci to randomly distributed sites of DSBs to promote HR repair.

To address this idea, we made use of full-length PALB2 variants in which lysine residues within the 7K-patch were substituted with Q (7K-patch null mutant) or R (non-acetylatable positively charged 7K-patch). These PALB2 variants, named 7Q or 7R, as well as PALB2 WT or an empty vector (EV), were stably expressed as FLAG-fusions at equivalent levels in U2OS-shPALB2 cells (*Figure 5A*, *Figure 5—figure supplement 1A*). As expected, our subcellular fractionation analyses showed a reduced level

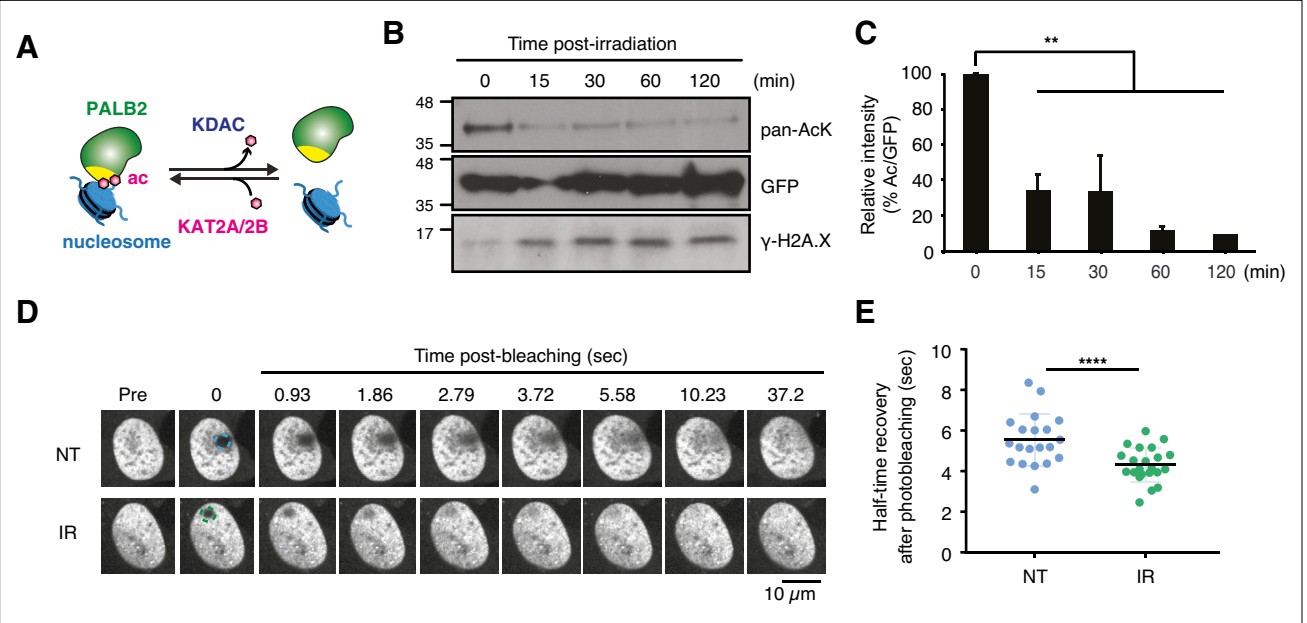

**Figure 4.** DNA damage triggers ChAM deacetylation and PALB2 mobilization. (**A**) Depiction of PALB2 acetylation and nucleosome binding, controlled by KAT2A/2B and KDACs. (**B**) HEK293T cells transiently expressing GFP-ChAM were treated with 4 Gy IR, and affinity-purified GFP-ChAM acetylation was assessed using anti-acetyl-lysine (pan-AcK) western blot. γ-H2A.X signal in whole cell lysate was detected to monitor DNA damage. (**C**) Histogram showing the relative level of ChAM acetylation following irradiation. The levels of acetylated and total GFP-ChAM were quantified by ImageJ, and the ratio of acetyl/total GFP-ChAM was normalised against that of the pre-irradiation sample (time 0). Error bars indicate SD from two independent experiments (n=2). Statistical analyses were performed using GraphPad Prism 9.4.1 and p-value is the unpaired one-way ANOVA (**p<0.01). (**D-E**) FRAP assay of FE-PALB2 expressed in U2OS-shPALB2 cells, either non-treated (NT) or treated with 4 Gy IR. Representative images of live cells before bleaching (pre-bleaching) and during a 37.5 s recovery period (post-bleaching) are shown in (*D*), where dashed circles indicate bleached areas. FRAP data are quantified and plotted in (**E**). Dots represent values of half-recovery time ($t_{1/2}$) for individual cells and bars mean values ± SD (NT, n=20; IR, n=21). Statistical analyses were performed using GraphPad Prism 7.02 and p-values are for the unpaired Student's *t*-test (**** p<0.0001). Representative real-time cell images are shown in *Figure 4—video 1* (NT) and *Figure 4—video 2* (IR).

The online version of this article includes the following video, source data, and figure supplement(s) for figure 4:

**Source data 1.** Original images for *Figure 4B*.

**Source data 2.** Numerical data for *Figure 4C and E*.

**Figure supplement 1.** PALB2 mobility increases upon genotoxic stress.

**Figure supplement 1—source data 1.** Numerical data for *Figure 4—figure supplement 1C*.

**Figure 4—video 1.** Representative FRAP image of FE-PALB2 expressed in U2OS cell without treatment (NT), related to Figure 4D and E. https://elifesciences.org/articles/57736/figures#fig4video1

**Figure 4—video 2.** Representative FRAP image of FE-PALB2 expressed in U2OS cell following IR treatment, related to Figure 4D and E. https://elifesciences.org/articles/57736/figures#fig4video2

of PALB2 7Q in the chromatin-enriched fraction compared to PALB2 WT or 7R (*Figure 5B and C*), agreeing with the direct nucleosome binding properties of the ChAM variants (*Figure 2D and E*). Similarly, our FRAP analyses of the PALB2 variants, conditionally expressed as FE-fusions in U2OS-shPALB2 cells, showed an increase in PALB2 7Q diffusion kinetics compared to those of PALB2 WT and 7R (*Figure 5D and E*, *Figure 5—videos 1–3*). A PALB2 variant with an internal deletion of ChAM, which abolishes its chromatin association (*Bleuyard et al., 2012*), equally exhibited a significant increase in PALB2 diffusion rate (*Figure 5—figure supplement 1B-D*, *Figure 5—videos 4; 5*). Collectively, we concluded that PALB2 7Q is constitutively highly mobile and is unable to engage with chromatin.

## PALB2 7K-patch is important for damage-induced mobility change

We next turned to investigate the impact of the mutations of the 7K-patch on PALB2 recruitment to randomly distributed DNA damage sites. Here, we exposed FE-PALB2-expressing U2OS-shPALB2 cells to doxycycline (Dox) for 4 days (*Figure 6A and B*) for fuller depletion of endogenous PALB2,

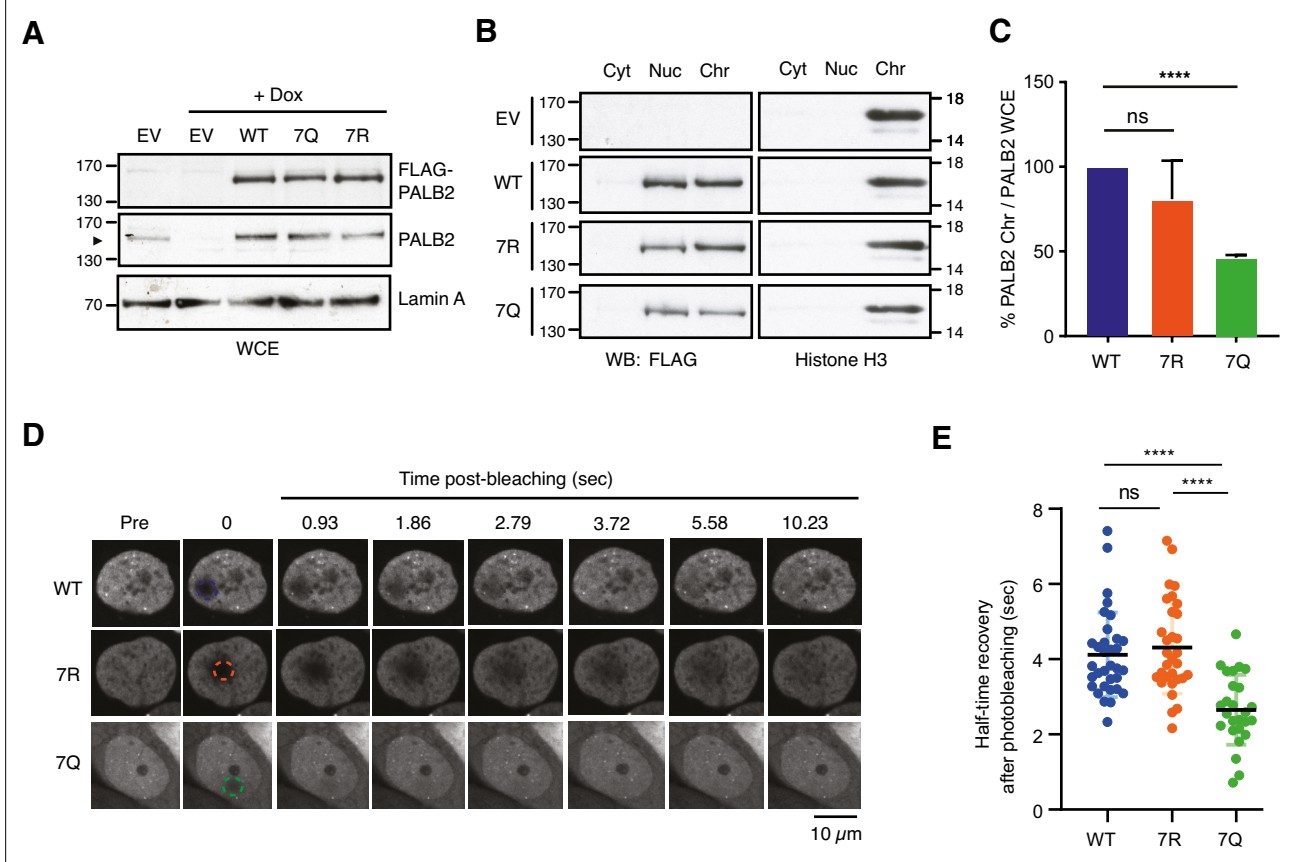

**Figure 5.** ChAM 7K-patch mediates PALB2 global chromatin association. (**A**) Western blot analysis of U2OS-shPALB2 cells constitutively expressing FLAG (EV) or FLAG-PALB2 variants. Where indicated, cells were treated with 2 µg/mL doxycycline (+Dox) to deplete endogenous PALB2. Lamin A was used as a loading control. (**B-C**) Subcellular distribution of FLAG-PALB2 variants in cytoplasmic (Cyt), nuclear soluble (Nuc) and chromatin-enriched (Chr) fractions upon doxycycline-induced PALB2 depletion. Histone H3 was used as a control for cellular fractionation. Protein levels, detected by western blotting, were quantified using ImageJ. PALB2 levels in the chromatin fraction were normalised against PALB2 levels in whole-cell extracts shown in (**A**) and expressed as a percentage of the WT in (**C**). Data indicate mean values ± SD from three independent experiments (n=3). Statistical analyses were performed using GraphPad Prism 7.02, and p-values are from the unpaired Student's *t*-test for pairwise comparisons (ns, not significant; **** $p < 0.0001$). (**D-E**) FRAP analysis of FE-PALB2 WT, 7R and 7Q in U2OS-shPALB2 cells in which endogenous PALB2 was depleted *via* doxycycline exposure. Representative images before bleaching (pre-bleaching) and during the recovery period (post-bleaching) are shown in (**D**), where dashed circles indicate bleached areas. FRAP data are quantified and plotted in (**E**). Dots represent values of half-recovery time ($t_{1/2}$) for individual cells and bars mean values ± SD (WT, n=33; 7R, n=31; 7Q, n=26). Statistical analyses were performed using GraphPad Prism 7.02 and p-values are from the unpaired Student's *t*-test for pairwise comparisons (ns, not significant; **** $p < 0.0001$). Representative real-time cell images are shown in *Figure 5—video 1* (WT), *Figure 5—video 2* (7R) and *Figure 5—video 3* (7Q).

The online version of this article includes the following video, source data, and figure supplement(s) for figure 5:

**Source data 1.** Original images for *Figure 5A*.

**Source data 2.** Original images for *Figure 5B*.

**Source data 3.** Numerical data for *Figure 5C and E*.

**Figure supplement 1.** PALB2 ChAM limits its mobility.

**Figure supplement 1—source data 1.** Numerical data for *Figure 5—figure supplement 1C*.

**Figure 5—video 1.** Representative FRAP image of FE-PALB2 WT expressed in U2OS cell, related to Figure 5D and E.
https://elifesciences.org/articles/57736/figures#fig5video1

**Figure 5—video 2.** Representative FRAP image of FE-PALB2 7R variant expressed in U2OS cell, related to Figure 5D and E.
https://elifesciences.org/articles/57736/figures#fig5video2

**Figure 5—video 3.** Representative FRAP image of FE-PALB2 7Q variant expressed in U2OS cell, related to Figure 5D and E.
https://elifesciences.org/articles/57736/figures#fig5video3

**Figure 5—video 4.** Representative FRAP image of FE-PALB2 WT expressed in U2OS cell, related to Figure 5—figure supplement 1B and C.

*Figure 5 continued on next page*

*Figure 5 continued*

https://elifesciences.org/articles/57736/figures#fig5video4

**Figure 5—video 5.** Representative FRAP image of FE-PALB2 with an internal deletion of the ChAM (ΔChAM) expressed in U2OS cell, related to Figure 5—figure supplement 1B and C.

https://elifesciences.org/articles/57736/figures#fig5video5

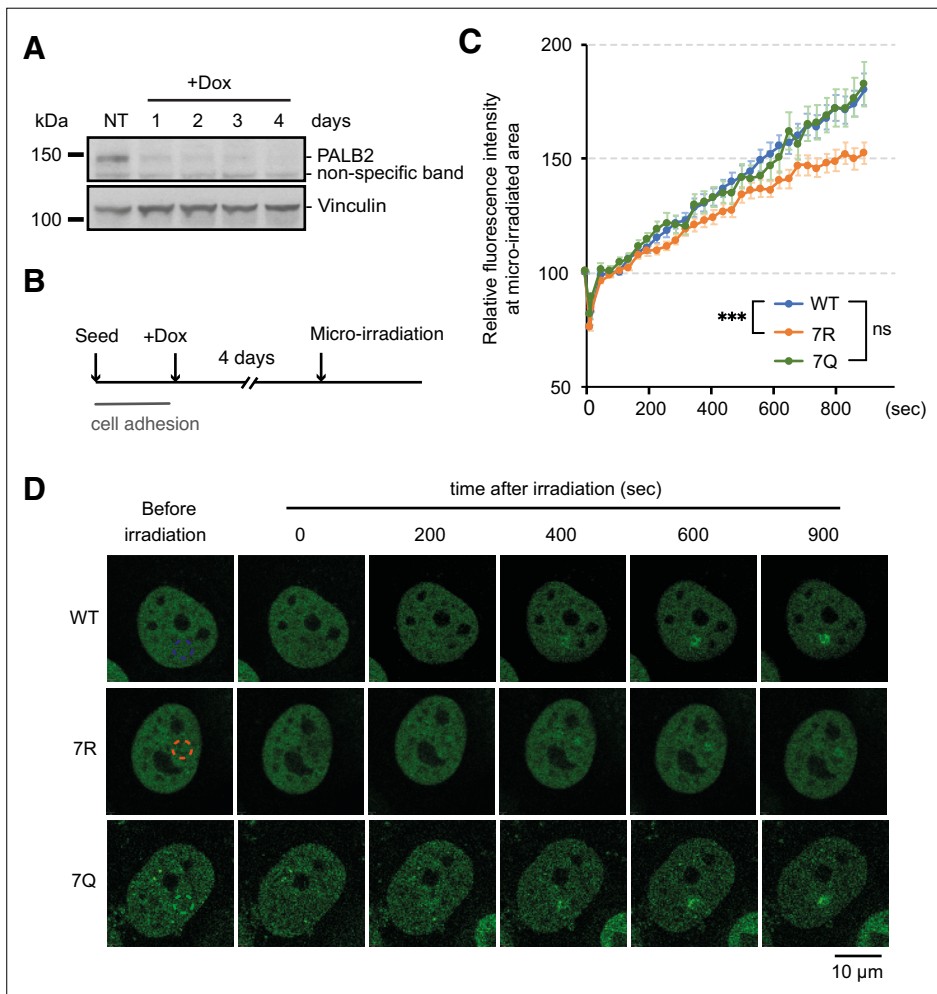

**Figure 6.** ChAM 7K-patch affect PALB2 localisation upon micro-irradiation. (**A**) Depletion of endogenous PALB2 in the U2OS-shPALB2 cell line following the exposure to doxycycline over 4 days. Vinculin was used as a loading control. (**B**) Diagrams of experimental procedures, assessing the recruitment of PALB2 upon micro-irradiation following depletion of endogenous PALB2 with 4 days of doxycycline exposure. (**C**) Time plots of fluorescence intensity of FE-PALB2 at laser damage sites relative to an unirradiated area. Data represent the mean of relative fluorescence intensity ± SEM from four independent experiments (a total of at least n=100 cells per condition). Statistical analysis was done for the last time-point using GraphPad Prism 8 and p-values are for the Mann–Whitney test (ns, not significant; ***p<0.001). (**D**) Representative images of FE-PALB2 variants conditionally expressed in U2OS-shPALB2 cells upon micro-irradiation. Dashed circles indicate irradiated areas.

The online version of this article includes the following source data and figure supplement(s) for figure 6:

**Source data 1.** Original images for *Figure 6A*.

**Source data 2.** Numerical data for *Figure 6B*.

**Figure supplement 1.** PALB2 7K-patch does not affect its interaction with canonical binding partners.

**Figure supplement 1—source data 1.** Original images for *Figure 6—figure supplement 1*, FLAG-IP.

**Figure supplement 1—source data 2.** Original images for *Figure 6—figure supplement 1*, input.

which might otherwise mask the phenotypes of the 7Q or 7R mutants by forming heterodimers (*Buisson and Masson, 2012*). Localised DNA damage was subsequently induced by laser micro-irradiation and the recruitment of FE-PALB2 variants was assessed by live cell imaging. This analysis revealed that the 7R substitution, but not the 7Q counterpart, impeded efficient recruitment of PALB2 to the micro-irradiated chromatin areas (*Figure 6C and D*). These impacts are unlikely to be associated with defective PALB2 protein complex formation, as all PALB2 variants maintained equivalent levels of interaction with BRCA1, BRCA2 and RAD51 in cells treated with IR (*Figure 6—figure supplement 1*). Rather, this observation is best explained if constitutively non-acetylated ChAM 7K-patch, as mimicked by the 7R substitutions, elicits sparse and non-specific chromatin association, preventing active displacement of PALB2 from these regions upon DNA damage. Hence, we consider that PALB2 7R is a variant which randomly binds chromatin with no locus specificity and is less efficiently mobilised upon DNA damage.

## PALB2 7K-patch is important for efficient recruitment of RAD51 at randomly distributed DNA damage sites

Having established the constitutively highly mobile (7Q) or weakly mobile (7R) PALB2 variants, we further evaluated the impact of PALB2 mobility on the proficiency of HR by assessing the formation of RAD51 foci in S phase cells upon IR exposure. For these experiments, cells were exposed to doxycycline for 4 or 5 days (*Figure 7A*) to further ensure the full depletion of endogenous PALB2. Cells depleted of endogenous PALB2 for 4 or 5 days readily exhibited impaired IR-induced RAD51 foci formation in S phase, marked by the incorporation of thymidine analogue EdU. This phenotype was efficiently rescued by PALB2 WT complementation, but only partially by PALB2 7R or 7Q (*Figure 7B*). Additionally, while we observed no significant increase of γ–H2A.X in these cells at 4 days after doxycycline exposure (*Figure 7C*), more noticeable increases of γ–H2A.X were detected in PALB2 7Q or 7R cells exposed to doxycycline for 5 days compared to WT cells (*Figure 7D–F*). Given that RAD51 foci and γ–H2A.X foci formation closely reflect HR proficiency and DNA damage, respectively, this finding suggested that both the 7Q and 7R substitutions confer HR defects and accumulation of DNA damage in the longer term.

## The PALB2 7K-patch is important for normal cell growth

While conducting the above experiment, we noticed that, similarly to IR treated conditions, PALB2-depleted, but otherwise untreated, cells expressing 7Q or 7R exhibited impaired spontaneous RAD51 foci and an increase of γ–H2A.X in S phase (*Figure 8A–F*). The physiological importance of this phenomenon is highlighted by the reduced survival of PALB2-depleted cells complemented with 7Q PALB2 and, more significantly, with 7R PALB2, compared to those expressing WT PALB2 in untreated cells (*Figure 9A and B*).

Intrigued by these observations, we sought the underlying molecular mechanism. It is widely described that transcriptionally active loci are vulnerable to DNA damage during DNA replication, particularly when challenged by genotoxic reagents that stabilise RNA:DNA hybrids, such as camptothecin (CPT) (*Marinello et al., 2013*). In undamaged cells, WT PALB2 is enriched at a small subset of, but not all, H3K36me[3] marked active genes, which in turn prevents CPT-induced DNA damage at these loci (*Bleuyard et al., 2017a*). We therefore asked whether the PALB2 7K-patch affects its occupancy at these previously identified PALB2-bound loci. Indeed, our ChIP-qPCR analyses revealed significant impairments of PALB2 7R and 7Q occupancies at these loci, namely the *ACTB*, *TCOF1* and *WEE1* genes (*Figure 9C and D*). These observations are in line with the view that the 7Q substitutions impede overall chromatin association, while 7R substitutions elicit sparse and non-specific chromatin association, limiting PALB2 enrichment at these loci. It is important to note, however, that reduced PALB2 association at these loci per se does not confer DNA damage in cis unless cells are challenged by CPT (*Bleuyard et al., 2017a*). Hence, this did not fully explain the increase of γ–H2A.X in cells expressing the 7Q or 7R variant in unperturbed conditions as shown in *Figure 8E*.

We postulated that, even in unperturbed cells, spontaneous DNA lesions may be induced at any locations of genomic DNA due to naturally occurring genotoxic metabolites, such as reactive oxygen species, and be processed to DSBs when encountered by the DNA replication machinery. In this scenario, the mobility of PALB2 would be important for the repair of these randomly distributed DSBs, as it is following exposure to IR. Indeed, our ChIP-qPCR assessment of PALB2 occupancy at

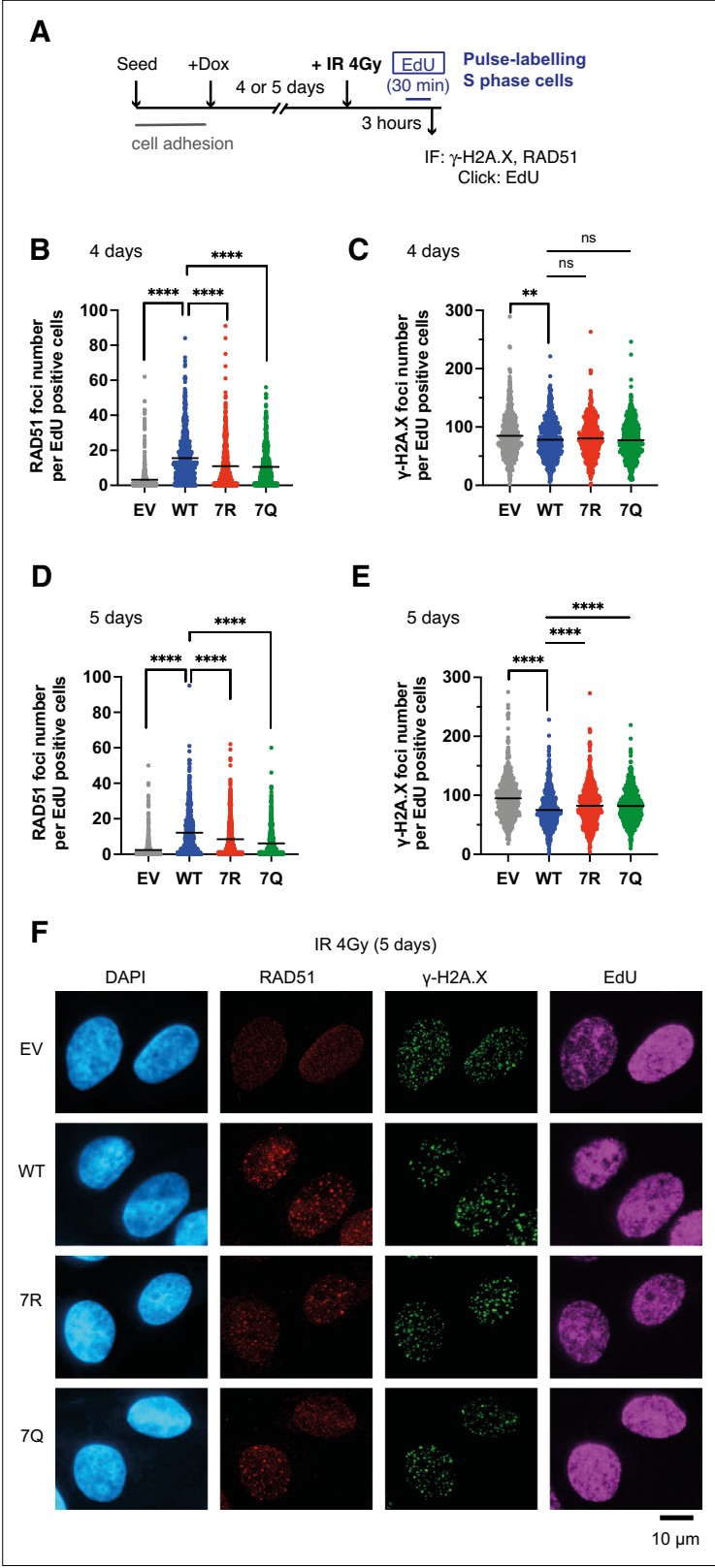

**Figure 7.** 7K-patch-mediated PALB2 chromatin association affects IR-induced RAD51 foci formation. (**A**) Diagrams of experimental procedures, assessing RAD51 in S phase cells, marked by EdU pulse labelling. (**B-C**) Quantification of the number of RAD51 foci (**B**) or γ-H2A.X foci (**C**) per cell, upon doxycycline exposure for 4 days and subsequently exposed to 4 Gy IR. Dots represent values for individual cells and bars mean values from

*Figure 7 continued on next page*

*Figure 7 continued*

three independent experiments (a total of n=600 cells per condition). Statistical analyses were performed using GraphPad Prism 8 and p-values are for the Mann–Whitney test (ns, not significant; **p<0.01; ****p<0.0001). (**D-E**) Quantification of the number of RAD51 foci (**D**) or γ-H2A.X foci (**E**) per cell, upon doxycycline exposure for 5 days. Dots represent values for individual cells and bars mean values from three independent experiments (a total of n=600 cells per condition). Statistical analyses were performed using GraphPad Prism 8 and p-values are for the Mann–Whitney test (****p<0.0001). (**F**) Representative images of RAD51 and γ-H2A.X foci in U2OS-shPALB2 cells stably expressing FLAG (EV) and FLAG-PALB2 variants, following doxycycline-induced endogenous PALB2 depletion for 5 days and subsequently exposed to 4 Gy IR. DAPI was used for nuclear staining.

The online version of this article includes the following source data for figure 7:

**Source data 1.** Numerical data for *Figure 7C, D, E and F*.

the known PALB2-bound loci revealed that the occupancy of PALB2 WT at active genes was dramatically reduced upon IR exposure to levels equivalent to those of the untreated 7Q or 7R mutant (*Figure 9D*). Together, these observations indicate that the 7Q or 7R variants are defective for HR repair of randomly induced DNA damage outside of active genes.

## Discussion

In this study, we have demonstrated that reversible lysine acetylation controls the mode of PALB2 chromatin association, fine-tuning RAD51 recruitment during S phase. Taking all our findings collectively, we propose the following model (illustrated in *Figure 9E*): (1) KAT2A/2B-mediated ChAM acetylation, jointly with MRG15, governs PALB2 enrichment at undamaged, transcriptionally active chromatin; (2) global DNA damage signalling triggers ChAM deacetylation, which in turn releases PALB2 from active genes; (3) in this way, PALB2, in complex with BRCA1, BRCA2 and RAD51, is effectively recruited to sites of damaged chromatin; and (4) deacetylated ChAM binding to exposed damaged chromatin allows appropriate engagement of this complex with damaged DNA and hence promotes RAD51 loading and HR repair.

These findings further our understanding of how the ChAM domain interacts with nucleosomes and how this association is dynamically regulated. It was previously demonstrated that the evolutionarily highly conserved N-terminal part of the ChAM affects its association with nucleosomes (*Bleuyard et al., 2017a*; *Bleuyard et al., 2017b*), while an interface composed of basic residues across the ChAM binds to an acidic surface on histone H2A in the nucleosome core particle (*Belotserkovskaya et al., 2020*). Our results presented here introduce an additional complexity, showing that the C-terminal 7K basic patch is essential for context-dependent ChAM interaction with nucleosomes. Specifically, ChAM binding to nucleosomes at active genes is promoted by 7K patch acetylation, while its dynamic recruitment to damaged chromatin is stimulated by 7K patch deacetylation (*Figures 3 and 9C and D*). Considering negative charge of nucleosomal DNA, it is not surprising that lysine acetylation, which neutralises its positive charge, might impact ChAM interaction with nucleosomes. However, our results have also revealed that the acetylation of the 7K patch provides a non-electrostatic and as-yet incompletely understood mechanism to enhance its association with nucleosomes. Intriguingly, the flanking regions of the ChAM also appear to influence ChAM binding to nucleosomes (*Figure 2A, B and C*): an N-terminal extension enhanced ChAM interaction with nucleosomes, while a C-terminal extension had a negative impact. These regions are rich in serine and threonine residues, including potential targets for cell cycle regulators (CDKs and PLK1) and DNA damage-responsive kinases (ATM and ATR) (*Figure 9—figure supplement 1*). These observations suggest that additional PTMs other than lysine acetylation might be involved in regulating ChAM association with nucleosomes. These would undoubtedly merit further investigation.

The increase in PALB2 mobility during the DDR shown in this study (*Figure 4D and E*) is reminiscent of previous reports demonstrating increased RAD51 mobility upon replication stress (*Srivastava et al., 2009*; *Yu et al., 2003*). Yu et al. showed that a fraction of RAD51, which is in complex with BRCA2, is immobile in unchallenged cells, but becomes mobile upon hydroxyurea-induced replicative stress (*Yu et al., 2003*). Notably, their data suggested that such increased RAD51 mobility was unlikely to be a consequence of altered interaction with BRCA2 upon replicative stress. Similarly, Jeyasekharan et al. reported increased BRCA2 mobility upon IR-induced DNA damage, which was dependent on

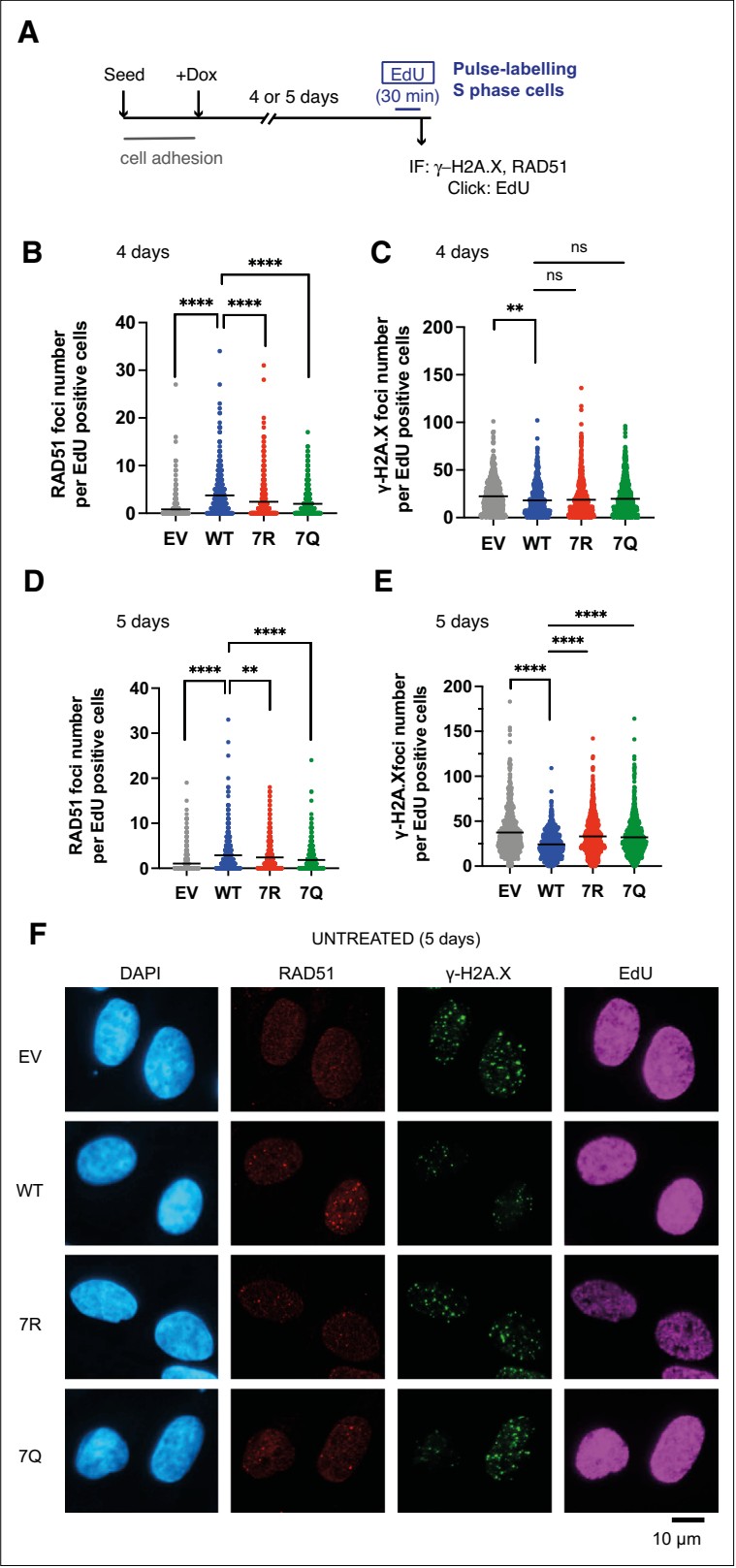

**Figure 8.** 7K-patch-mediated PALB2 chromatin association affects spontaneous RAD51 foci formation in normally growing cells. (**A**) Diagrams of experimental procedures, assessing RAD51 in S phase cells, marked by EdU pulse labelling. (**B-C**) Quantification of the number of RAD51 foci (**B**) or γ-H2A.X foci (**C**) per cell, upon doxycycline exposure for 4 days. Dots represent values for individual cells and bars mean values from three independent

*Figure 8 continued on next page*

*Figure 8 continued*

experiments (a total of n=600 cells per condition). Statistical analyses were performed using GraphPad Prism 8 and p-values are for the Mann–Whitney test (ns, not significant; **p<0.01; ****p<0.0001) (**D-E**) Quantification of the number of RAD51 foci (**E**) or γ-H2A.X foci (**F**) per cell, upon doxycycline exposure for 5 days. Dots represent values for individual cells and bars mean values from three independent experiments (a total of n=600 cells per condition). Statistical analyses were performed using GraphPad Prism 8 and p-values are for the Mann–Whitney test (****P<0.0001). (**F**) Representative images of RAD51 and γ-H2A.X foci in U2OS-shPALB2 cells stably expressing FLAG (EV) and FLAG-PALB2 variants, following doxycycline-induced endogenous PALB2 depletion for 5 days. DAPI was used for nuclear staining.

The online version of this article includes the following source data for figure 8:

**Source data 1.** Numerical data for *Figure 8C, D, E and F.*

the DNA damage signalling kinases ATM and ATR and coincided with active HR events (*Jeyasekharan et al., 2010*). Given that BRCA2 and PALB2 form a stable complex together with a fraction of RAD51 and that their interaction is important for HR repair, we suggest that the BRCA2-PALB2 complex and associated RAD51 are mobilised together upon genotoxic stress.

The mechanisms finely regulating the mobilisation and localisation of HR factors are not fully elucidated but undoubtedly involve PTMs. We propose that dynamic modes of PTM-mediated chromatin association synergistically govern the recruitment of repair factors to defined regions of chromatin. In undamaged cells, PALB2 is enriched at actively transcribed genes through KAT2A/2B-mediated ChAM acetylation and H3K36me$^3$-mediated MRG15 interaction, while maintaining transient chromatin interaction. This transient mode of association may allow a severance mechanism, in which PALB2 continuously monitors the presence of DNA damage across these regions. Indeed, cells expressing PALB2 7R, which is capable of bulk chromatin association (*Figure 5*) but fails to bind specific active genes (*Figure 9D*), exhibited an increase in the basal level of γ–H2A.X compared to those expressing PALB2 WT (*Figure 8F*) and reduced survival (*Figure 9B*), underlining the importance of PALB2 acetylation in unperturbed cells. Upon DNA damage, however, ChAM is deacetylated (*Figure 4B and C*) and, as a consequence, PALB2 affinity for undamaged chromatin is reduced (*Figure 9D*). We suggest that damage-induced ChAM deacetylation allows its recruitment to sites of DNA damage through its interaction with damage-sensing factors, fulfilling its function in promoting HR repair.

Markedly, PALB2 occupancy at active genes, as detected by ChIP-qPCR (*Figure 9D*), was effectively reduced to null following IR exposure (i.e. comparable to the level in cells complemented with empty vector), although it may represent only the sub-fraction of PALB2 bound to the loci tested. Regardless, these observations raise a fundamental question: why does PALB2 dissociate from active genes with the potential risk of leaving these regions unprotected? Notably, the abundance of endogenous PALB2 protein is estimated to be considerably lower than that of BRCA2 or RAD51, for example ~60 times less than BRCA2 and ~600 times less than RAD51 in HeLa cells (*Kulak et al., 2014*). The majority of PALB2 is also found to be associated with BRCA2 on nuclear structures (*Xia et al., 2006*) and with MRG15 (*Bleuyard et al., 2017b*). Hence, we envision that, in endogenous conditions, PALB2 primarily associates with BRCA2 and RAD51 on actively transcribed loci to protect these regions (though PALB2-free pools of BRCA2 and RAD51 presumably exist), but when DNA damage occurs elsewhere, the same complex needs to be mobilised to promote HR repair. In light of these considerations, our experimental platform assessing cells expressing exogenous PALB2 variants, at higher than endogenous levels, might have somewhat compromised sensitivity to highlight the physiological impact of the PALB2 mobilisation. Either way, we envision that the under-enrichment of PALB2 at active genes is unlikely to confer severe consequences for the transcription-replication conflict, as IR would trigger the global DNA damage checkpoint, slowing replication fork progression (*Lajitha et al., 1958*; *Ord and Stocken, 1958*; *Watanabe, 1974*) while broken DNA is repaired.

Intriguingly, while PALB2 7K variants, with either Q or R substitutions, equally interact with BRCA1 in both undamaged and IR-treated conditions (*Figure 6—figure supplement 1*), their recruitment to sites of DNA damage appeared more affected with the 7R mutant (*Figure 6*). We postulate that the random mode of the 7R mutant chromatin association prevents its damage-induced mobilisation, which is normally triggered by ChAM deacetylation specifically occurring at transcriptionally active chromatin upon DNA damage and hence making PALB2 globally available for HR. In the meantime, PALB2 recruitment itself does not appear to guarantee RAD51 assembly. Both the 7R and 7Q

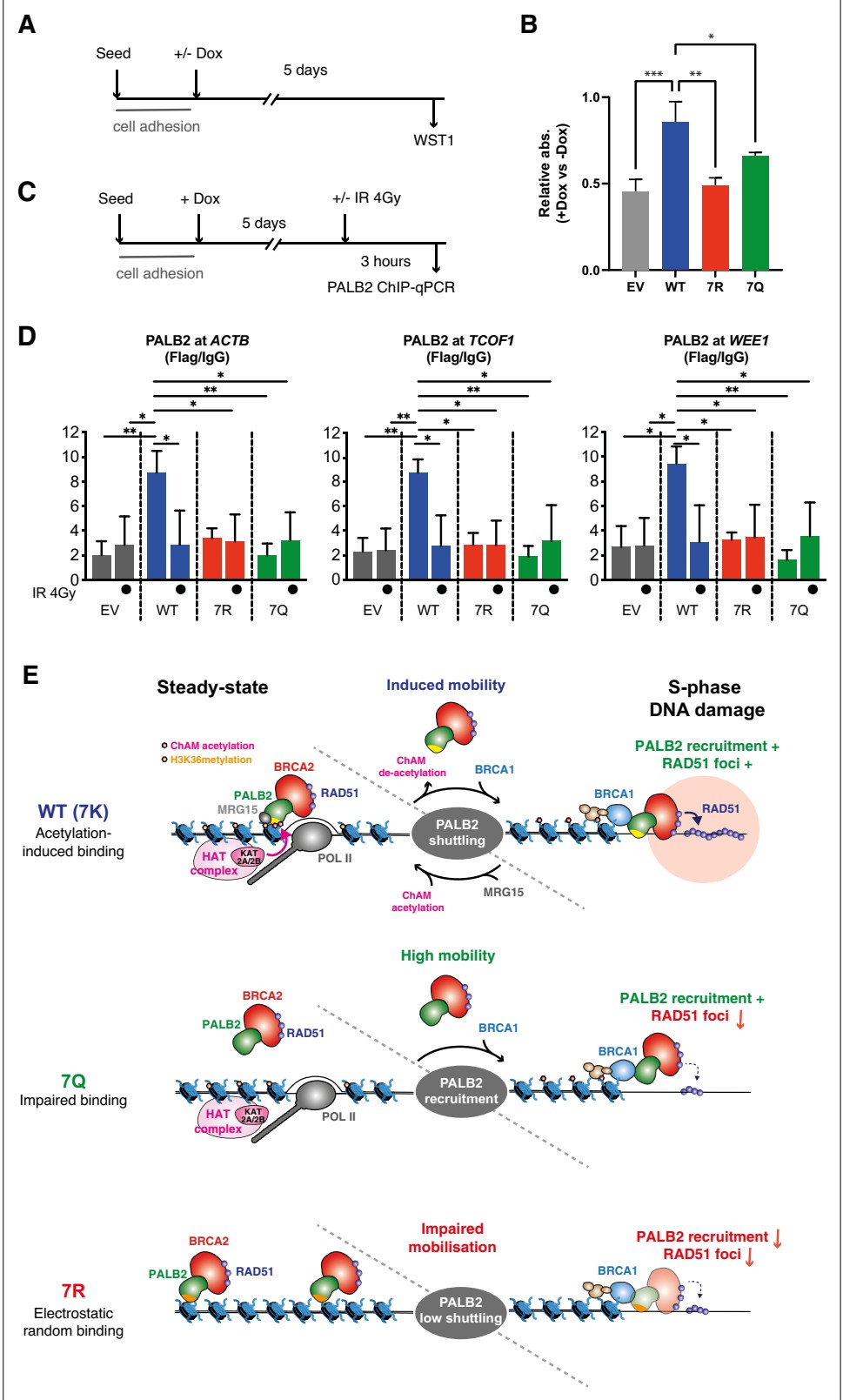

**Figure 9.** 7K-patch is important for normal cell survival. (**A**) Diagrams of experimental procedures, assessing survival of U2OS-shPALB2 stably expressing FLAG (EV) or FLAG-PALB2 variants upon doxycycline (Dox)-induced endogenous PALB2 depletion by WST-1 assay. (**B**) Cellular survival normalised against those without doxycycline exposure. Data represent mean values ± SD from three independent experiments (n=3). Statistical analyses

*Figure 9 continued on next page*

*Figure 9 continued*

were performed using GraphPad Prism 9 and p-values are the unpaired one-way ANOVA (*p<0.05; **p<0.01; ***p<0.001). (**C**) Diagrams of experimental procedures, assessing the FLAG-PALB2 fusion enrichment at previously reported PALB2-bound loci by ChIP-qPCR. (**D**) ChIP-qPCR analysis of FLAG-PALB2 WT, 7Q and 7R enrichment at known PALB2-bound loci (*Bleuyard et al., 2017a*), namely the coding regions of the *ACTB*, *TCOF1*, and *WEE1* genes, in cells untreated or treated with 4 Gy IR. Data are expressed as fold enrichment over IgG and represent mean values ± SD from three independent experiments (n=3) with triplicate qPCR reactions. Statistical analyses were performed using GraphPad Prism 8 and p-values are from two-way ANOVA with Tukey's multiple comparisons test (*p<0.05; **p<0.01). (**E**) Model for PALB2 acetylation function in the maintenance of genome stability. MRG15 and KAT2A/2B-mediated ChAM acetylation, which occurs locally at active genes, jointly promote PALB2 enrichment at undamaged transcriptionally active chromatin. DNA damage triggers ChAM deacetylation at these loci and temporarily releases PALB2 from chromatin. This allows PALB2 to interact with damage sensors, such as BRCA1, which in turn recruits the entire HR complex to sites of DNA damage. ChAM binding to naked DNA through the deacetylated 7K-patch may promote RAD51 loading and HR repair. The constitutively highly mobile PALB2 variant (7Q) is efficiently recruited to sites of DNA damage but, unlike WT PALB2, unable to promote efficient RAD51 foci formation. The randomly chromatin-bound and less capably mobilised PALB2 variant (7R) is inefficiently recruited to sites of DNA damage and hence fails to promote HR.

The online version of this article includes the following source data and figure supplement(s) for figure 9:

**Source data 1.** Numerical data for *Figure 9B*.

**Source data 2.** Numerical data for *Figure 9D*.

**Figure supplement 1.** Summary of ChAM properties and post-translational modifications.

substitutions elicited significant impairment of RAD51 foci formation (*Figures 7 and 8*), suggesting that the 7K-patch promotes optimal RAD51 engagement with broken DNA, conceivably in part through its direct interaction with the nucleosome acidic patch regions exposed at damaged chromatin (*Belotserkovskaya et al., 2020*). It should be noted, however, that the full impact of the 7R and 7Q mutations on HR repair remains unclear. We endeavoured to assess HR proficiency of these cell lines using our reporter systems, which monitor the HR-mediated integration of a promoterless GFP gene and the resultant expression of fluorescent protein as the readout (*Rodrigue et al., 2019*; *Yata et al., 2012*). Through these approaches, significant HR defects were not detected (data not shown). This could potentially be explained by the model that the PALB2 7K-patch is particularly important for HR repair of DSBs arising outside active genes, the detection of which is not straightforward using available techniques. Further investigation using more elaborate strategies, which allow the evaluation of HR repair outside active genes, is warranted to fully appreciate the importance of PALB2 ChAM acetylation in HR repair.

Overall, the results of this study highlight the importance of the dynamic regulation of PALB2, where both its acetylated and non-acetylated forms play critical roles in genome integrity control. Our work also suggests that caution should be exercised in the use of amino acid substitutions for functional studies of lysine acetylation. K to Q substitutions confer charge loss, as seen in lysine acetylation, but also significantly reduce the size of the side chain (compared to acetylated lysine) and, hence, do not truly mimic acetyl-lysine. Non-acetylatable K to R substitution similarly affects the size of the side chain, albeit less dramatically than K to Q. These limitations are distinctly reflected in our biochemical analyses of ChAM variants with amino acid substitutions, properties of which did not match those of synthetic ChAM peptides with and without acetylated lysine residues (compare *Figures 2G and 3D*). Robust in vivo assessment of acetylation events would ideally involve the development of a chemically modifiable residue that exhibits improved similarity to acetylated lysine in a reversible manner.

This study highlights a new molecular mechanism by which KAT2A/2B controls genome stability. Our KAT2A/2B-dependent acetylome study identified 398 novel potential protein targets (*Fournier et al., 2016*), indicating that the mechanism of KAT2 action in controlling genome stability is multifaceted and more complex than previously anticipated. It is important to note that, in the cell, KAT2A/2B do not act on their own but within large protein complexes, namely the SAGA (Spt-Ada-Gnc5-Acetyltransferase) and the ATAC (Ada-two-A-containing Acetyltransferase) complexes, at transcriptionally active chromatin (*Nagy and Tora, 2007*). Components of these complexes regulate their enzymatic activity and substrate specificity. Markedly, ENY2 and ATXN7L3, components of the SAGA module that catalyses deubiquitination of histone H2Bub1 have been shown to play a role

in preventing unscheduled HR repair in human cells (*Evangelista et al., 2018*). We postulate that KAT2A/2B-containing complexes concurrently regulate the HR machinery and chromatin in undamaged cells, such that they ensure the activation of HR repair only when needed. KAT2A/2B-containing complexes are key regulators of processes controlling genome stability and will merit further investigation in the future.

More broadly, lysine acetylation is known to control biological processes, such as transcription, based on the metabolic status of the cell (*Lee et al., 2014*). Therefore, a better appreciation of the role of lysine acetylation in the DDR could expand the scope of studies aiming to understand how reprogrammed metabolism could increase genome instability in cancer cells (*Nowell, 1976*). Since PTMs are essential for physiological homeostasis and metabolites are necessary cofactors for the deposition of these PTMs, we envision that reprogrammed metabolism in cancer cells could alter the PTM landscape of DDR proteins and hence contribute to genome instability. Future studies assessing aberrant PALB2 lysine acetylation in cancer tissues and whether KDAC or bromodomain inhibitors, currently in clinical trials for cancer therapy, would modulate PALB2 acetylation may offer therapeutic potential. Furthermore, deciphering the metabolic regulation of the DDR and DNA repair could highlight additional cellular pathways that might be targeted for cancer therapy, possibly in combination with existing therapies that otherwise may give rise to resistance.

## Acknowledgements

We are grateful to Nicola O'Reilly (Francis Crick Institute Peptide Chemistry Technology Platform) for synthesis of ChAM peptides, Andrew Jefferson, Carina Mónico, Nadia Halidi, Niloufer Irani, Deirdre Kavanagh (Micron Oxford Advanced Bioimaging) as well as Yan Coulombe (CHU de Québec) for assistance with microscopy, Christine Ralf (Sir William Dunn School) for technical support in generating constructs, Michal Maj for assistance with FACS analysis and Chris Norbury (Sir William Dunn School) for critical reading of the manuscript. FE receives an MRC project grant (MR/W017601), was supported by a Wellcome Trust Senior Research Fellowship in Basic Biomedical Science (101009/Z/13/Z) and is thankful for the support from the Edward P Abraham Research Fund (RF 282). JYM is a Tier I Canada Research Chair in DNA repair and cancer therapeutics and is supported by a CIHR foundation grant (FDN-388879). SJS was supported by the Francis Crick Institute which receives its core funding from Cancer Research UK (FC001156), the UK Medical Research Council (FC001156), and the Wellcome Trust (FC001156). LT received funding from the European Research Council (ERC) Advanced Grant (ERC-2013–340551, Birtoaction).

## Additional information

### Funding

| Funder | Grant reference number | Author |
|---|---|---|
| Wellcome Trust | 101009/Z/13/Z | Fumiko Esashi |
| H2020 European Research Council | ERC-2013-340551 | László Tora |
| Edward P Abraham Research Fund | RF 260 | Fumiko Esashi |
| Canadian Institutes of Health Research | FDN-388879 | Jean-Yves Masson |
| Medical Research Council | MR/W017601 | Fumiko Esashi |
| Edward P Abraham Research Fund | RF 282 | Fumiko Esashi |
| Canada Research Chairs | Tier I Canada Research Chair in DNA repair and cancer therapeutics | Jean-Yves Masson |
| Francis Crick Institute | | Stephen J Smerdon |

| Funder | Grant reference number | Author |
| --- | --- | --- |
| European Research Council | Advanced Grant ERC-2013–340551 Birtoaction | László Tora |

The funders had no role in study design, data collection and interpretation, or the decision to submit the work for publication. For the purpose of Open Access, the authors have applied a CC BY public copyright license to any Author Accepted Manuscript version arising from this submission.

## Author contributions

Marjorie Fournier, Conceptualization, Resources, Data curation, Validation, Investigation, Visualization, Methodology, Writing – original draft, Writing – review and editing; Amélie Rodrigue, Formal analysis, Methodology, Validation, Visualization, Writing – review and editing; Larissa Milano, Formal analysis, Validation, Visualization, Methodology, Writing – review and editing; Jean-Yves Bleuyard, Resources, Data curation, Formal analysis, Validation, Investigation, Visualization, Methodology, Writing – review and editing; Anthony M Couturier, Data curation, Formal analysis, Investigation, Visualization, Writing – review and editing; Jacob Wall, Data curation, Formal analysis, Visualization, Writing – review and editing; Jessica Ellins, Data curation, Formal analysis, Investigation; Svenja Hester, Data curation, Investigation, Methodology; Stephen J Smerdon, Conceptualization, Resources, Methodology, Writing – original draft, Writing – review and editing; László Tora, Resources, Supervision, Methodology, Writing – review and editing; Jean-Yves Masson, Conceptualization, Formal analysis, Supervision, Funding acquisition, Validation, Investigation, Writing – review and editing; Fumiko Esashi, Conceptualization, Resources, Data curation, Formal analysis, Supervision, Funding acquisition, Validation, Investigation, Visualization, Methodology, Writing – original draft, Project administration, Writing – review and editing

## Author ORCIDs

Marjorie Fournier http://orcid.org/0000-0002-4075-8719
Amélie Rodrigue http://orcid.org/0000-0001-9883-7084
Jean-Yves Bleuyard http://orcid.org/0000-0002-6727-5362
Anthony M Couturier http://orcid.org/0000-0002-1512-9558
Stephen J Smerdon http://orcid.org/0000-0001-5688-8465
László Tora http://orcid.org/0000-0001-7398-2250
Jean-Yves Masson http://orcid.org/0000-0002-4403-7169
Fumiko Esashi http://orcid.org/0000-0003-0902-2364

## Decision letter and Author response

Decision letter https://doi.org/10.7554/eLife.57736.sa1
Author response https://doi.org/10.7554/eLife.57736.sa2

---

# Additional files

## Supplementary files

• Transparent reporting form

## Data availability

The mass spectrometry proteomics data have been deposited to the ProteomeXchange Consortium via the PRIDE (*Perez-Riverol et al., 2019*) partner repository with the dataset identifier PXD014678 and PXD014681. All other data generated or analysed during this study are included in the manuscript and supporting file. Raw images of western blots and DNA/protein gels are avilable through the Open Science Framework.

The following dataset was generated:

| Author(s) | Year | Dataset title | Dataset URL | Database and Identifier |
|---|---|---|---|---|
| Fournier M, Esashi F | 2022 | KAT2-mediated acetylation switches the mode of PALB2 chromatin association to safeguard genome integrity | https://doi.org/10.17605/OSF.IO/8E9MS | Open Science Framework, 10.17605/OSF.IO/8E9MS |

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

# Appendix 1

## Appendix 1—key resources table

| Reagent type (species) or resource | Designation | Source or reference | Identifiers | Additional information |
|---|---|---|---|---|
| Strain, strain background (*Escherichia coli*) | ArticExpress (DE3) | Agilent Technologies | | |
| Cell line (*Homo-sapiens*) | HEK293T | ATCC | | |
| Cell line (*Homo-sapiens*) | U2OS Flp-In T-REx | Prof. Daniel Durocher (Lunenfeld-Tanenbaum Research Institute) | | |
| Cell line (*Homo-sapiens*) | U2OS Flp-In T-REx FE-PALB2-WT | This study | | U2OS Flp-In T-REx harbouring inducible Flag-EGFP-PALB2 WT |
| Cell line (*Homo-sapiens*) | U2OS Flp-In T-REx FE-PALB2-ΔChAM | This study | | U2OS Flp-In T-REx harbouring inducible Flag-EGFP-PALB2 ΔChAM |
| Cell line (*Homo-sapiens*) | U2OS Flp-In T-REx P2shRNA | *Bleuyard et al., 2017b* | | U2OS Flp-In T-REx harbouring inducible shRNA targeting endogenous PALB2 |
| Cell line (*Homo-sapiens*) | U2OS Flp-In T-REx P2shRNA-EV | *Bleuyard et al., 2017b* | | U2OS Flp-In T-REx P2shRNA cells, complemented with empty vector (EV) |
| Cell line (*Homo-sapiens*) | U2OS Flp-In T-REx P2shRNA-FLAG-PALB2 | *Bleuyard et al., 2017b* | | U2OS Flp-In T-REx P2shRNA cells, complemented with 3xFLAG-PALB2 |
| Cell line (*Homo-sapiens*) | U2OS Flp-In T-REx P2shRNA-FLAG-PALB2 7Q | This paper | | U2OS Flp-In T-REx P2shRNA cells, complemented with 3xFLAG-PALB2-7Q |
| Cell line (*Homo-sapiens*) | U2OS Flp-In T-REx P2shRNA-FLAG-PALB2-7R | This paper | | U2OS Flp-In T-REx P2shRNA cells, complemented with 3xFLAG-PALB2-7R |
| Cell line (*Homo-sapiens*) | U2OS Flp-In T-REx P2shRNA- FE-PALB2-WT | This paper | | U2OS Flp-In T-REx P2shRNA cells complemented with inducible Flag-EGFP-PALB2 WT |
| Cell line (*Homo-sapiens*) | U2OS Flp-In T-REx P2shRNA- FE-PALB2-7Q | This paper | | U2OS Flp-In T-REx P2shRNA cells complemented with inducible Flag-EGFP-PALB2-7Q |
| Cell line (*Homo-sapiens*) | U2OS Flp-In T-REx P2shRNA- FE-PALB2-7R | This paper | | U2OS Flp-In T-REx P2shRNA cells complemented with inducible Flag-EGFP-PALB2-7R |
| Antibody | anti-FLAG (Mouse monoclonal) | Sigma | Cat# F1804 | WB (1:1000) ChIP (10 µg per sample) |
| Antibody | Control IgG (Mouse monoclonal) | Jackson Immunoresearch | 015-000-003 | WB (1:1000) ChIP (10 µg per sample) |
| Antibody | anti-pan-acetyl-lysine (Rabbit polyclonal) | Cell signalling Technology | Cat# 9441 S | WB (1:1000) |
| Antibody | anti-PALB2 (Rabbit polyclonal) | Bethyl | Cat# A301-246A | WB (1:500) |
| Antibody | anti-PALB2 (Rabbit polyclonal) | *Rodrigue et al., 2019* | | WB (1:5000) |
| Antibody | anti-BRCA2 (Mouse monoclonal) | Millipore | Cat# OP95 | WB (1:1000) |
| Antibody | anti-RAD51 (Rabbit polyclonal) | *Yata et al., 2014* | 7946 | WB (1:5000) |
| Antibody | anti-RAD51 (Rabbit polyclonal) | BioAcademia | 70–001 | IF (1/1000) |

*Appendix 1 Continued on next page*

*Appendix 1 Continued*

| Reagent type (species) or resource | Designation | Source or reference | Identifiers | Additional information |
|---|---|---|---|---|
| Antibody | anti-lamin A (Rabbit polyclonal) | Sigma | L1293 | WB (1:2000) |
| Antibody | anti-vinculin (Rabbit polyclonal) | Sigma | V9131 | WB (1:200000) |
| Antibody | anti-gamma H2AX (Mouse monoclonal) | Millipore | 05–636 | WB (1:1000) IF (1:2000) |
| Antibody | anti-MRG15 (Rabbit polyclonal) | Cell signalling Technology | Cat# D2Y4 | WB (1:1000) |
| Antibody | anti-BRCA1 (Mouse monoclonal) | Sigma | Cat# OP107 | WB (1:1000) |
| Antibody | anti-GST (Mouse monoclonal) | Santa Cruz Biotechnology | Cat# sc-138 | WB (1:1000) |
| Antibody | anti-biotin coupled with horseradish peroxidase (HRP) (Mouse monoclonal) | Sigma | Cat# A0185 | WB (1:1000) |
| Antibody | anti-KAT2A/GCN5 (Mouse monoclonal) | Cell signalling Technology | Cat# 3305 | WB (1:1000) |
| Antibody | anti-alpha-tubulin (Mouse monoclonal) | Cell signalling Technology | Cat# 3873 | WB (1:2000) |
| Antibody | Secondary antibody coupled with horseradish peroxidase (HRP) / Goat anti-mouse (Goat polyclonal) | Dako | Cat# P0447 | WB (1:1000) |
| Antibody | Secondary antibody coupled with horseradish peroxidase (HRP) / Goat anti-rabbit (Goat polyclonal) | Dako | Cat# P0448 | WB (1:1000) |
| Antibody | Secondary antibody coupled with horseradish peroxidase (HRP) / Goat anti-mouse (Goat polyclonal) | Jackson ImmunoResearch | 515-035-062 | WB (1: 20000) |
| Antibody | Secondary antibody coupled with horseradish peroxidase (HRP) / Goat anti-rabbit (Goat polyclonal) | Jackson ImmunoResearch | 111-035-144 | WB (1: 20000) |
| Antibody | Secondary antibody coupled with Alexa Fluor/ Goat anti-mouse (Goat polyclonal) | invitrogen | A-11001 | IF (1/1000) |
| Antibody | Secondary antibody coupled with Alexa Fluor/ Goat anti-mouse (Goat polyclonal) | invitrogen | A-11017 | IF (1/400) |
| Antibody | Secondary antibody coupled with Alexa Fluor/ Goat anti-rabbit (Goat polyclonal) | invitrogen | A-11011 | IF (1/1000) |
| Chemical compound, drug | sodium butyrate (NaB) | Sigma | 303410 | |
| Chemical compound, drug | Trichostatin (TSA) | Sigma | T8552 | |

*Appendix 1 Continued on next page*

*Appendix 1 Continued*

| Reagent type (species) or resource | Designation | Source or reference | Identifiers | Additional information |
|---|---|---|---|---|
| Chemical compound, drug | WST-1 reagent | Merck Life Sciences Uk Limited | 5015944001 | |
| Sequence-based reagent | siRNA: nontargeting control | Dharmacon | D001810-10-05 | 50 pmole |
| Sequence-based reagent | siRNA: targeting KAT2A | Dharmacon | L-009722-02-0005 | 50 pmole |
| sequence-based reagent | siRNA: targeting KAT2B | Dharmacon | L-005055-00-0005 | 50 pmole |
| Recombinant DNA reagent | pcDNA5/FRT-GW/N3×FLAG | *Bleuyard et al., 2017b* | | |
| Recombinant DNA reagent | pENTR3C | Invitrogen | | |
| Recombinant DNA reagent | pcDNA-DEST53 | Invitrogen | | |
| Recombinant DNA reagent | pCMV-SPORT6-PALB2 | Source BioSciences | IMAGE clone 6045564 | |
| Recombinant DNA reagent | pGEX-6P-1 | GE Healthcare | | GST expression in bacteria cells |
| Recombinant DNA reagent | pGEX-6P-1_PALB2 | This paper | | GST-PALB2 full length expression in bacteria cells |
| Recombinant DNA reagent | pGEX-6P-1_PALB2_Fr1 | This paper | | GST-PALB2 fragment 1 (1-320) expression in bacteria cells |
| Recombinant DNA reagent | pGEX-6P-1_PALB2_Fr2 | This paper | | GST-PALB2 fragment 2 (295-610) expression in bacteria cells |
| Recombinant DNA reagent | pGEX-6P-1_PALB2_Fr3 | This paper | | GST-PALB2 fragment 3 (580-900) expression in bacteria cells |
| Recombinant DNA reagent | pGEX-6P-1_PALB2_Fr4 | This paper | | GST-PALB2 fragment 4 (867–1186) expression in bacteria cells |
| Recombinant DNA reagent | pOG44 | Invitrogen | | |
| Recombinant DNA reagent | pENTR3C-ChAM #1 | This paper | | ChAM #1 (395-450) in the gateway entry vector |
| Recombinant DNA reagent | pENTR3C- ChAM #2 | This paper | | ChAM #2 (395-433) in the gateway entry vector |
| Recombinant DNA reagent | pENTR3C- ChAM #3 | This paper | | ChAM #3 (353-433) in the gateway entry vector |
| Recombinant DNA reagent | pENTR3C- ChAM #4 | This paper | | ChAM #4 (353-450) in the gateway entry vector |
| Recombinant DNA reagent | pENTR3C- ChAM #5 | This paper | | ChAM #5 (353-499) in the gateway entry vector |
| Recombinant DNA reagent | pENTR3C-PALB2 | *Bleuyard et al., 2017b* | | Full length PALB2 in the gateway entry vector |
| Recombinant DNA reagent | pENTR3C-PALB2-7Q | This paper | | Full length PALB2-7Q in the gateway entry vector |
| Recombinant DNA reagent | pENTR3C-PALB2-7R | This paper | | Full length PALB2-7R in the gateway entry vector |
| Recombinant DNA reagent | pcDNA-DEST53- ChAM #1 | This paper | | GFP-ChAM #1 (395-450) expression in mammalian cells |
| Recombinant DNA reagent | pcDNA-DEST53- ChAM #2 | This paper | | GFP-ChAM #2 (395-433) expression in mammalian cells |
| Recombinant DNA reagent | pcDNA-DEST53- ChAM #3 | This paper | | GFP-ChAM #3 (353-433) expression in mammalian cells |
| Recombinant DNA reagent | pcDNA-DEST53- ChAM #4 | This paper | | GFP-ChAM #4 (353-450) expression in mammalian cells |

*Appendix 1 Continued on next page*

*Appendix 1 Continued*

| Reagent type (species) or resource | Designation | Source or reference | Identifiers | Additional information |
|---|---|---|---|---|
| Recombinant DNA reagent | pcDNA-DEST53- ChAM #5 | This paper | | GFP-ChAM #5 (353-499) expression in mammalian cells |
| Recombinant DNA reagent | pGEX4T3 | GE Healthcare | | GST expression in bacteria cells |
| Recombinant DNA reagent | pGEX4T3-ChAM | *Bleuyard et al., 2012* | | GST-ChAM WT expression in bacteria cells |
| Recombinant DNA reagent | pGEX4T3-ChAM-3Q4K | This paper | | GST-ChAM-3Q4K expression in bacteria cells |
| Recombinant DNA reagent | pGEX4T3-ChAM-3K4Q | This paper | | GST-ChAM-3K4Q expression in bacteria cells |
| Recombinant DNA reagent | pGEX4T3-ChAM-7Q | This paper | | GST-ChAM-7Q expression in bacteria cells |
| Recombinant DNA reagent | pGEX4T3-ChAM-3R4K | This paper | | GST-ChAM-3R4K expression in bacteria cells |
| Recombinant DNA reagent | pcDNA5/FRT-GW/ N3×FLAG-PALB2 | *Bleuyard et al., 2017b* | | Constitutive 3xFlag- PALB2 expression in mammalian cells |
| Recombinant DNA reagent | pcDNA5/FRT-GW/ N3×FLAG-PALB2-7Q | This study | | Constitutive 3xFlag- PALB2 7Q expression in mammalian cells |
| Recombinant DNA reagent | pcDNA5/FRT-GW/ N3×FLAG-PALB2-7R | This study | | Constitutive 3xFlag- PALB2 7R expression in mammalian cells |
| Recombinant DNA reagent | pcDNA5/FRT/TO/FE-PALB2 | *Bleuyard et al., 2012* | | Inducible Flag-EGFP-PALB2 fusion expression in mammalian cells |
| Recombinant DNA reagent | pcDNA5/FRT/TO/FE-PALB2 7Q | This paper | | Inducible Flag-EGFP-PALB2 7Q expression in mammalian cells |
| Recombinant DNA reagent | pcDNA5/FRT/TO/FE-PALB2 7R | This paper | | Inducible Flag-EGFP-PALB2 7R expression in mammalian cells |
| Recombinant DNA reagent | pcDNA5/FRT/TO/FE-PALB2_ΔChAM | *Bleuyard et al., 2012* | | Inducible Flag-EGFP-PALB2 ΔChAM expression in mammalian cells |
| Sequence-based reagent | PALB2-F1_fo1 | This paper | PCR primers | 5'-atggatccatggacgagcctccc-3' |
| Sequence-based reagent | PALB2-F1_re1 | This paper | PCR primers | 5'-atgcggccgcattagaacttgtgggcag-3' |
| Sequence-based reagent | PALB2-F2_fo1 | This paper | PCR primers | 5'-atggatccgcacaaggcaaaaaaatg-3' |
| Sequence-based reagent | PALB2-F2_re1 | This paper | PCR primers | 5'-atgcggccgctgtgatactgagaaaagac-3' |
| Sequence-based reagent | PALB2-F3_fo1 | This paper | PCR primers | 5'-atggatccttatccttggatgatgatg-3' |
| Sequence-based reagent | PALB2-F3_re1 | This paper | PCR primers | 5'-atgcggccgcagctttccaaagagaaac-3' |
| Sequence-based reagent | PALB2-F4_fo1 | This paper | PCR primers | 5'-atggatcctgttccgtagatgtgag-3' |
| Sequence-based reagent | PALB2-F4_re1 | This paper | PCR primers | 5'-atgcggccgcttatgaatagtggtatacaaat-3' |
| Sequence-based reagent | PALB2_395_Fo | This paper | PCR primers | 5'- actggatcctcttgcacagtgcctg-3' |
| Sequence-based reagent | PALB2_353_Fo | This paper | PCR primers | 5'- actggatccaaatctttaaaatctcccagtg-3' |
| Sequence-based reagent | PALB2_450_Re | This paper | PCR primers | 5'- tatctcgagttaattttttacttgcatccttatttttta-3' |
| Sequence-based reagent | PALB2_433_Re | This paper | PCR primers | 5'- tatctcgagttacaaatgactctgaatgacagc-3' |
| Sequence-based reagent | PALB2_499_Re | This paper | PCR primers | 5'- tatctcgagttacaagtcattatcttcagtggg-3' |

*Appendix 1 Continued on next page*

*Appendix 1 Continued*

| Reagent type (species) or resource | Designation | Source or reference | Identifiers | Additional information |
|---|---|---|---|---|
| Sequence-based reagent | Patch 1-K-Rev | This paper | PCR primers | 5'-tcagagtcatttggatgtcaagaaaaaaggttt-3' |
| Sequence-based reagent | Patch 2-WT-Fwd | This paper | PCR primers | 5'-aaaaataaaaataaggatgcaagtaaaaat-3' |
| Sequence-based reagent | Patch 1-R-Rev | This paper | PCR primers | 5'-tcagagtcatttggatgtcaggagaagagggttt-3' |
| Sequence-based reagent | Patch 2-R-Fwd_FL | This paper | PCR primers | 5'-agaaatagaaatagggatgcaagtagaaatttaaacctttccaat-3' |
| Sequence-based reagent | Patch 1-Q-Rev | This paper | PCR primers | 5'-tcagagtcatttggatgtccagcaacaaggttt-3' |
| Sequence-based reagent | Patch 2-Q-Fwd_FL | This paper | PCR primers | 5'-caaaatcaaaatcaggatgcaagtcaaaatttaaacctttccaat-3' |
| Sequence-based reagent | Patch 2-Q-Fwd_ChAM | This paper | PCR primers | 5'-caaaatcaaaatcaggatgcaagtcaaaattgagcggccgcact-3' |
| Sequence-based reagent | Beta-Actin_in3-fo | This paper | PCR primers | 5'-taacactggctcgtgtgacaa-3' |
| Sequence-based reagent | Beta-Actin_in3-re | This paper | PCR primers | 5'-aagtgcaaagaacacggctaa-3' |
| Sequence-based reagent | Chr5_TCOF1_peak2_fo | This paper | PCR primers | 5'-ctacccgatccctcaggtca-3' |
| Sequence-based reagent | Chr5_TCOF1_peak2_re | This paper | PCR primers | 5'-tcagggctctatgaggggac-3' |
| Sequence-based reagent | Chr11_WEE1_mid_fo | This paper | PCR primers | 5'-ggccgaggcttgaggtatatt-3' |
| Sequence-based reagent | Chr11_WEE1_mid_re | This paper | PCR primers | 5'-ataaccccaaagaacacaggtca-3' |
| Peptide, recombinant protein | ChAM-WT | This paper | Francis Crick Institute Peptide Chemistry Technology Platform | AEKHSCTVPEGLLFPAEYYVRTTRSMSNCQRKVAVEAVIQSHLDVKKKGFKNKNKDASKN |
| Peptide, recombinant protein | ChAM-K436(Ac)-K437(Ac)-K438(Ac) | This paper | Francis Crick Institute Peptide Chemistry Technology Platform | AEKHSCTVPEGLLFPAEYYVRTTRSMSNCQRKVAVEAVIQSHLDVK(Ac)K(Ac)K(Ac)GFKNKNKDASKN |
| Peptide, recombinant protein | ChAM-K436(Ac) | This paper | Francis Crick Institute Peptide Chemistry Technology Platform | AEKHSCTVPEGLLFPAEYYVRTTRSMSNCQRKVAVEAVIQSHLDVK(Ac)KKGFKNKNKDASKN |
| Peptide, recombinant protein | ChAM-K437(Ac) | This paper | Francis Crick Institute Peptide Chemistry Technology Platform | AEKHSCTVPEGLLFPAEYYVRTTRSMSNCQRKVAVEAVIQSHLDVKK(Ac)KGFKNKNKDASKN |
| Peptide, recombinant protein | ChAM-K438(Ac) | This paper | Francis Crick Institute Peptide Chemistry Technology Platform | AEKHSCTVPEGLLFPAEYYVRTTRSMSNCQRKVAVEAVIQSHLDVKKK(Ac)GFKNKNKDASKN |
| Software, algorithm | Image J | https://imagej.nih.gov/ij/ | *Schneider et al., 2012* | |
| Software, algorithm | Proteome Discoverer v1.4 | ThermoFischer Scientific | | |
| Software, algorithm | FlowJo software V10 | FlowJo | | |
| Commercial assay, kit | SensiFAST SYBR No-Rox kit | Bioline | BIO-98005 | |

