## [Editor Report]

The manuscript provides fundamental insights into the role of acetylation of PALB2, a protein involved in Fanconi anemia and homologous recombination though its association with BRCA1 and BRCA2. The evidence that PALB2 acetylation regulates its nuclear mobility is multi-faceted and convincing and the major strength is the definition of the role of de-acetylation of the ChAM domain of PALB2 to mobilize the protein under genotoxic stress. Individuals with an interest in genome stability will be the audience for this important study.

---

## [Decision Letter]

[Editors' note: this paper was reviewed by Review Commons.]

Thank you for submitting your article "KAT2-mediated acetylation switches the mode of PALB2 chromatin association to safeguard genome integrity" for consideration by *eLife*. Your article has been reviewed by 3 peer reviewers, and the evaluation has been overseen by a Reviewing Editor and Jessica Tyler as the Senior Editor.

The reviewers have discussed the reviews with one another and the Reviewing Editor has drafted this decision to help you prepare a revised submission.

The manuscript by Fournier et al. highlights the importance of acetylation in the ChAM domain of PALB2 in regulating nucleosome binding and DNA repair. The text is well-written text and the experiments are well-designed. we read the manuscript, the reviews as well as the rebuttal and plans for a revision. We believe the revision and the proposed added experiments will be needed to cement the conclusions. Of the 5 experiments, #1 and #2 are critical, and we believe #4 and #5 are also important as BRCA1 is a key factor for PALB2 (#5) and the effect on HR (#4) should be documented experimentally. We do not consider experiment #3 (PALB2 foci) as critical. We encourage the authors to plan and execute this revision as they outlined with the above exception of experiment #3. You may want to consider combining KAT2A/B depletion and/or using KDAC inhibitors in the experiments with the 7R and 7K mutants, but we leave that suggestion to them. The authors may take as long as necessary to deliver the revision, under the present COVID19 guidelines of *eLife*. We look forward to receiving the revised manuscript.

[Editors' note: further revisions were suggested prior to acceptance, as described below.]

Thank you for resubmitting your work entitled "KAT2-mediated acetylation switches the mode of PALB2 chromatin association to safeguard genome integrity" for further consideration by *eLife*. Your revised article has been evaluated by Jessica Tyler (Senior Editor) and a Reviewing Editor.

The manuscript has been improved but there are some remaining issues that need to be addressed, as outlined below:

The efforts in this revision were appreciated by all reviewers. However, as detailed in reviews #1 and 3, there are significant concerns about the overall model and specific experiments, which will require significant additional experimentation.

The key issues are:

1) A fuller characterization of the 7R mutant is required in the HR-reporter assay, the sensitivity assays, in the nucleosome binding assay, and in ChIP assays at transcriptionally-active loci along with some controls. The importance of the analysis of the 7R mutant was highlighted in the previous review cycle.

2) The authors should acknowledge inconsistencies and revise their model to be compatible with their data.

The reviews contain specific critiques and concerns that will require significant changes in the text. The authors may want to consider removing the DNA binding assays.

*Reviewer #1 (Recommendations for the authors):*

In this revised manuscript by Fournier et al. (*eLife*-57736R1), the authors have added experiments and analysis to address concerns raised previously. This reviewer appreciates the elegant systems employed in much of this work, as well as the effort that went into this revision. The revised manuscript continues to offer potential insights into the regulation and function of PALB2 in DNA repair, and into the role of acetylation in regulating DNA damage responses. However, there are still major concerns, which in the opinion of this reviewer, should be addressed prior to potential publication. These include issues with certain newly added figures, and apparent inconsistencies or contradictions between different results. As a consequence, the model presented in Figure 7D is not yet fully supported.

1. The 2nd paragraph on p. 21 of the unmarked revision discusses levels of γ-H2AX shown in Figures 5D-E in the context of DNA replication (this paragraph is part of a subsection of the Results beginning on p. 20). Earlier in this same paragraph the authors hypothesize that "PALB2 association with active genes … prevents the induction of DNA damage during DNA replication". In this context, Figures 5D-E are misleading and are not necessarily related to DNA replication. Specifically, the cells were not synchronized and DNA damage arises over several days, leading to questions about the relationship to DNA replication. This would be better supported by performing ChIP assays using γ-H2AX antibody to determine whether DNA damage specifically accumulates at transcriptionally active sites, where conflicts with DNA replication may arise, as compared to other sites. Further, no measures of statistical significance are given for Figure 5E.

2. While HR data was added at the request of a reviewer, the intent is presumably to strengthen the hypothesis that ChAM acetylation regulates PALB2 function in DNA repair. Four points about this:

A) No measure of significance is given for the HR results shown in Figure 7—figure supplement 1. Further, the error bars are often very large. As such, no convincing conclusions can be made about the effects of the 7Q and 7R mutants on HR.

B) In any case, if the finding was sufficiently established that both the 7Q and 7R mutants disrupt DSB-HR, as suggested by the authors, how do they rectify this with the observation that only 7Q but not 7R sensitizes U2OS cells to Olaparib in Figure 7C? Does the 7R mutant perturb HR with no consequences for cellular survival in the presence of DNA damage? And, if so, what additional function is perturbed by the 7Q mutant that results in sensitivity to Olaparib?

C) In the last paragraph on p. 26, the authors write, "We suggest that damage-induced ChAM deacetylation allows its recruitment to sites of DNA damage through its interaction with damage sensing factors, fulfilling its function in promoting HR repair". If all of this is true, since PALB2-7R was never acetylated and therefore did not need to be deacetylated or released from sites of active transcription, then, according to their model, why is this mutant defective for HR (Figure 7—figure supplement 1)? Is this finding consistent with the model presented in Figure 7D?

D) On p. 27 (Discussion), the authors state, "Interestingly, the PALB2 7R variant, unlike its 7Q counterpart, had little impact on RAD51 foci formation … This could be explained by our findings that DNA damage triggers ChAM deacetylation and promotes PALB2 mobilisation, which we propose is the critical event in initiating HR repair". This statement seems to overlook data shown in Figure 7—figure supplement 1 that claims PALB2-7R appears is deficient for HR. Further, proficiency for RAD51 foci in response to IR typically is associated with functional HR, but 7R has nearly normal RAD51 foci while suggested to be deficient for HR. Can the authors' explain this based upon their model?

3. In the model presented in Figure 7D, the authors suggest (1st paragraph of the Discussion, on p. 24) that DNA damage triggers ChAM deacetylation (presumably on PALB2 localized to transcriptionally active chromatin) which "releases PALB2 from active genes and increases its mobility". I have the following concerns about the model (in addition to those detailed in point #2):

A) When discussing their model, it is currently unclear whether the authors are proposing that ChAM deactylation is occurring specifically at "transcriptionally active chromatin" in response to DNA damage and whether this makes PALB2 "globally" available for HR.

B) The data shown in Figures 4B-C seems to contradict the authors' model (Figure 7D). If the role of ChAM deacetylation is simply to release "PALB2 from active genes and increase(s) its mobility", then the 7Q mutant (mimic of acetylated lysine), by being more mobile (Figure 7D-E), should be available for "HR repair complexes" that form in chromatin. Because the 7Q mutant shows less association with chromatin than WT PALB2 or the 7R mutant (Figures 7B-C), one wonders if the 7Q mutant impairs some other PALB2 function besides release from transcriptionally active chromatin upon DNA damage. Additionally, the 7R (non-acetylable) mutant appears functional as indicated by normal cellular resistance to Olaparib (Figure 7C). This does not support any claim there is a specific need that PALB2 be recruited to transcriptionally active chromatin and subsequently released.

C) Unless I overlooked it, it seems that the authors did not specifically test the effect of the 7R mutant on the association of PALB2 (or the ChAM domain) with nucleosomes; this may be helpful in better interpreting various results in the context of their model.

D) Since the 7R mutant shows even less association with transcriptionally active chromatin (Figure 5F) than PALB2-7Q, and PALB2-7R is resistant to Olaparib (Figure 7C), then it would seem that protection of "transcriptionally active chromatin … during DNA replication (2nd paragraph of p. 24) is not essential for cell survival in response to DNA damage.

E) The model that acetylation of the 7K patch releases PALB2 from transcriptionally active chromatin would be better supported by testing this in the presence of IR (or Olaparib), since DSB-initiated HR and cellular resistance to Olaparib are among DNA damage response-related assays featured in the manuscript. Does IR or Olaparib indeed release WT PALB2 from transcriptionally-active loci?

F) According to the model presented, are transcriptionally-active loci left deprotected after acetylation of the ChAM motif that results in increased mobilization of PALB2 for HR?

G) In the absence of assays of PALB2 with transcriptionally active loci after exposure to IR (or PARP inhibitor), the authors have not convincingly demonstrated that PALB2 is mobilized from these loci after DNA damage. While the authors note that Bleuyard et al. 2017b demonstrate that CPT induces decreased association of PALB2 with transcriptionally active chromatin, CPT could have a different effect than IR by inducing more replication stress.

4. The authors note that both K to R and K to Q substitutions alter the size of the side chain, albeit to different degrees. How confident are the authors that K to R and K to Q substitutions are accurate non-acetyl and acetyl mimics, respectively? Is it possible that both substitutions are simply disrupting the structure of the ChAM, and perhaps PALB2 more globally, to some degree?

5. In contrast to association of PALB2 with transcriptionally active chromatin regulating interaction with BRCA1, as proposed, structural changes across the ChAM, or in PALB2 more broadly, could potentially cause the observed decreased interactions of the 7R and 7Q PALB2 mutants with BRCA1.

*Reviewer #2 (Recommendations for the authors):*

Most of the concerns raised previously have been taken into account and the manuscript has now been much improved. I can therefore recommend this manuscript for publication.

*Reviewer #3 (Recommendations for the authors):*

Fournier et al. – KAT2-mediated acetylation switches the mode of PALB2 chromatin association to safeguard genome integrity

The authors have responded to the comments made by each reviewer, and have made several amendments to the manuscript itself, generally improving its legibility and providing additional supporting data / experiments. Saying this, there are still some outstanding questions that should be addressed:

A) It is still not clear as to the (intended) purpose of the GST-ChAM DNA-binding experiments and what these actually contribute to the manuscript as a whole. They could be readily removed from the manuscript without impacting the set of conclusions.

Note: there is no appreciable DNA binding by any of the mutant forms of the GST-ChAM construct as compared to the two controls, WT or indeed PALB2-FL (how was this protein produced, is this the bacterially expressed protein?).

It is therefore not clear how the authors can support the following statement.

Page 18, Lines 444-446: "… however, the ChAM 3R4K variant also showed impaired binding to DNA in an EMSA assay, albeit less pronounced than that of its K to Q (3Q4K) counterpart (Figure 2H)".

B) Whilst it is appreciated that the authors have now created two additional GST-ChAM construct (3Q4K and 3R4K) they haven't performed the requested control experiment, i.e., with the 7R construct. This is necessary to support the set of experiments described at a later point in the manuscript. What is the effect of the 7R construct in the GST-pulldown experiments – does this still bind to nucleosomes / histone H3?

C) Figure 3F, why is there a such a significant increase in PALB2 mobility in the FRAP experiment for the negative control siCntrl? See also Figure 3—figure supplement 1

D) The data for MMC/Olaparib treatment presented in the response to authors (Figure R3) should be included as supplementary data within the revised manuscript.

Page 23, Lines 572-576

"… though similar changes in ChAM acetylation were not detectable upon MMC or olaparib DNA damaging treatment. We noted that IR triggered strong γ- H2A.X induction (Figure 6B), which was not observed upon exposure of HEK293T cells to MMC or olaparib, suggesting that the reduction of ChAM acetylation reflects the cellular response to DNA damage.

E) Figure 7 —figure supplement 1, panel B (HR assay)

The means from individual experiments should be displayed on the graph, as it is not clear if the wide error bar (which presumably should also extend downwards, as + and – 1 SD) and thus interpretation of the result is due to a single outlier measurement.

F) Discussion

The authors allude to unpublished data within the discussion / conclusion – such statements should ideally be removed.

Also, our preliminary results indicate that a PALB2 variant defective in BRCA1 binding exhibits higher accumulation at genic regions than its wild-type counterpart.

Indeed, our preliminary results demonstrated that extracellular glucose concentration affect PALB2 dynamics, as reducing glucose level in the growth medium increases PALB2 dynamics.

[Editors' note: further revisions were suggested prior to acceptance, as described below.]

Thank you for resubmitting your work entitled "KAT2-mediated acetylation switches the mode of PALB2 chromatin association to safeguard genome integrity" for further consideration by *eLife*. Your revised article has been evaluated by Jessica Tyler (Senior Editor) and a Reviewing Editor.

The manuscript has been significantly improved but there are some remaining issues that need to be addressed, as outlined below:

In the current revision of their manuscript, Fournier et al. have, in general, addressed the concerns that were raised previously. This includes removal of confounding results and addition of new data, such as survival assays, recruitment to sites of DNA damage, and ChIP assays for occupancy at transcriptionally active sites with and without IR, for WT PALB2, and the 7Q and 7R mutants. Overall, the data and presentation are now much more cohesive and are integrated into an interesting model in Figure 9E. As in previous versions, an impressive array of complementary techniques have been employed in this manuscript. This manuscript will make an important contribution to the literature on DNA damage responses.

Having said all of this, it should be noted that the manuscript still requires certain clarifications in the text and/or figures. This will not require additional experimentation only text changes and clarifications that would strengthen this manuscript further and improve overall clarity.

1) There needs to more clarity, about roles in DNA repair/cell survival away from transcriptionally active genes, of the full cellular pool of PALB2 vs the sub-fraction that is specifically associated with transcriptionally active sites. a) Do the authors propose that any PALB2 remains at transcriptionally active sites after the induction of DNA damage elsewhere and if so whether this may provide ongoing protection of active genes and b) whether the authors are proposing that a PALB2-BRCA2-RAD51 complex is mobilized from transcriptionally active genes and whether the PALB2-BRCA2-RAD51 complex exists away from transcriptionally active genes at steady state.

2) The statements about the role of the 7K-patch in HR needs to be softened and clarified how results obtained with the 7R and 7Q variants provide complementary information about PALB2's regulation and function in response to DNA damage. There were reasons for removing data from HR reporter assays, but a reader may consider conclusions about the role of the 7K-patch in HR based only on RAD51 foci to not be fully supported. As such, it is suggested that the authors briefly discuss their HR reporter assay results (in either the Results or Discussion) as data not shown and indicate why these results may not be congruent with those obtained for RAD51 foci. Finally, in the absence of HR reporter assay data, I would suggest toning down the headings in various places to less definitively link the 7K patch to HR repair (line 610; lines 1026-1027; lines 1044-1045. In general, RAD51 focus formation cannot be equalized with HR.

3) A measure of the statistical significance of differences in Figure 4C is needed.

4) It looks like there is a non-specific band in the blot for PALB2 in Figure 6A. It would be helpful to indicate, with a marker in Figure 6A and/or by the description in the figure legend, which, if any, of the two bands is non-specific there.

5) The foci are often difficult to see without significant enlargement in Figures 7D and 8D. Perhaps making this figure part larger relative to the overall figure would help with this. Alternatively, the intensities could be enhanced in the images if the current relative size of the figure part is retained.

6) There is a single statement that appears to have been carried forward from the previous version which should be removed (page 22, line 524-525) 'while suppressing non-specific DNA binding'.

---

## [Author Response]

The manuscript by Fournier et al. highlights the importance of acetylation in the ChAM domain of PALB2 in regulating nucleosome binding and DNA repair. The text is well-written text and the experiments are well-designed. we read the manuscript, the reviews as well as the rebuttal and plans for a revision. We believe the revision and the proposed added experiments will be needed to cement the conclusions. Of the 5 experiments, #1 and #2 are critical, and we believe #4 and #5 are also important as BRCA1 is a key factor for PALB2 (#5) and the effect on HR (#4) should be documented experimentally. We do not consider experiment #3 (PALB2 foci) as critical. We encourage the authors to plan and execute this revision as they outlined with the above exception of experiment #3. You may want to consider combining KAT2A/B depletion and/or using KDAC inhibitors in the experiments with the 7R and 7K mutants, but we leave that suggestion to them. The authors may take as long as necessary to deliver the revision, under the present COVID19 guidelines of eLife. We look forward to receiving the revised manuscript.

We thank all the reviewers for taking the time to evaluate our manuscript and for giving us an opportunity to revise our manuscript. As recommended by the *eLife* Editor, we conducted four key experiments, the results of which are now incorporated in the revised manuscript:

Experiment 1: Biochemical analyses of ChAM 7R mutant (Figure 2G and H).Experiment 2: ChIP-qPCR of PALB2 variant at previously identified PALB2 bound genes (Figure 5F)Experiment 4: PALB2 variant homology-directed gene targeting assay (Figure 7—figure supplement 1B)Experiment 5: PALB2 variant interaction with BRCA1 (Figure 7—figure supplement 1D)

In the revised manuscript, we have further extended our discussion to clarify concerns raised by the reviewers. We believe that these newly incorporated results and rewriting significantly improve our manuscript, shedding light on the importance of PALB2 acetylation at the ChAM in both normally growing cells and upon DNA damage.

Below, we provide an updated version of our point-to-point response to the reviewers. Please note that the Figure numbers referenced by the reviewers in the original manuscript do not match those of the revised manuscript. To facilitate straightforward review, our responses include the relevant Figure, page and line numbers for the revised manuscript.

Point-by-pointReviewer #1 (Evidence, reproducibility and clarity (Required)):Fournier et al. detect acetylation within the chromatin association motif (ChAM) of PALB2 and demonstrate that KAT2 can acetylate these 7 lysine residues within this region. They then generate K to R mutations (7R) or K to Q mutations (7Q) at these sites and perform assays of fluorescence recovery after photobleaching (FRAP) to measure mobility as a measure of chromatin association, RAD51 foci, PALB2 recruitment at sites of laser-induced DNA damage, and sensitivity to olaparib. They find increased mobility of the 7Q mutant of PALB2 but not 7R in the absence of exogenous DNA damage, as well as defects in DNA damage-induced RAD51 foci and resistance to olaparib. On this basis, the authors conclude that acetylation is required for the association of PALB2 with undamaged chromatin and that deacetylation permits mobilization and association with BRCA1 to enable proper DNA repair. While the manuscript is generally well-written, many of the systems are rather elegant, and this study may yield novel insights into the regulation and function of PALB2 in DNA repair, there are some missing experiments to be added and important contradictions that should be resolved in order to fully establish the new model the authors propose.Major comments:1. There are some concerns about the interpretation of experiments with the 7R and 7Q mutants of PALB2. For example, in the description of results in Figure S2C, the authors state "K to R substitutions maintain the charge yet are unable to accept acetylation and hence mimic constitutively non-acetyl lysine". However, in Figure 4B the association of the 7R mutant with chromatin is similar to WT and in Figure 7D,E the relative immobility of the 7R mutant is very similar to WT PALB2. Thus, the conclusion that acetylation is required for PALB2 association with damaged undamaged chromatin and for release of PALB2 upon DNA damage does not appear justified. Perhaps the authors need to better consider whether the 7R mutant mimics acetylation because of its charge. Even so, the mutant then maintains the charge normally associated with acetylated PALB2, calling into question whether deacetylation indeed "releases PALB2 from undamaged chromatin".

We agree with the reviewer’s point that there is no or little difference between WT and the 7R mutant in regard to their enrichment on non-damaged chromatin, as detected by fractionation (Figure 4B and C), or their mobility, as detected by FRAP (Figure 4D and E). To better address the question raised by the reviewer, we have conducted biochemical analyses of the ChAM K to R substitutions, testing their direct interaction with DNA (Experiment 1; Figure 2G and H).

Our analysis of the ChAM 3R4K variant, substituting three conserved lysine residues K436, K437 and K438 with R, revealed that K to R substitutions decreased its binding to DNA, although to a lesser extent than that of a ChAM Q mutant (3Q4K), compared to WT. This observation indicates that (1) ChAM interaction with DNA is mediated not solely by its electrostatic charge, (2) K to R substitutions alters ChAM biochemical properties, and (3) the 7R mutation likely impairs PALB2 association with chromatin as a result of its reduced affinity for DNA. This is described on page 18, lines 437-446.

The discrepancy between this new finding and the fractionation/FRAP results can be explained by the fact that, in a cellular context, full-length PALB2 is enriched at a fraction of H3K36me3-marked exons (which comprise only 1-1.5 % of the genome), as shown in our previous genome-wide ChIP-seq analysis (Bleuyard et al., 2017, PNAS). We reasoned that bulk fractionation or FRAP analyses might not be sensitive enough to highlight the impact of the 7R mutation, but the ChIP-qPCR method, detecting PALB2 association at defined genic regions would be more appropriate. Thus, we have expanded our ChIP-qPCR analyses to validate our proposed model (Experiment 2: Figure 5F).

Indeed, our ChIP-qPCR analysis of PALB2 variants at previously identified PALB2-bound loci (i.e. *ACTB*, *TCOF1* and *WEE1*) revealed a statistically significant decrease of PALB2 7R at these loci compared to WT PALB2 in undamaged conditions. This observation supports that PALB2 7R elicits decreased PALB2 association at these loci and that bulk fractionation or FRAP analyses were unsuited to detect the impact of the 7R mutation. This point is now discussed on page 22-23, lines 541-560.

Overall, the new results led us to revise our interpretation of the 7R mutant. While K to R substitutions alter the biochemical properties of ChAM, the R-associated electrostatic charge, analogously to non-acetylated K, allows dynamic PALB2 association with chromatin except at those actively transcribed loci which are normally bound by PALB2. We propose that, at KAT2A/B-enriched actively transcribed regions, PALB2 is targeted for acetylation, which in turn promotes PALB2 association with chromatin, likely together with MRG15-mediated tethering. This mechanism might potentially contribute to the suppression of HR repair at actively transcribed regions in undamaged cells. This point is now discussed on pages 29-30, lines 726-740.

2. Related to questions of interpreting results utilizing the 7R and 7Q mutants of PALB2, in Figure 7B,C the 7R mutant but not 7Q supports RAD51 foci and resistance to olaparib similar to WT PALB2. The authors then state in the Discussion that "our work also suggests that caution should be exercised in the use of K to Q substitutions for functional studies of lysine acetylation". Thus, which mutant is giving the correct and reliable results?

We apologise for the miscommunication if this point was unclear. Using biochemical approaches, we established that ChAM acetylation, but not K to Q substitutions, facilitates its association with nucleosomes (please compare Figure 2E and Figure 3B). This observation clearly demonstrates that K to Q substitutions do not mimic acetylation at these residues, but instead render PALB2 ChAM functionally null. The PALB2 7Q phenotypes therefore demonstrate the importance of the 7K-patch for ChAM function in HR repair, rather than its acetylation status. In the revised manuscript, we have emphasised the differences between lysine acetylation and the K to Q substitutions throughout (e.g. page 18-19, lines 446-450; page 19, lines 461-465; page 29, lines 716719).

Perhaps even more importantly, if results with the 7Q mutant are suspect, the conclusion that deacetylation is required for HR (or DNA repair) is suspect because that is the only case where the authors see a defect in RAD51 foci and resistance to olaparib. Similarly, if the 7R mutant "mimics nonacetyl-lysine" then the fact that it has normal RAD51 foci and resistance to olaparib contradicts the conclusion that deacetylation is required for DNA repair.

As responded in the previous points, we now articulated our view in the revised manuscript. Briefly, K to Q substitutions confer loss of charge but also significantly reduce the size of the lysine side chain compared to that of an acetylated lysine, as depicted in Figure 2—figure supplement 1B. We thus believe that changes in the side chain size might negatively impact on PALB2/ChAM binding to chromatin. This thus makes it impossible to assess the effect of charge loss in the context of the 7Q mutation. Importantly, however, the 7Q mutant, which behaves like a ChAMnull mutant as defined in page 18-19, lines 446-450, was used to demonstrate the importance of ChAM 7K-patch in normally growing conditions (Figure 5) and HR-mediated DNA repair (Figure. 7A-C and Figure 7—figure supplement 1B).

K to R substitutions also alter the side chain size, i.e. a modest size increase compared to nonacetylated lysine (Figure 2B). We believe that this affects impaired binding of the ChAM K to R variant to naked DNA in vitro (Figure 2H). Cells expressing the PALB2 7R variant showed delayed S phase progression, DNA damage accumulation in undamaged condition (Figure 5A-E), loss of PALB2 association at defined PALB2 target loci (Figure 5F) and somewhat impaired gene-targeting efficiency upon ZFN-induced DSB induction at the AAVS1 locus (Figure 7—figure supplement 1B). Conversely, we observed little defects of the 7R variant upon exposure to genotoxic treatments in terms of its recruitment to DNA damage sites (Figure 7—figure supplement 1E and F), RAD51 foci formation (Figure 7A and B) and olaparib resistance (Figure 7C). The seemingly normal behaviour of the 7R variant can be explained by our observation that the induction of DNA damage triggers ChAM deacetylation (Figure 6B and C) and an increase of PALB2 mobility (Figure 6D-G). In other words, under conditions where the DNA damage response (DDR) is robustly activated, K to R substitutions might more closely mimic a de-acetylated form of PALB2 and be beneficial for HR.

In summary, it seems clear that amino acid substitutions, K to either Q or R, are not fully optimal in assessing the real impact of acetylation and deacetylation in cellular contexts. Nonetheless, we believe these mutants are useful tools to understand the importance of target residues and to infer the potential impact of acetylation. This point is discussed extensively on pages 28-29, lines 690-723.

3. There are multiple concerns about Figures 5 and S5. In Figure 5A-C, difference in cell cycle progression after synchronization are relatively small and no rationale/interpretation is given for how this may be related to PALB2 function is given. In Figure 5D,E differences in the levels of γ-H2AX as a marker of DNA damage between different forms of PALB2 do not become readily apparent until about 6 or more days after addition of doxycycline. As such, it seems that these could be indirect effects and it is unclear how strongly this supports the importance of PALB2 acetylation in the DNA damage response.

We apologise if these points were unclear. We have previously established that steadystate PALB2 chromatin association, jointly mediated by the ChAM and MRG15 interaction, protects a subset of active genes from DNA damage that may otherwise arise from replication-transcription conflicts (Bleuyard et al., 2017, PNAS). The results presented in Figure 5 led us to propose that PALB2 chromatin association is, at least in part, mediated by the ChAM 7K patch, and its impairment (either by 7Q or 7R substitutions) leaves actively transcribed genes normally bound by PALB2 unprotected and exposed to DNA damage during replication. This model nicely supports our observations that both 7Q/7R mutants exhibit slow S-phase progression and accumulation of γ-H2AX over time. These points are now articulated in the revised manuscript on pages 21-23, lines 523-560:

In Figure S5, it is interesting that there are differences in the association of different forms of PALB2 with 3 distinct active loci, but no error bars or measures of statistical significance are given. Further at 2 of the 3 loci, the association of the 7Q mutant is closer to WT than the 7R mutant. Taken together, neither Figure 5 nor Figure S5 strongly support the key conclusion that acetylation regulates the association of PALB2 with actively transcribed genes to protect them.

We appreciate this criticism. To better evaluate how the ChAM 7K-patch impacts on the association of PALB2 with actively transcribed genes, we have performed biological replicates of the ChIP-qPCR analyses of Flag-PALB2 WT, 7R and 7Q at *ACTB*, *TCOF1* and *WEE1* loci (Experiment 2; Figure 5F). The results showed consistently that the 7K substitutions to either 7R or 7Q confer statistically significant decreases in their binding to these loci compared to PALB2 WT. This observation supports our model that PALB2 7Q/7R variants leave actively transcribed genes unprotected and exposed DNA damage, which can cause cellular defects such as slower cell cycle progression.

4. Figures 6D-G and S6A-D conclude that "DNA damage triggers ChAM deacetylation and induces PALB2 mobilization" based upon FRAP experiments utilizing WT PALB2. But there is no control to demonstrate that this is a specific effect driven by the state of PALB2 acetylation. For example, DNA damage might cause global acetylation changes resulting in relaxed chromatin in which proteins that are not subject to acetylation-deacetylation also show increased mobility.

We thank the reviewer for this valuable comment. It is true that we cannot formally exclude the possibility that changes in PALB2 mobility are indirect consequence of damage-induced chromatin reorganisation/increased chromatin mobility. However, our analyses clearly demonstrate that ChAM acetylation increases its association with nucleosomes (Figure 3B), while non-nucleosome binding ChAM-null (7Q or deletion) increases PALB2 mobility (Figure 2E, Figure 4E and Figure 4—figure supplement 1C). Further, WT PALB2 mobility increases after KAT2 depletion (i.e. reduction of chromatin acetylation of KAT2 targets, hence leading to chromatin compaction) (Figure 3F), but reduces upon KDAC inhibition (i.e. global increase in acetylation, hence leading to chromatin relaxation) (Figure 3G). Considering all these observations collectively, the increase in PALB2 mobility detectable upon DNA damage is unlikely to reflect global chromatin relaxation, and that PALB2 acetylation influences its mobility in both challenged and unchallenged cells. We have clarified this point on page 24, lines 583-589.

5. Figure 7B shows that the 7Q mutant has diminished RAD51 foci while Figure S7C,D suggests based upon a different methodology (laser-induced damage) that the 7Q mutant does not affect PALB2 recruitment. Since the issue of recruitment is key to the mechanism proposed, the authors should examine PALB2 foci instead as this may be a more sensitive assay of PALB2 recruitment.

We appreciate the reviewer’s point. Nonetheless, we believe that the laser-induced experiments provide high sensitivity and resolution for PALB2 recruitment kinetics, as the data were obtained with real-time live-cell imaging. In line with this observation, our preliminary foci analysis indicated that PALB2 7Q was efficient in forming foci upon DNA damage (data not shown). We therefore did not extend this analysis, in accordance with the *eLife* editor recommendation, where PALB2 foci analyses were considered not essential (Experiment 3).

6. The authors state in the last sentence of the Results section that "lysine residues within the ChAM 7K-patch are indispensable for PALB2 function in HR" but never test the mutants for HR using reporter assays. The manuscript would be strengthened by performing such assays.

We have now conducted homology directed recombination (HDR) reporter assays using cells expressing the PALB2 7K variants (Experiment 4; Figure 7—figure supplement 1B). Here, we assessed the gene-targeting efficiency of a GFP-reporter construct that can be integrated at the AAVS1 locus as a readout for HDR. This approach revealed that, compared to cells expressing WT PALB2, GFP integration at the AAVS1 locus upon ZFN-induced DSB was barely detectable in cells expressing PALB2 7Q or 7R (Figure 7—figure supplement 1B and outlined on pages 24-25, lines 603609). This observation supports the notion that the PALB2 7K-patch is indeed important for HDR at this locus under the conditions employed.

Of note, unlike this ZFN-induced HDR reporter assay, the PALB2 7R mutant was competent in supporting IR-induced RAD51 foci formation and olaparib resistance. We propose that different levels of DNA damage might underlie the difference between these phenotypes. 4Gy IR and 1-2 µM olaparib are expected to induce far more DNA damage compared to ZFN-induced DSB (three AAVS1 loci in triploid U2OS cells), triggering robust DDR, where the 7K-patch is naturally deacetylated on WT PALB2, hence little difference between WT and the 7R mutant. Under the condition in which a limited number of DSB is induced, i.e. modest DDR induction, 7K-patch-mediated chromatin association might remain important at least at an actively transcribed AAVS1 locus.

7. The model for the role of ChAM acetylation in regulating PALB2 function presented in Figure 7D is not fully supported by the data presented. Critically, while association with RAD51 and BRCA2 is tested in Figure S7B, the authors hypothesize that deacetylation is required to release PALB2 to enable association with BRCA1 but this is not tested utilizing the mutants.

As suggested, we have assessed BRCA1 association with PALB2 WT and 7R/7Q mutants by pull-down (Experiment 5; Figure 7—figure supplement 1D). Unexpectedly, this analysis revealed decreased binding of BRCA1 to either 7Q or 7R mutant, indicating that PALB2’s competence in associating with active genes (Figure 5F) affects its binding to BRCA1.

On the basis of this observation, we have revised our discussion on the mechanism by which PALB2 7Q and 7R variants could be recruited to sites of DNA damage (pages 28; lines 690-698). Briefly, PALB2 recruitment to sites of DNA damage might be enabled even with a reduced interaction with BRCA1, or without direct interaction with BRCA1, for example via a mechanism that is promoted by DNA-damage-associated small RNA (sdRNA) (Hatchi et al., 2021). Understanding the mechanism underlying this observation warrant future investigation and is now an active area of our investigation.

Also, there are some specific points that should be considered in the context of the model. This includes how DNA damage may trigger deacetylation, and whether it is the deacetylated state or the process of deacetylation of ChAM that is critical. Also, if acetylation is important for protecting active genes in the absence of DNA damage, is deacetylation necessary to release PALB2 local or global. This is important, because if it is local there needs to be a specific mechanism for local deacetylation, while if deacetylation is global that could result in transcriptionally active genes becoming unprotected.

We thank this reviewer for this valuable comment. We agree that, while this study establishes that ChAM is deacetylated upon DNA damage, it remains unclear whether the dynamic ‘de-acetylation’ of PALB2 is important for HR repair, and whether or not this is a local event. Regardless, we would like to highlight that PALB2-bound genes are mostly periodic, e.g. those required for cell cycle progression (Bleuyard et al., 2017, PNAS). It would therefore be reasonable to speculate that DNA damage triggers the suppression of periodic gene expression as a part of DNA damage checkpoint signalling, possibly in a KDAC-dependent manner, which then allows release of PALB2 without risking DNA damage that could otherwise be caused by replication-transcription conflict. Mobilised PALB2 might then be recruited to sites of DNA damage for HR repair. In the revised manuscript, we made our best effort to better describe our model, accompanied by the revised model shown in Figure 7D.

Further study will be required to fully evaluate this model, for example by identifying the specific KDAC involved in ChAM deacetylation and tracking individual PALB2 molecules, which we consider to be beyond the scope of the present study.

Minor Comments:a. Some parts of the Materials and methods are overly long (such as the subsection on "Protein purification" and "immunofluorescence microscopy") and could be shortened by consolidating experimental details that are largely the same for related processes.

According to the *eLife* manuscript guideline, we provide all Materials and methods in the main text.

b. In the description of Figure 1D, the statement "7K-patch, which is common to PALB2 orthologs" is misleading since there is not complete conservation of each lysine residue across each ortholog.

We apologise for this error. We have now amended the description on page 17, lines 404407: ‘*Notably, while sequence alignment of PALB2 orthologues showed some divergence in the Cterminal region of ChAM, an enrichment of lysine residues was consistently detected (Bleuyard et al., 2012) (Figure 1D).*’

c. Figures 3E,F and S3B,C perform FRAP in cells with knockdown of KAT2A/B as a surrogate for chromatin association. The authors note that this global reduction in acetylation increases PALB2 diffusion, but there is concern that this experiment is not very informative because the increased mobility may have nothing to do acetylation of PALB2.

Please refer to our answer to this reviewer’s point 4.

Reviewer #2 (Evidence, reproducibility and clarity (Required)):This manuscript reports the control of PALB2 – chromatin interaction by the acetylation of a particular lysine-rich domain of the protein called ChAM. This acetylation is shown to be mediated by the acetyltransferases KAT2A/B. Following these investigations, the authors made an effort to place their findings in the context of DNA replication and DNA repair.The proposed model is that the acetylation-dependent interaction of PALB2 with chromatin could ensure the protection of the genome during DNA replication and control DNA repair.Specific remarks1) Based on different experiments, essentially the one shown in Figure 3B, the authors conclude that the acetylation of the ChAM domain enhances its association with nucleosomes.However, taking into account the experimental setting, this conclusion should be largely tuned down. Indeed, this enhanced acetylation-dependent nucleosome binding was observed when the experiment was carried out in the presence of excess of free naked DNA.Under these conditions, the non-acetylated ChAM fragments became mostly trapped by DNA (clearly shown in Figure 3C/D), and hence would not be available for nucleosome binding, while the acetylated ChAM fragments would remain available for nucleosome association because of their reduced DNAbinding ability.Consequently, the acetylation of the ChAM domain would only play a role on the availability of PALB2 for chromatin/nucleosome binding and not directly stimulate nucleosome binding. Therefore, the nucleosome-binding capacity of ChAM by itself should not be dependent on ChAM domain acetylation.If true, this hypothesis could also be relevant in vivo since the poly-K in the ChAM domain could also non-specifically interact with nuclear RNAs and hence its acetylation, by releasing it from nuclear RNAs, would make it available for chromatin-binding.The importance of RNAs in the regulation of PALB2 nucleosome-binding could be tested in the experiments shown in Figure 2C and 2E by adding RNase to the pull-down medium (WT +/-RNase or addition of increasing exogenous RNAs).

We are grateful for the reviewer’s detailed comments and find the potential involvement of RNA very intriguing. Indeed, transcriptionally active loci, which are bound by PALB2, are enriched in nascent RNA, and such local RNA may play an important role in promoting the association of acetylated PALB2 with nucleosomes. However, we believe that investigating the role of RNAs in PALB2 nucleosome binding is beyond the scope of this study. As discussed extensively in response to this Reviewer’s point 2 below, we believe the mode of interaction of ChAM with nucleosomes to be highly complex, being jointly mediated by the N-terminal conserved region and the C-terminal lysine cluster. In the revised manuscript, we have discussed in greater details the model of ChAM interaction with nucleosomes (pages 26-27, lines 638-660), based on published and current results presented in this study.

2) The real question is as follows. While acetylation makes the protein available for nucleosome binding, which part of the ChAM domain is actually mediating nucleosome binding and whether lysine acetylation could be directly involved in this binding. Another question would be to identify the elements in the nucleosome mediating this interaction, histones (core domain, tails, posttranslational modifications, specific histone types), histone-DNA, etc…

We entirely agree with the reviewer’s question. Despite the increasing recognition of the physiological importance of the PALB2 ChAM and our efforts in understanding the mode of association of ChAM with nucleosomes (including the potential involvement of histone tail modifications), this specific question remains enigmatic.

Explicitly, our previous work demonstrated that substitutions of residues within the evolutionarily highly conserved N-terminal part of the ChAM perturb its association with nucleosomes (Bleuyard et al., 2017, PNAS; Bleuyard et al., 2017, Wellcome Open Research). A recent study by the laboratory of Prof Jackson proposed that basic residues across the ChAM are part of a binding interface with an acidic patch of histone H2A in its nucleosomal context (Belotserkovskaya et al., Nat Comm. 2020). Our results presented in this study introduced an additional complexity, showing that the C-terminal 7K basic patch is essential for ChAM-nucleosome interaction. Intriguingly, our study also suggests that the regions flanking ChAM, which are phosphorylated at multiple residues, play roles in regulating ChAM binding to nucleosomes (Figure 2B and C; please refer also to our answer to the reviewer’s minor point 6). The summary of these observations is shown in Figure 9—figure supplement 1.

We are currently working towards solving the structure of ChAM in complex with a nucleosome, which may help to clarify this very important question. At this point, we think a complete description of all the elements required for ChAM-nucleosome interaction is beyond the scope of this manuscript, and should be addressed in future work. In the revised manuscript, we have provided an updated overview of the ChAM elements affecting nucleosome interaction and post-translational modifications (acetylation, phosphorylation) flanking ChAM regions, as summarised in Figure R1, are now presented in Figure 7—figure supplement 2.

3) Taking into account the authors conclusions on the role of ChAM domain acetylation and its impact on PALB2 mobility, in Figure 4D/E, one should expect a difference of t_1/2_ when wild-type and 7R mutant are assayed by FRAP. At least the measures of t_1/2_ in the wild-type should have been more heterogeneous compared to the 7R mutant due to the acetylation of the wild-type PALB2 by the endogenous HATs (the impact of endogenous HATs on the wild-type sequence is shown in Figure 3F). Could the authors comment on this?

We appreciate this reviewer’s point, which is related to reviewer 1’s points 1 and 3, questioning why no difference between WT and the 7R mutant was detected by FRAP assay and cellular fractionation. To address this issue carefully, we have conducted further characterisation of PALB2 K to R variants using DNA binding assays and ChIP-qPCR (Experiments 1 and 2).

As outlined in our response to reviewer 1’s points 1 and 3, we found that K to R substitutions reduced ChAM binding to naked DNA in vitro (Experiment 1; Figure 2 H) and PALB2 association with three target genes *ACTB*, *TCOF1* and *WEE1* in vivo (Experiment 2; Figure 5 F). Overall, these analyses revealed that K to R substitutions cause reduced DNA and chromatin association at defined loci in a manner independent of electrostatic charges. The discrepancy between these new results (DNA binding and ChIP-qPCR) and previously presented results (FRAP assay and cellular fractionation analysis) can be explained by the fact that PALB2 associates with only a small subset of genes (Bleuyard et al., 2007, PNAS); FRAP assays and cellular fractionation analyses were most likely not sensitive enough to detect minute but critical differences. This point is now extensively discussed in the revised manuscript p19-20, lines 487-506.

4) It would be better to remove the data presented in Figure 5 since, as currently presented, these investigations remain shallow and do not bring much information on what is happening. The presented data are rather confusing since, in the absence of further investigations, it is not clear which one(s) of the mechanisms involved in the control of DNA replication is controlled by PALB2 and many explanations, including artefacts, remain possible.The manuscript would gain in interest if the authors would devote the functional studies only to the repair part (Figure 6 and 7).

Our ChIP-qPCR analyses revealed significantly decreased binding of PALB2 7R and 7Q compared to WT at previously defined PALB2-bound loci (Figure 5 F). This observation now nicely explains why both PALB2 7R- and 7Q-expressing cells displayed delays in cell cycle progression and accumulated DNA damage over time, likely due to a lack of protection of PALB2-bound genes during replication in both these cell lines. We therefore feel it is important to keep these results in the manuscript to allow readers to comprehend the role of the 7K-patch in both undamaged and damaged conditions.

Minor points5) High background of non-enzymatic acetylation of PALB2 fragments makes the identification of KAT2A/B specific acetylation not very convincing.The immunoblot detection of acetylation fragments shown in Figure S1 is much more convincing. Therefore, the authors may consider to present Figure S1 as a main Figure and Figure 1B as a supplementary one.

As suggested by the reviewer, panels Figure 1—figure supplement 1B and Figure 1B have now been swapped in the revised manuscript.

6) It would be interesting if the authors would comment on why the presence of regions flanking the ChAM domain (Figure 1A, construct #5) significantly reduces chromatin (Figure 1B) and nucleosome binding (Figure 1C).

We are grateful for this reviewer’s comment. Indeed, we noticed that the inclusion of the ChAM C-terminal flanking region perturbs its chromatin association. This region is highly enriched in serine and threonine residues which could be targeted for phosphorylation by cell cycle regulators (CDKs and PLK1) and DNA damage-responsive kinases (ATM and ATR). It is tempting to speculate that, when phosphorylated, this flanking region could mask the basic patch of the ChAM, hence facilitating the release of PALB2 from undamaged chromatin region and its recruitment to sites of DNA damage. In the revised manuscript, we provide the complete list of PTMs in Figure 7—figure supplement 2 and discuss this point on page 26-27, lines 638-660.

Reviewer #3 (Evidence, reproducibility and clarity (Required)):KAT2-mediated acetylation switches the mode of PALB2 chromatin association to safeguard genome integrityThe authors describe a series of experiments examining the consequence of acetylation, within a defined motif (Chromatin Association Motif; ChAM), on the cellular roles of the protein PALB2 (Partner and Localizer of BRCA2).The key conclusions drawn by the authors are generally convincing and are supported by the presented experimental results, which indicate that acetylation of PALB2 by KAT2A/KAT2B modulates its cellular behaviour and response to DNA damage. However please see specific comments below:Major CommentsExpression of full-length PALB2 in the heterologous host *E. coli* is highly problematic, as the WD40 domain is generally not correctly folded. The authors use the ArticExpress strain to try and solve/alleviate this problem – but it is clear from the Materials and methods section that an ATP-wash step has had to be introduced in order to release the recombinant protein from the chaperone system encoded by the ArticExpress system; i.e. indicating poor / mis-folding. Whilst this does not strictly have an effect on the results presented in Figure 1 (detection of in vitro acetylation sites), they have implications for the wider scientific community, as this may lead to the erroneous assumption that is possible to produce functional / folded full-length PALB2 in this way.

We apologise if the manuscript conveyed the message that we are able to produce functionally active, full-length PALB2 in bacteria, which was clearly not our intention. Our aim was to test whether KAT2A was able to acetylate PALB2 in vitro. We agree that the folding and the biochemical properties (e.g. WD40-mediated BRCA2 binding) of the bacterially produced full length PALB2 were not fully assessed. We believe that this does not affect the overall conclusions of this study. In the revised manuscript, the error has now been clarified on page 9, line 219 in Material and Methods.

In vitro modification assays are prone to producing post-translational modifications that are not fully reflective of those observed in vivo, and therefore need to be treated with some caution. This is highlighted by the relatively low modification of K438 in vitro by KAT2A; esp. as this is an acetylation site that has been previously mapped in vivo (by the authors). It would have been useful to include / see the effects on PALB2 function in vivo by modification / alteration of this single site.

We appreciate the reviewer’s comment. Redundancy of acetylation acceptor residues within a lysine cluster is common, as is also the case for many ubiquitination events, hence we analysed the 7K-patch mutant for phenotypic studies. For the same reason, we trust that the outcome of the characterisation of a K438 mutant would not significantly change our conclusions.

Figure 3C and Figure 3D do not fully support or reflect the conclusions drawn by the authors – any peptide containing a cluster of positive charged residues are likely to interact with DNA through charge neutralisation of the phosphodiester backbone, concomitantly any alteration to this region of charge (i.e. via acetylation) will perturb this interaction.

We totally agree with the reviewer’s view. In the revised main text referring to the results shown in Figure 3C and D, we state “As anticipated, lysine acetylation, which neutralises the positive charge on the lysine side chain, conferred reduced affinity for negatively charged DNA” (page 20, lines 475-477).

Furthermore, experiments performed with the synthetic acetylated peptides do not agree with those carried out with the GST-ChAM constructs – GST-ChAM interacts with the nicked and linear forms of the pBS plasmid (Figure 2F) but does not interact with the supercoiled form. The WT synthetic ChAM peptide, in contrast, interacts with all three plasmid states at high concentrations. It is suggested that these two figures are removed.

It is true that we cannot exclude potential differences between GST-ChAM and synthetic ChAM peptides: for example, the 26 kDa GST, which can form a dimer, might partly affect the biochemical properties of ChAM in DNA binding. However, we believe that the differences are more likely caused by the concentration of ChAM used. While we used the synthetic ChAM peptides at concentrations of 2.97, 5.94, 29.3 µM for Figure 3C, we used 5.94 µM of GST-ChAM for Figure 2F; we apologise for the omission of the exact experimental conditions used. This notion is supported by the side-by-side experiment shown id Author response image 1. In the revised manuscript, we have included the concentration of GST-ChAM used in Figure 2F (page 34, line 853) to be clear.

**Author response image 1. sa2fig1:** ChAM binding to DNA is dose Dependent. Indicated concentration of recombinant GST-ChAM and synthetic ChAM were incubated with 300 ng of pBluescript, and DNA-binding was assessed by EMSA. Binding to DNA was not detectable with up to 300 nM ChAM in our previous study (Bleuyard et al., 2012, EMBO Rep).

p. 18: the authors used a PALB2 variant, where the lysines in the 7K patch are mutated to arginine – but don't fully characterise the effects of introducing these particular mutations on the ability of the ChAM fragment to bind to DNA, or indeed to nucleosomes; this is an important control.

We appreciate the reviewer’s comment and, indeed, the importance of the 7R variant biochemical characterisation for accurate interpretation of in vivo phenotypes. We have now conducted the biochemical characterisation of a ChAM K to R mutant (Experiment 1; Figure 2G and H). We have generated a ChAM mutantwith K to R substitutions at positions K436, K437 and K438 (the ChAM 3R4K variant) and purified it from *E. coli* (Figure 2G). Our analyses of the ChAM variants revealed reduced DNA binding by ChAM 3R4K compared to wild-type (Figure 2H), albeit better binding than 3Q4K. This observation was unexpected as K to R substitutions are expected to maintain electrostatic charges.

These observations led us to revise our interpretation of the 7R variant; K to R substitution impairs the function of ChAM in associating defined transcriptionally active loci (Figure 5F), although maintains better overall chromatin association compared to the K to Q variant (Figure 4). This explains why both K to Q/R substitutions confer defects in some cellular phenotypes, such as cell growth in undamaged conditions (Figure 5A-E) and homology mediated gene-targeting efficiency (Figure 7—figure supplement 1B). This point is extensively discussed in the revised manuscript pages 21-23, lines 503-560, under the section ‘*PALB2 ChAM 7K-patch acetylation is required for the protection of actively transcribed genes during DNA replication’*

Please also refer to our response to reviewer 1’s points 1 and 3 and reviewer 2’s point 3.

Figure 6: it would be good to show a second supporting example for deacetylation of PALB2 in response to DNA damage – perhaps treatment with MMC?

We appreciate the reviewer’s comment. Indeed, we have conducted the analysis upon MMC and olaparib exposure. Curiously, however, no clear change in ChAM acetylation was detectable (Author response image 2). Note that, for this experiment, we assessed the acetylation level of a GFP-fusion of ChAM, exogenously expressed in HEK293T cells, along with endogenous γ-H2AX as a readout for DNA damage signalling. Unlike ionising radiation, which triggered strong induction of gammaH2AX (Figure 6), no clear increase of γ-H2AX was detectable upon the MMC/olaparib exposure conditions used. Hence, we propose that the reduction of ChAM acetylation reflects the cellular response to DNA damage. These points have now been clarified in the revised manuscript (pages 23, lines 568-576).

**Author response image 2. sa2fig2:** ChAM acetylation upon DNA damage treatment. HEK293T cells transiently expressing GFP-tagged ChAM were treated with (A) 2.5μM olaparib and collected after 0, 15, 30, 60 or 120min of treatment and (B) 1mM mitomycin C (MMC) and collected after 0, 15, 30, 60, 120 or 240min of treatment. In both conditions, affinity-purified GFP-ChAM acetylation was assessed using anti-GFP and anti-acetyl-lysine (pan-AcK) antibodies. γ-H2AX signal in whole cell lysate (input) was detected to monitor DNA damage.

Minor Commentsp. 16: 'Our MS analysis of the chromatin-associated GFP-ChAM fragment identified actelyation of all seven lysines within the 7K-patch (Figure 3A, marked with arrows).This part of the manuscript is potentially a little confusing, as Figure 3A references a series of synthetic peptides rather than the GFP-ChAM fragments themselves.

We apologise for the confusion that we have now corrected. Indeed, Figure 3A shows (1) MS of the chromatin-associated fraction of GFP-ChAM (the top part with arrows) and (2) a schematic diagram of synthetic peptides that we used for biochemical analyses (the bottom part). Figure 3A has been modified accordingly.

p. 20: Furthermore, using the FRAP approach, we observed clear differences in diffusion rates of FEPALB2 following damage by IR, MMC, or olaparib treatment…FE-PALB2 = FL-PALB2?

We apologise for the confusion. In our study, FE-PALB2 refers to Flag-EGFP tagged PALB2 (full-length). This is defined in the text “To this end, a tandem FLAG- and EGFP-tagged full-length wildtype (WT) PALB2 (FE-PALB2)” (page 20, lines 486-487).

[Editors' note: further revisions were suggested prior to acceptance, as described below.]

The manuscript has been improved but there are some remaining issues that need to be addressed, as outlined below:The efforts in this revision were appreciated by all reviewers. However, as detailed in reviews #1 and 3, there are significant concerns about the overall model and specific experiments, which will require significant additional experimentation.The key issues are:1) A fuller characterization of the 7R mutant is required in the HR-reporter assay, the sensitivity assays, in the nucleosome binding assay, and in ChIP assays at transcriptionally-active loci along with some controls. The importance of the analysis of the 7R mutant was highlighted in the previous review cycle.

We have thoroughly reconducted the requested experiments to the best of our ability. These re-evaluations have revealed more detailed properties of the PALB2 7Q and 7R mutants, leading to some key changes in our interpretation and the model.

In brief, despite our greatest efforts, we were unable to obtain concrete evidence supporting the involvement of the PALB2 7K patch in HR repair using HR reporter assays (i.e. DSB-induced gene targeting at *AAVS1* or *LMNA* loci). Also, the olaparib sensitivity of the 7Q variant proved to be insignificant. However, it was clear that S phase RAD51 foci were less efficiently formed in both the 7Q and 7R mutants upon irradiation (Figure 7B and E). Significantly, we also found impaired RAD51 foci formation in the 7Q and 7R mutants even in unperturbed conditions (Figure 8B and E). This is followed by a surge of S phase γ-H2A.X foci (Figure 8F) and impaired cell survival (Figure 9B). These observations suggest that these mutants are defective in RAD51 foci formation at randomly distributed DNA damage, stochastically occurring as part of normal physiology (e.g. metabolically generated ROS), resulting in the accumulation of unrepaired DNA and consequential cell death.

Interestingly, PALB2-7R, but not PALB2-7Q, was less efficiently recruited to DNA damage induced by micro-irradiation (Figure 6), revealing distinct mechanisms underlying the impaired RAD51 foci formation with these mutants. Specifically, the PALB2-7R mutant fails to be recruited to these sites effectively, but PALB2-7Q, while being successfully recruited, is defective in promoting RAD51 foci formation.

Biochemically, recombinant ChAM 7R, but not ChAM 7Q, binds purified nucleosomes, albeit to a lesser extent than WT (7K) (Figure 2D and E). This finding corroborates our observations that the PALB2 7R largely remains in the chromatin-enriched fraction (Figure 5). AlphaFold2 modelling further predicts greater structural plasticity of ChAM WT (7K) compared to ChAM 7Q or ChAM 7R (Figure 2—figure supplement 2), indicating that ChAM WT has better capacity to explore various structural configurations for its optimum association with nucleosomes compared to the 7R and 7Q counterparts. Indeed, both 7R and 7Q binding to active genes are severely compromised compared to WT (Figure 9D). IR triggered robust release of WT PALB2 from these loci, concomitantly with the ChAM deacetylation.

2) The authors should acknowledge inconsistencies and revise their model to be compatible with their data.

Taking all experimental results together, we propose that the ChAM 7K-patch regulates dual modes of PALB2 chromatin association, namely (1) the steady-state PALB2 association with actively transcribed chromatin through the acetylatable 7K, and (2) the damage-induced PALB2 engagement with damaged chromatin through the non-acetylated 7K. Specifically, in unperturbed conditions, WT PALB2 resides at actively transcribed chromatin through MRG15 (Bleuyard et al., PNAS, 2017) and KAT2A-mediated acetylation of the 7K-patch (this study), which is important for protecting active genes from replicative stress (Bleuyard et al., PNAS, 2017). Spontaneous DNA damage, IR and potentially other genotoxic stresses trigger the deacetylation of the 7K-patch, enabling PALB2 mobilisation and its re-localisation to DNA damage sites, including those distant from PALB2-bound actively transcribed regions. At sites of DNA damage, the non-acetylated 7K-patch further supports efficient RAD51 loading onto ssDNA, likely involving its non-specific electrostatic interaction with exposed nucleosomes (Belotserkovskaya et al., Nat Comms, 2020). In this way, WT PALB2, but not the 7Q or 7R mutant, prevents the accumulation of DNA damage and facilitates survival.

The absence of phenotypes with the HR reporter system and upon olaparib exposure was unexpected. It is noteworthy, however, that the HR reporter systems monitor DNA damage repair within active genes (i.e., AAVS and LMNA) and that PARP1 non-randomly occupies regulatory regions of active genes (Krishnakumar, R., Science, 2008; Krishnakumar and Kraus, Mol Cell, 2010). Hence, it is likely these systems highlight HR events only at actively transcribed genes and nearby loci. We suggest that the regulation of the 7K-patch has more pronounced impact on DNA damage in regions that are distant from transcriptionally active genes. Regardless, given the uncertainty of these observations, we have removed these results from the revised manuscript.

The reviews contain specific critiques and concerns that will require significant changes in the text. The authors may want to consider removing the DNA binding assays.

As suggested, we have removed the DNA binding assays, incorporated a significant amount of new experimental data and also considerably changed the main text according to the results. We thank the reviewers for constructive criticisms, which have improved the study and made it more robust.

Reviewer #1 (Recommendations for the authors):In this revised manuscript by Fournier et al. (eLife-57736R1), the authors have added experiments and analysis to address concerns raised previously. This reviewer appreciates the elegant systems employed in much of this work, as well as the effort that went into this revision. The revised manuscript continues to offer potential insights into the regulation and function of PALB2 in DNA repair, and into the role of acetylation in regulating DNA damage responses. However, there are still major concerns, which in the opinion of this reviewer, should be addressed prior to potential publication. These include issues with certain newly added figures, and apparent inconsistencies or contradictions between different results. As a consequence, the model presented in Figure 7D is not yet fully supported.1. The 2nd paragraph on p. 21 of the unmarked revision discusses levels of γ-H2AX shown in Figures 5D-E in the context of DNA replication (this paragraph is part of a subsection of the Results beginning on p. 20). Earlier in this same paragraph the authors hypothesize that "PALB2 association with active genes … prevents the induction of DNA damage during DNA replication". In this context, Figures 5D-E are misleading and are not necessarily related to DNA replication. Specifically, the cells were not synchronized and DNA damage arises over several days, leading to questions about the relationship to DNA replication. This would be better supported by performing ChIP assays using γ-H2AX antibody to determine whether DNA damage specifically accumulates at transcriptionally active sites, where conflicts with DNA replication may arise, as compared to other sites. Further, no measures of statistical significance are given for Figure 5E.

We appreciate this reviewer’s comment and assessed the level of γ-H2A.X in S-phase cells, marked by EdU incorporation (Figures8 and 9). This approach is widely used to robustly assess replication-associated DNA damage. This assessment confirmed that the 7Q and 7R mutants indeed accumulate more γ-H2A.X in S phase cells upon shRNA-mediated depletion of endogenous PALB2; this difference reached statistical significance at day 5 after the expression of shRNA. This finding is reflected by the reduced survival of cells expressing the 7Q or 7R mutant even in the absence of genotoxic treatments.

Unfortunately, γ-H2A.X ChIP-qPCR proved challenging with the currently available antibody and we were unable to validate whether these S-phase γ-H2A.X foci indeed arise at transcriptionally active sites in 7R/Q cells. Nonetheless, we consider this very unlikely as, in our previous study, we observed no increase of γ-H2A.X at these loci in cells expressing another PALB2 variant which is similarly unable to bind active genes (via MRG15 interaction), unless challenged by CPT, which induces covalent TOP1-DNA binding (Bleuyard et al., PNAS, 2017).

Significantly, while characterising the cellular phenotypes, we noticed impaired S-phase RAD51 foci formation even in unperturbed 7R/Q cells on day 4, prior to the rise of γ-H2A.X in these cell lines at day 5 (Figure 8). A similar trend was found in 7R/Q cells exposed to IR on day 4 (Figure 7). Given that IR stochastically induces various types of DNA damage across the genome, which could, in turn, impede DNA replication, these observations suggest that the 7Q/R mutants are both defective in the efficient recruitment of RAD51 to DNA damage at random locations of the genome, conferring the global accumulation of DNA damage.

We further revealed that the PALB2 7R, but not the 7Q counterpart, is less efficiently recruited to laserinduced DNA damage (Figure 6), suggesting that the mechanism underlying RAD51 recruitment to sites of DNA damage is different between these mutants. We propose that defective engagement of the 7Q mutant with damaged chromatin undermines stable RAD51 foci formation, while the 7R mutant, which is ‘glued’ to chromatin non-specifically through electrostatic interaction, failed to be mobilised upon DNA damage, conferring impaired recruitment of RAD51 to sites of DNA damage.

2. While HR data was added at the request of a reviewer, the intent is presumably to strengthen the hypothesis that ChAM acetylation regulates PALB2 function in DNA repair. Four points about this:A) No measure of significance is given for the HR results shown in Figure 7—figure supplement 1. Further, the error bars are often very large. As such, no convincing conclusions can be made about the effects of the 7Q and 7R mutants on HR.

Despite our greatest effort, we were unable to detect significant HR defects using our HR reporter systems. Specifically, we exploited two reporter assays to evaluate HR competency, by inducing DSB at *AAVS1* by ZFN or at *LMNA* by Cas9, and providing a matching donor plasmid carrying GFP with homologous arms. Successful integration was monitored through the expression of GFP, quantified by FACS or microscopy, and was used as the readout of HR proficiency. However, neither of these assays has provided consistent results in relation to the function of the 7K-patch in HR repair. We highlight, however, several limitations of these HR reporter assays. Firstly, these assays depend on high transfection efficiency and only detect integration of a reporter plasmid at defined, actively transcribed loci (i.e., *AAVS* and *LMNA*), which might not be representative of all types of HR across the genome. Also, unlike physiological HR repair which uses an unbroken sister chromatid as a repair template, these assays use exogenously introduced plasmid DNA as an HR template. Finally, these assays introduce ‘clean’ two-ended DSB, which is likely processed differently from those naturally occurring ‘dirty’ DNA breaks. Accordingly, we have decided to remove these gene-targeting results.

In the revised manuscript, we have conducted thorough analyses of RAD51 foci (Figures7 and 8), which usually reflects the HR competency. Indeed, RAD51 foci formation is widely used as a reliable readout of HR proficiency in the field, (as the reviewer also comments in D) in this section below (‘proficiency for RAD51 foci in response to IR typically is associated with functional HR’). Accordingly, we infer HR proficiency on the basis of RAD51 foci formation.

B) In any case, if the finding was sufficiently established that both the 7Q and 7R mutants disrupt DSB-HR, as suggested by the authors, how do they rectify this with the observation that only 7Q but not 7R sensitizes U2OS cells to Olaparib in Figure 7C? Does the 7R mutant perturb HR with no consequences for cellular survival in the presence of DNA damage? And, if so, what additional function is perturbed by the 7Q mutant that results in sensitivity to Olaparib?

Through our careful re-evaluation, it has become evident that all our cell lines show comparable resistance to olaparib with similar levels of γ-H2A.X and RAD51 foci. While the source of PARP-trapping lesions remains contentious, PARP1 is shown to be intimately linked with transactivation (Krishnakumar, R. Science, 2008; Krishnakumar and Kraus, Mol Cell, 2010). Hence, it is tempting to speculate that olaparib induces more lesions at transcriptionally active chromatin. The lack of phenotypes in the PALB2 7K patch mutant upon olaparib exposure could be explained by a model in which PARP-trapped lesions at actively transcribed chromatin can be effectively repaired without significant involvement of the PALB2 7K patch or, more broadly, without the steady-state presence of PALB2 at these regions. Indeed, transcription itself has been shown to enhance HR efficiency (Ouyang et al., Nature, 2021).

Accordingly, in the revised manuscript, we shift our focus to the cellular phenotypes in unperturbed conditions and IR response, where statistically significant impairment of S-phase RAD51 foci was observed in both the 7Q and 7R mutant cells (see our response to point 1 above, also outlining our revised model that explains the molecular mechanism underlying these phenotypes).

C) In the last paragraph on p. 26, the authors write, "We suggest that damage-induced ChAM deacetylation allows its recruitment to sites of DNA damage through its interaction with damage sensing factors, fulfilling its function in promoting HR repair". If all of this is true, since PALB2-7R was never acetylated and therefore did not need to be deacetylated or released from sites of active transcription, then, according to their model, why is this mutant defective for HR (Figure 7—figure supplement 1)? Is this finding consistent with the model presented in Figure 7D?

We apologise for the confusion. In the revised manuscript, we provide additional results showing that ChAM-7R binds to nucleosomes even without acetylation (Figure 2D and E), although its association with transcriptionally active loci in vivo is drastically compromised (Figure 9D). Further, the recruitment of PALB2-7R to laser-induced DNA damage was significantly compromised (Figure 6). Collectively, we propose that PALB2 7R is randomly ‘glued’ to chromatin via electrostatic interaction, making this variant less efficiently mobilised and hence less efficiently recruited to sites of DNA damage.

D) On p. 27 (Discussion), the authors state, "Interestingly, the PALB2 7R variant, unlike its 7Q counterpart, had little impact on RAD51 foci formation … This could be explained by our findings that DNA damage triggers ChAM deacetylation and promotes PALB2 mobilisation, which we propose is the critical event in initiating HR repair". This statement seems to overlook data shown in Figure 7—figure supplement 1 that claims PALB2-7R appears is deficient for HR. Further, proficiency for RAD51 foci in response to IR typically is associated with functional HR, but 7R has nearly normal RAD51 foci while suggested to be deficient for HR. Can the authors' explain this based upon their model?

As stated above, we reassessed IR-induced RAD51 foci formation in S phase, when HR is considered most active (Figure 7B and E). This assessment revealed that both PALB2-7Q and -7R are defective in efficient RAD51 foci formation, indicating impaired HR proficiency. We revised the manuscript accordingly.

3. In the model presented in Figure 7D, the authors suggest (1st paragraph of the Discussion, on p. 24) that DNA damage triggers ChAM deacetylation (presumably on PALB2 localized to transcriptionally active chromatin) which "releases PALB2 from active genes and increases its mobility". I have the following concerns about the model (in addition to those detailed in point #2):A) When discussing their model, it is currently unclear whether the authors are proposing that ChAM deactylation is occurring specifically at "transcriptionally active chromatin" in response to DNA damage and whether this makes PALB2 "globally" available for HR.

These are correct – we thank the reviewer for rewording. In support of this idea, our new ChIP-qPCR assessing PALB2 occupancy at transcriptionally active loci revealed that it is indeed reduced upon IR (Figure 9D). We made this clear in Descussion in the revised manuscript (page 29).

B) The data shown in Figures 4B-C seems to contradict the authors' model (Figure 7D). If the role of ChAM deacetylation is simply to release "PALB2 from active genes and increase(s) its mobility", then the 7Q mutant (mimic of acetylated lysine), by being more mobile (Figure 7D-E), should be available for "HR repair complexes" that form in chromatin. Because the 7Q mutant shows less association with chromatin than WT PALB2 or the 7R mutant (Figures 7B-C), one wonders if the 7Q mutant impairs some other PALB2 function besides release from transcriptionally active chromatin upon DNA damage. Additionally, the 7R (non-acetylable) mutant appears functional as indicated by normal cellular resistance to Olaparib (Figure 7C). This does not support any claim there is a specific need that PALB2 be recruited to transcriptionally active chromatin and subsequently released.

The reviewer is absolutely correct, except that we consider that the 7Q is functionally null in direct chromatin association rather than mimicking acetylated lysine. Indeed, our new experimental results indicate that, while PALB2-7Q can be recruited to sites of laser-induced DNA damage (Figure 6), it fails to promote RAD51 foci formation (Figure 7B and E). It was recently shown that PALB2, through its basic residues within the ChAM, can bind to the acidic surface of nucleosomes at sites of DNA damage (Belotserkovskaya et al., Nat Comms, 2020). We speculate that this engagement allows optimal loading of RAD51 onto ssDNA for HR repair. Our results also support the notion that PALB2 7R is defective in HR repair, albeit unlikely at actively transcribed genes (see our discussion on the HR reporter and sensitivity to olaparib above).

The need for steady-state PALB2 association at transcribed genes was fully evaluated in our previous publication (Bleuyard et al., PNAS, 2017). When PALB2 association at active genes is compromised, these regions accumulate DNA damage when exposed to the DNA topoisomerase I poison camptothecin (CPT). It remains unclear why PALB2 needs to be released upon DNA damage, but the copy number of PALB2 is estimated at 10 times lower than that of BRCA2 (Kulak et al., Nat Methods, 2014), and we speculate that random and static PALB2 association with chromatin, as seen with the PALB2 7R mutant, is toxic to the cells.

C) Unless I overlooked it, it seems that the authors did not specifically test the effect of the 7R mutant on the association of PALB2 (or the ChAM domain) with nucleosomes; this may be helpful in better interpreting various results in the context of their model.

We included the biochemical assessment of the 7R ChAM binding to nucleosomes in the revised manuscript (Figure 2D and E). The purified recombinant ChAM 7R, along with WT and the 7Q, was assessed for its binding to nucleosomes in vitro. Our results show that ChAM 7R indeed maintains nucleosome binding, albeit at a slightly reduced level compared to the non-acetylated WT. Conversely, the 7Q counterpart largely lost its binding to nucleosomes.

D) Since the 7R mutant shows even less association with transcriptionally active chromatin (Figure 5F) than PALB2-7Q, and PALB2-7R is resistant to Olaparib (Figure 7C), then it would seem that protection of "transcriptionally active chromatin … during DNA replication (2nd paragraph of p. 24) is not essential for cell survival in response to DNA damage.

We appreciate this critique. As discussed above, we have previously shown that PALB2 association at active genes is important to protect these regions from transcription-replication conflict, which is enhanced by CPT (Bleuyard et al., PNAS, 2017). CPT covalently links TOPI to DNA, which is significantly different from noncovalent PARP-trapping by olaparib (Murai et al., Cancer Res, 2012; Langelier et al., Science, 2012). Unlike TOPI-trapped lesions, the absence of PALB2 at active genes appears to cope well with PARP-trapped lesions, likely due to its more dynamic nature of trapping. Nonetheless, we removed data associated with olaparib treatment to make this manuscript better focused on thier response to randomly induced DNA damage.

E) The model that acetylation of the 7K patch releases PALB2 from transcriptionally active chromatin would be better supported by testing this in the presence of IR (or Olaparib), since DSB-initiated HR and cellular resistance to Olaparib are among DNA damage response-related assays featured in the manuscript. Does IR or Olaparib indeed release WT PALB2 from transcriptionally-active loci?

We have reconducted PALB2 ChIP in unperturbed conditions and also in IR-treated cells (Figure 9D). We observe clear enrichment of WT PALB2 at *ACTB*, *TCOF* and *WEE1* loci in unperturbed cells, but this association is greatly reduced in IR treated cells. This observation is consistent with our FRAP results showing increased PALB2 mobility in IR-treated cells (Figure 4D and E). Both the 7Q and 7Q mutants showed reduced association with these loci in unperturbed conditions, and little change of their association upon IR-treatment.

F) According to the model presented, are transcriptionally-active loci left deprotected after acetylation of the ChAM motif that results in increased mobilization of PALB2 for HR?

It is widely known that, upon DNA damage, cells suppress overall progression of DNA replication via activation of checkpoints. Hence, we consider that these regions have a reduced risk of generating DNA damage arising from the transcription-replication conflict (Bleuyard et al., PNAS, 2017) while PALB2 acts at damaged DNA for repair.

G) In the absence of assays of PALB2 with transcriptionally active loci after exposure to IR (or PARP inhibitor), the authors have not convincingly demonstrated that PALB2 is mobilized from these loci after DNA damage. While the authors note that Bleuyard et al. 2017b demonstrate that CPT induces decreased association of PALB2 with transcriptionally active chromatin, CPT could have a different effect than IR by inducing more replication stress.

As indicated under (E) above, we now include PALB2 ChIP at transcriptionally active loci in IR-treated cells (Figure 9D). It is indeed efficiently released from these loci upon IR.

4. The authors note that both K to R and K to Q substitutions alter the size of the side chain, albeit to different degrees. How confident are the authors that K to R and K to Q substitutions are accurate non-acetyl and acetyl mimics, respectively? Is it possible that both substitutions are simply disrupting the structure of the ChAM, and perhaps PALB2 more globally, to some degree?

This is a matter of our active investigation, characterising experimentally the structure of ChAM in the complex with a nucleosome. Nonetheless, in this manuscript, we included AlphaFold2 modelling of ChAM WT (7K), 7Q and 7R mutants (Figure 2—figure supplement 2). It predicts that WT (7K) forms various structural configurations, while 7Q and 7R mutants form consistent structural variations. From this modelling, we infer that WT ChAM has a great flexibility, but that the 7Q and 7Q mutations confer rigidity. This might potentially restrict the way the 7Q and 7R mutants bind nucleosomes, preventing optimum interaction.

5. In contrast to association of PALB2 with transcriptionally active chromatin regulating interaction with BRCA1, as proposed, structural changes across the ChAM, or in PALB2 more broadly, could potentially cause the observed decreased interactions of the 7R and 7Q PALB2 mutants with BRCA1.

As commented above (point 4), we now include AlphaFold2 modelling of ChAM WT (7K), 7Q and 7R mutants (Figure 2—figure supplement 2). The region linking the ChAM and the N-terminal BRCA1 binding domain is largely disordered, and the ChAM mutation is unlikely to affect BRCA1 binding. Indeed, our reassessment of PALB2BRCA1 interaction revealed no impact of the mutations in BRCA1 binding.

Reviewer #3 (Recommendations for the authors):Fournier et al. – KAT2-mediated acetylation switches the mode of PALB2 chromatin association to safeguard genome integrityThe authors have responded to the comments made by each reviewer, and have made several amendments to the manuscript itself, generally improving its legibility and providing additional supporting data / experiments. Saying this, there are still some outstanding questions that should be addressed:A) It is still not clear as to the (intended) purpose of the GST-ChAM DNA-binding experiments and what these actually contribute to the manuscript as a whole. They could be readily removed from the manuscript without impacting the set of conclusions.

As recommended, we removed DNA binding assays from the current manuscript. As such, we conducted no further investigation to address the reviewer’s comments below.

Note: there is no appreciable DNA binding by any of the mutant forms of the GST-ChAM construct as compared to the two controls, WT or indeed PALB2-FL (how was this protein produced, is this the bacterially expressed protein?).It is therefore not clear how the authors can support the following statement.Page 18, Lines 444-446: "… however, the ChAM 3R4K variant also showed impaired binding to DNA in an EMSA assay, albeit less pronounced than that of its K to Q (3Q4K) counterpart (Figure 2H)".B) Whilst it is appreciated that the authors have now created two additional GST-ChAM construct (3Q4K and 3R4K) they haven't performed the requested control experiment, i.e., with the 7R construct. This is necessary to support the set of experiments described at a later point in the manuscript. What is the effect of the 7R construct in the GST-pulldown experiments – does this still bind to nucleosomes / histone H3?

We now include 7R ChAM in the nucleosome binding assay (see also our response to reviewer 1, point 3, C), presented in Figure 2D and E. Our results show that ChAM 7R indeed maintained nucleosome binding, albeit at a slightly reduced level compared to the non-acetylated WT, while the 7Q mutant largely lost its binding to nucleosomes.

C) Figure 3F, why is there a such a significant increase in PALB2 mobility in the FRAP experiment for the negative control siCntrl? See also Figure 3—figure supplement 1

We speculate that siRNA treatment itself confers stress to the cells. Indeed, we consider siCntrl is a better control for siKAT2A/B and, as such, removed the non-treated data from the plot (Figure 3D)

D) The data for MMC/Olaparib treatment presented in the response to authors (Figure R3) should be included as supplementary data within the revised manuscript.Page 23, Lines 572-576"… though similar changes in ChAM acetylation were not detectable upon MMC or olaparib DNA damaging treatment. We noted that IR triggered strong γ- H2A.X induction (Figure 6B), which was not observed upon exposure of HEK293T cells to MMC or olaparib, suggesting that the reduction of ChAM acetylation reflects the cellular response to DNA damage.

In the revised manuscript, we focused on the cellular response to IR. As such, we removed the data associated with MMC and olaparib.

E) Figure 7 —figure supplement 1, panel B (HR assay)The means from individual experiments should be displayed on the graph, as it is not clear if the wide error bar (which presumably should also extend downwards, as + and – 1 SD) and thus interpretation of the result is due to a single outlier measurement.

We repeated the HR reporter assay, targeting *AAVS* and *LMNA* loci. As extensively explained in our response to reviewer 1, point 2, (A), we were unable to gain consistent results using these systems, but would also like to highlight several limitations of the HR reporter. As such, we removed the result from the revised manuscript.

In its place, as reviewer 1 also comments in point 2 (D) (i.e. ‘*proficiency for RAD51 foci in response to IR typically is associated with functional HR’)*, RAD51 foci formation was used as a reliable readout of HR proficiency in this study.

(F) DiscussionThe authors allude to unpublished data within the discussion / conclusion – such statements should ideally be removed.

As suggested, we removed the statements below.

“Also, our preliminary results indicate that a PALB2 variant defective in BRCA1 binding exhibits higher accumulation at genic regions than its wild-type counterpart.”

“Indeed, our preliminary results demonstrated that extracellular glucose concentration affect PALB2 dynamics, as reducing glucose level in the growth medium increases PALB2 dynamics.”

[Editors' note: further revisions were suggested prior to acceptance, as described below.]

The manuscript has been significantly improved but there are some remaining issues that need to be addressed, as outlined below:In the current revision of their manuscript, Fournier et al. have, in general, addressed the concerns that were raised previously. This includes removal of confounding results and addition of new data, such as survival assays, recruitment to sites of DNA damage, and ChIP assays for occupancy at transcriptionally active sites with and without IR, for WT PALB2, and the 7Q and 7R mutants. Overall, the data and presentation are now much more cohesive and are integrated into an interesting model in Figure 9E. As in previous versions, an impressive array of complementary techniques have been employed in this manuscript. This manuscript will make an important contribution to the literature on DNA damage responses.

We thank the editors for taking the time to evaluate our manuscript once again and are glad to hear they found the manuscript significantly improved.

Having said all of this, it should be noted that the manuscript still requires certain clarifications in the text and/or figures. This will not require additional experimentation only text changes and clarifications that would strengthen this manuscript further and improve overall clarity.

We appreciate the remaining points and have revised the manuscript accordingly. Please find our point-by-point response below:

1) There needs to more clarity, about roles in DNA repair/cell survival away from transcriptionally active genes, of the full cellular pool of PALB2 vs the sub-fraction that is specifically associated with transcriptionally active sites. a) Do the authors propose that any PALB2 remains at transcriptionally active sites after the induction of DNA damage elsewhere and if so whether this may provide ongoing protection of active genes and b) whether the authors are proposing that a PALB2-BRCA2-RAD51 complex is mobilized from transcriptionally active genes and whether the PALB2-BRCA2-RAD51 complex exists away from transcriptionally active genes at steady state.

To address the reviewer’s point, we would like to highlight the following three observations. Firstly, the abundance of endogenous PALB2 protein is estimated to be markedly lower than that of other HR factors. For example, Kulak et al., (PMID: 24487582) estimated that there are ~150 molecules of PALB2 per cell, compared to ~10,000 for BRCA2 and ~100,000 for RAD51, in HeLa. Secondly, Xia et al., (PMID: 16793542) reported, on the basis of BRCA2 or PALB2 immunoprecipitation from fractionated cells, that ‘*the majority of nuclear structure bound BRCA2 is associated with PALB2’* and that *‘much of the PALB2 is associated with BRCA2.*’ Finally, we have found that PALB2 associates with MRG15, a chromodomain protein that recognises an epigenetic marker of active genes (i.e. H3K36me3) with high affinity (Bleuyard et al., PMID: 28673974).

Considering all these observations together, we envision that the majority of endogenous PALB2 is enriched at actively transcribed chromatin in complex with BRCA2, RAD51 and MRG15, although PALB2-free pools of BRCA2 and RAD51 must exist. Upon the induction of DNA damage at random loci, PALB2 at active chromatin is mobilised, and instead becomes enriched at damaged loci, likely through BRCA1-mediated DNA damage recognition. In this context, our experiment assessing cells expressing exogenous PALB2 variants at elevated levels might have limited sensitivity to highlight the physiological impact of the PALB2 mobility as in endogenous conditions, explaining the somewhat subtle cellular phenotypes in our study.

To specifically address point (a), our ChIP-qPCR data (Figure 9D) indicate that PALB2 occupancy at active genes is effectively reduced to null (i.e. comparable to the level in cells complemented with empty vector) following IR exposure, although we cannot totally exclude the possibility that PALB2 bound at other (untested) actively transcribed genes responds differently. Regardless, we consider it unlikely that damage-induced under enrichment of PALB2 at active genes severely impacts the transcription-replication conflict, as DNA damage would trigger the canonical DNA damage checkpoint (i.e. supressing cell cycle progression), hence slowing the progression of DNA replication (i.e. no active movement of the replication machinery). Regarding point (b), we indeed consider that the PALB2-BRCA1-RAD51 complex is mobilised from actively transcribed chromatin at which it is enriched at steady-state.

These points are clarified in the revised manuscript with the following paragraph (page 29):

“Markedly, PALB2 occupancy at active genes, as detected by ChIP-qPCR (Figure 9D), was effectively reduced to null following IR exposure (i.e. comparable to the level in cells complemented with empty vector), though it may represent only the sub-fraction of PALB2 bound to the loci tested. Regardless, these observations raise a fundamental question: why does PALB2 dissociate from active genes with the potential risk of leaving these regions unprotected? Notably, the abundance of endogenous PALB2 protein is estimated to be considerably lower than that of BRCA2 or RAD51, e.g. ~60 times less than BRCA2 and ~600 times less than RAD51 in HeLa cells (Kulak et al., 2014). The majority of PALB2 is also found to be associated with BRCA2 on nuclear structures (Xia et al., 2006) and with MRG15 (Bleuyard et al., 2017b). Hence, we envision that, in endogenous conditions,

PALB2 primarily associates with BRCA2 and RAD51 on actively transcribed loci to protect these regions (though PALB2-free pools of BRCA2 and RAD51 presumably exist), but when DNA damage occurs elsewhere, the same complex needs to be mobilised to promote HR repair. In light of these considerations, our experimental platform assessing cells expressing exogenous PALB2 variants, at higher than endogenous levels, might have somewhat compromised sensitivity to highlight the physiological impact of the PALB2 mobilisation. Either way, we envision that the under-enrichment of PALB2 at active genes is unlikely to confer severe consequences for the transcription-replication conflict, as IR would trigger the global DNA damage checkpoint, slowing replication fork progression (Lajtha et al., 1958; Ord and Stocken, 1958; Watanabe, 1974) while broken DNA is repaired.”

2) The statements about the role of the 7K-patch in HR needs to be softened and clarified how results obtained with the 7R and 7Q variants provide complementary information about PALB2's regulation and function in response to DNA damage. There were reasons for removing data from HR reporter assays, but a reader may consider conclusions about the role of the 7K-patch in HR based only on RAD51 foci to not be fully supported. As such, it is suggested that the authors briefly discuss their HR reporter assay results (in either the Results or Discussion) as data not shown and indicate why these results may not be congruent with those obtained for RAD51 foci. Finally, in the absence of HR reporter assay data, I would suggest toning down the headings in various places to less definitively link the 7K patch to HR repair (line 610; lines 1026-1027; lines 1044-1045. In general, RAD51 focus formation cannot be equalized with HR.

We appreciate this comment and accordingly included the following paragraph in the Discussion (page 30):

“In the meantime, PALB2 recruitment itself does not appear to guarantee RAD51 assembly. Both the 7R and 7Q substitutions elicited significant impairment of RAD51 foci formation (Figures 7 and 8), suggesting that the 7Kpatch promotes optimal RAD51 engagement with broken DNA, conceivably in part through its direct interaction with the nucleosome acidic patch regions exposed at damaged chromatin (Belotserkovskaya et al., 2020). It should be noted, however, that the full impact of the 7R and 7Q mutations on HR repair remains unclear. We endeavoured to assess HR proficiency of these cell lines using our reporter systems, which monitor the HR-mediated integration of a promoterless GFP gene and the resultant expression of fluorescent protein as the readout (Rodrigue et al., 2019; Yata et al., 2012). Through these approaches, significant HR defects were not detected (data not shown). This could potentially be explained by the model that the PALB2 7K-patch is particularly important for HR repair of DSBs arising outside active genes, the detection of which is not straightforward using available techniques. Further investigation using more elaborate strategies, which allow the evaluation of HR repair outside active genes, is warranted to fully appreciate the importance of PALB2 ChAM acetylation in HR repair.”

We have also toned down the headings, which had referred to ‘HR repair’, to a more accurate description (e.g. RAD51 foci formation)

3) A measure of the statistical significance of differences in Figure 4C is needed.

The statistical significance of changes between time 0 and all combined time points (15, 30, 60 and 120 mins), as assessed by the one-way ANOVA, is now reported in Figure 4C.

4) It looks like there is a non-specific band in the blot for PALB2 in Figure 6A. It would be helpful to indicate, with a marker in Figure 6A and/or by the description in the figure legend, which, if any, of the two bands is non-specific there.

We have now included the positions of molecular weight marker in Figure 6A and indicated the correct band for PALB2. The raw scan of the blot is also included in the ‘source data’.

5) The foci are often difficult to see without significant enlargement in Figures 7D and 8D. Perhaps making this figure part larger relative to the overall figure would help with this. Alternatively, the intensities could be enhanced in the images if the current relative size of the figure part is retained.

As suggested, we have enlarged these figure parts and increased the intensity to the best of our capacity.

6) There is a single statement that appears to have been carried forward from the previous version which should be removed (page 22, line 524-525) 'while suppressing non-specific DNA binding'.

We apologise for this error. This sentence has been removed.